# Conserved and divergent gene regulatory programs of the mammalian neocortex

Nathan R. Zemke[1,2,19], Ethan J. Armand[1,3,19], Wenliang Wang[4], Seoyeon Lee[1], Jingtian Zhou[3,4], Yang Eric Li[1], Hanqing Liu[4,5], Wei Tian[4], Joseph R. Nery[4], Rosa G. Castanon[4], Anna Bartlett[4], Julia K. Osteen[6], Daofeng Li[7], Xiaoyu Zhuo[7], Vincent Xu[7], Lei Chang[1], Keyi Dong[1,2], Hannah S. Indralingam[1,2], Jonathan A. Rink[6], Yang Xie[1], Michael Miller[1,2], Fenna M. Krienen[8,9], Qiangge Zhang[10,11], Naz Taskin[12], Jonathan Ting[12], Guoping Feng[10,11], Steven A. McCarroll[9,10], Edward M. Callaway[13,14], Ting Wang[7,15], Ed S. Lein[12,16], M. Margarita Behrens[6], Joseph R. Ecker[4,17 ✉] & Bing Ren[1,2,18 ✉]

Divergence of *cis*-regulatory elements drives species-specific traits[1], but how this manifests in the evolution of the neocortex at the molecular and cellular level remains unclear. Here we investigated the gene regulatory programs in the primary motor cortex of human, macaque, marmoset and mouse using single-cell multiomics assays, generating gene expression, chromatin accessibility, DNA methylome and chromosomal conformation profiles from a total of over 200,000 cells. From these data, we show evidence that divergence of transcription factor expression corresponds to species-specific epigenome landscapes. We find that conserved and divergent gene regulatory features are reflected in the evolution of the three-dimensional genome. Transposable elements contribute to nearly 80% of the human-specific candidate *cis*-regulatory elements in cortical cells. Through machine learning, we develop sequence-based predictors of candidate *cis*-regulatory elements in different species and demonstrate that the genomic regulatory syntax is highly preserved from rodents to primates. Finally, we show that epigenetic conservation combined with sequence similarity helps to uncover functional *cis*-regulatory elements and enhances our ability to interpret genetic variants contributing to neurological disease and traits.

Throughout evolution, sequence divergence in the non-coding regions of the genome is believed to be a major driving force behind the emergence of species-specific traits[1]. Today, the large number of available genome sequences of eukaryotic species enables us to use comparative genomics to map functionally important sequences under evolutionary constraints, including *cis*-regulatory elements (CREs)[2,3]. However, sequence conservation alone cannot provide definitive evidence of the functional role of a regulatory element as not all functional elements have conserved sequences, and some non-functional elements may be sequence conserved. Moreover, sequence conservation cannot reveal information about the cell- and tissue-specific activity of an element. We currently possess little knowledge of how gene regulatory programs evolve. In other words, how sequence divergence leads to altered gene expression patterns across different species remains largely unexplored. Filling this knowledge gap is critical to understanding the consequences of genetic divergence on species-specific phenotypes.

Previous bulk sequencing assays have revealed general principles concerning the conservation of CREs and tissue-specific gene expression patterns. For example, enhancers exhibit rapid turnover during mammalian evolution[4,5], and conserved enhancers have lower cell type specificity[6,7]. By contrast, sequence divergent enhancers have a substantial role in establishing tissue and species-specific traits[8,9]. Such divergent enhancers are often mediated by de novo insertion of transposable elements (TEs) carrying clusters of transcription-factor-binding sites[6,10,11]. Notably, the conservation of CREs[12,13] and expression[14] generally decreases as development progresses. Despite the established divergence of *cis*-regulatory sequences throughout evolution, DNA motifs recognized by sequence-specific DNA binding proteins are highly conserved[15],

[1]Department of Cellular and Molecular Medicine, University of California, San Diego School of Medicine, La Jolla, CA, USA. [2]Center for Epigenomics, University of California, San Diego School of Medicine, La Jolla, CA, USA. [3]Bioinformatics and Systems Biology Program, University of California, San Diego, La Jolla, CA, USA. [4]Genomic Analysis Laboratory, The Salk Institute for Biological Studies, La Jolla, CA, USA. [5]Division of Biological Sciences, University of California, San Diego, La Jolla, CA, USA. [6]Computational Neurobiology Laboratory, The Salk Institute for Biological Studies, La Jolla, CA, USA. [7]Department of Genetics, The Edison Family Center for Genome Sciences & Systems Biology, Washington University School of Medicine, St Louis, MO, USA. [8]Princeton Neuroscience Institute, Princeton University, Princeton, NJ, USA. [9]Department of Genetics, Harvard Medical School, Boston, USA. [10]Stanley Center for Psychiatric Research, Broad Institute of MIT and Harvard, Cambridge, MA, USA. [11]McGovern Institute for Brain Research, Department of Brain and Cognitive Sciences, Massachusetts Institute of Technology, Cambridge, MA, USA. [12]Allen Institute for Brain Science, Seattle, WA, USA. [13]Systems Neurobiology Laboratories, The Salk Institute for Biological Studies, La Jolla, CA, USA. [14]Department of Neurosciences, University of California San Diego, La Jolla, CA, USA. [15]McDonnell Genome Institute, Washington University School of Medicine, St. Louis, MO, USA. [16]Department of Neurological Surgery, University of Washington, Seattle, WA, USA. [17]Howard Hughes Medical Institute, The Salk Institute for Biological Studies, La Jolla, CA, USA. [18]Institute of Genomic Medicine, Moores Cancer Center, School of Medicine, University of California San Diego, La Jolla, CA, USA. [19]These authors contributed equally: Nathan R. Zemke, Ethan J. Armand. ✉e-mail: ecker@salk.edu; biren@ucsd.edu

suggesting the existence of a conserved flexible genomic regulatory syntax. To characterize such gene regulatory syntax, it is important to perform an integrated analysis of chromatin landscape, structure and gene expression in a cell-type-specific manner across multiple species.

The primary motor cortex (M1) is a region of the neocortex that is preserved across eutherian mammals that is critical for volitional fine motor movements[16]. Recently, several reports, part of the BRAIN Initiative Cell Census Network, have characterized the vast complexity of cellular taxonomy, gene expression and epigenome of brain cells in multiple mammalian species[17–21]. A single-cell comparative analysis of the mouse, marmoset and human transcriptomes of M1 cells revealed a high degree of species-specific marker gene expression[18]. However, our understanding of how genome evolution influences species-specific gene expression remains limited. We therefore examined whether sequence divergence at non-coding CREs is associated with driving species-specific biology through the evolution of gene regulatory programs.

Here we sought to characterize the evolution of gene regulatory programs by performing comparative epigenomic analyses. Specifically, we performed single-cell multiomics assays on brain tissue from mouse, marmoset, macaque and human, profiling four different molecular modalities: gene expression, chromatin accessibility, DNA methylation and chromatin conformation. In doing so, we mapped candidate CREs (cCREs), and profiled their dynamic epigenetic states across 21 brain cell types in the M1 from four species. To use these data for our comparative study, we developed a framework for assessing the evolution of gene regulatory features and demonstrated co-evolution of the epigenome and three-dimensional (3D) genome with the transcriptome. Although not all cCREs contribute to gene expression in the same manner, epigenetically conserved cCREs are more likely to activate gene expression and contain disease-risk-associated variants. Species-biased cCREs that are predicted to regulate gene expression are more likely to contribute to divergent gene expression. Genome browser tracks are publicly available for viewing at the WashU Comparative Epigenome Browser data hub (https://epigenome.wustl.edu/BrainComparativeEpigenome/).

## Single-cell assays of the M1 in four mammals

To gain a detailed picture of how gene regulatory programs evolve, we performed a comparative epigenomics study in the M1 of human, macaque, marmoset and mouse (Fig. 1a). Two single-nucleus genomics assays were used—10x multiome (10x Genomics) and snm3C-seq[22,23] (also known as single-cell methyl-Hi-C[22,23])—to simultaneously profile the transcriptome with chromatin accessibility in the same cell and DNA methylation with 3D genome in the same cell, respectively. We profiled 40,937 human nuclei, 34,773 macaque nuclei, 34,310 marmoset nuclei and 47,404 mouse nuclei using 10x multiome, and 8,198 human nuclei, 5,737 macaque nuclei, 4,999 marmoset nuclei and 5,349 mouse nuclei using snm3C-seq (Fig. 1b). We next performed unsupervised clustering on the basis of gene expression or DNA methylation, and integrated datasets across species using orthologous genes as features (Fig. 1b and Extended Data Fig. 1a,b). Cell types were identified at the subclass resolution using a combination of marker-gene activity and reference mapping to the available M1 datasets from mouse, marmoset and human[17,18]. Although we identified each cell type in all four species, cell type fractions were highly species specific (Fig. 1c). Notably, an expansion of the oligodendrocyte proportion and a reduction in the excitatory neuron proportion was observed from mouse to human, consistent with previous reports[24] (Fig. 1c). Specific subclasses of excitatory and inhibitory neurons were enriched in human, such as cortical layer 6 (L6) intratelencephalic CAR3 (L6 IT CAR3), Chandelier and VIP neurons, while L5/6 near projecting, L5 IT, L5 ET and PVALB neurons were consistently

lower in human donors (Fig. 1d). Our data reveal evolutionary divergence of cell type composition in mammalian M1, demonstrating the necessity for cell-type-resolved data for cross-species comparative analysis. For downstream analyses, we combined sequencing reads for each cell type, resulting in species- and cell-type-resolved epigenome and transcriptome landscapes for each molecular modality (Fig. 1e).

## Comparing gene expression across species

We evaluated the divergence and conservation of transcription between species for each gene identified as one-to-one orthologues in all four species. We defined gene expression conservation as the ability to predict the expression level of a gene in a specific cell type, given the expression level of the same cell type in a different species. To account for the dependence relationships between cell types, we used generalized least squares (GLS) regression[25] for each pair of species (Methods) (Fig. 1f, Extended Data Fig. 2a and Supplementary Table 1). We observed considerable correspondence between gene expression conservation and the average PhastCons score across a gene's exons (Extended Data Fig. 2b). Next, to assess the divergence of gene expression, we performed differential expression analysis using edgeR for each cell type between each species pair[26] (Fig. 1g and Supplementary Table 2). We identified species-biased genes that are differentially upregulated in a single cell type compared with in each other species (Methods).

Of 13,822 gene orthologues expressed in at least one of the four species, we identified 2,689 (~20%) mammal-conserved genes with similar patterns of expression across cell types in all four species (Fig. 1h and Extended Data Fig. 2c). We also identified 2,638 (~20%) genes with conserved patterns of expression only among primates (Supplementary Table 3). Across species, we identified 3,511 (~25%) genes with species-biased expression patterns, finding that the number of biased genes is concordant with evolutionary diversity (human, 1,376; macaque, 451; marmoset, 638; mouse, 1,367) (Fig. 1i, Extended Data Fig. 2c and Supplementary Table 3).

We noted that the majority of mammal conserved genes displayed broad gene expression patterns across cell types and we therefore further divided them into categories of ubiquitous and non-ubiquitous on the basis of cell type specificity of expression (Methods and Extended Data Fig. 2d). To identify biological processes displaying high levels of conservation and divergence, we next performed Gene Ontology (GO) enrichment analysis. Ubiquitous mammal-conserved genes were most enriched for GO categories related to the regulation of protein expression, such as ubiquitin-dependent catabolic processes and mRNA processing (Extended Data Fig. 2e). Non-ubiquitous mammal-conserved genes showed enrichment for transcriptional regulation through RNA polymerase II and DNA-templated regulation, nervous system development and cation channel regulation (Fig. 1j). These genes also showed highly correlated patterns of expression across cell types (Extended Data Fig. 2c). Among primate-conserved genes, the number of ubiquitously expressed genes dropped considerably (Extended Data Fig. 2d), and these genes were enriched for translational processes (Extended Data Fig. 2f). Among non-ubiquitous genes, we saw strong enrichment for neuronal functions such as synaptic transmission and axonogenesis (Extended Data Fig. 2g). These differences in enrichment suggest different targets of functional conservation at different evolutionary timescales, with the stronger selection placed on genes that regulate many functions over genes encoding cell-type-specific functions, consistent with previous research[27].

Among non-ubiquitously expressed human-biased genes, we found the highest enrichment for the GO term extracellular matrix organization (Fig. 1k), which is known to be crucial for diverse aspects of neural development[28]. GO enrichment analysis for human-biased genes

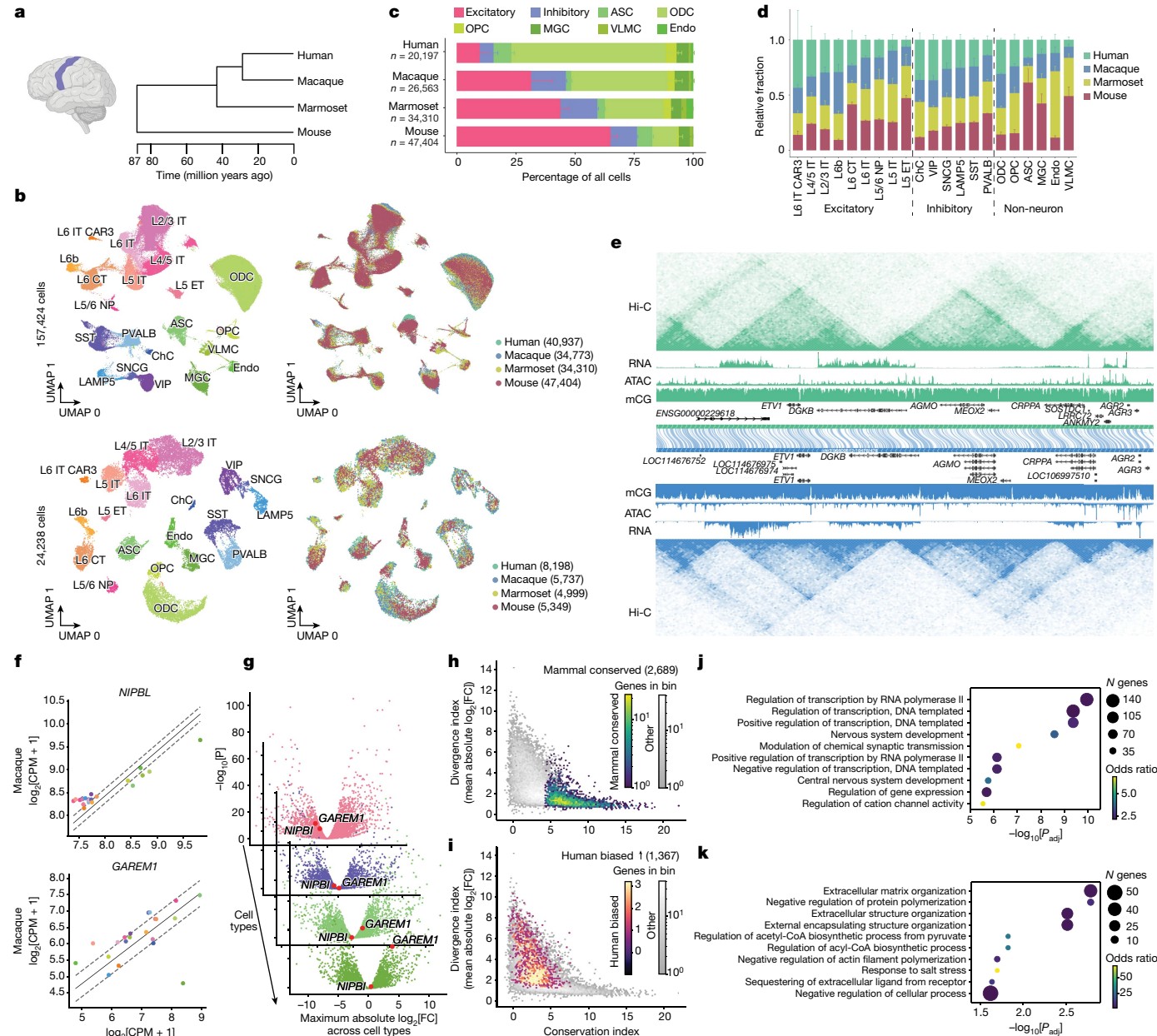

**Fig. 1 | Cross-species evolutionary comparison of single-cell multiomics analysis of the M1. a**, Illustration of the human M1 (left), created using BioRender. Dendrogram representing the evolutionary distance of each species in our study (right) from TimeTree[69]. **b**, Uniform manifold approximation and projection (UMAP)[70] embeddings of 10x multiome RNA (top) and snm3C-seq DNA methylation (bottom) clustering, annotated by cell type (left) or species (right). The numbers in parentheses indicate the cell counts from each species. NP, near projecting neurons. ChC, Chandelier neurons. **c**, The fraction of each cell type from unbiased nucleus-sorted samples. Data are mean ± s.d. across donors (primates, $n = 3$) or pools (mouse, $n = 8$). **d**, The relative abundance between species for each cell type among its particular class (excitatory, inhibitory or non-neuron). Data are mean ± s.d. **e**, The WashU Comparative Epigenome Browser displaying an alignment between human (hg38; top, green) and macaque (rheMac10; bottom, blue) genomes with L2/3 IT excitatory data tracks for Hi-C, RNA, assay for transposase-accessible chromatin (ATAC) and mCG. **f**, The conservation index, showing GLS regression for *NIPBL* (GLS T statistic = 15.460) and *GAREM1* (GLS T statistic = 3.673) between human and macaque coloured by cell type. The error bars indicate the 95% confidence interval calculated using GLS regression. **g**, The divergence index, showing differential expression between human and macaque in L5 IT neurons, PVALB neurons, ASCs and MGCs. *NIPBL* and *GAREM1* are shown in red. FC, fold change. **h**, The relationship between the average conservation index across species and the average divergence index across species. Mammal-conserved genes are highlighted. **i**, The relationship between the average conservation index across species and the average divergence index across species. Human-biased genes are highlighted. **j**, Top significant GO analysis terms for non-ubiquitous mammal-conserved genes. $P_{adj}$, adjusted $P$. **k**, Top significant GO analysis terms for non-ubiquitous human-biased genes in any cell type.

from individual cell types further revealed diverse functional terms (Extended Data Fig. 2h–k). For example, the human-biased genes from L5/6 near-projecting neurons were enriched in triglyceride catabolic process, while human-biased genes from oligodendrocyte precursor cells (OPCs) were related to negative regulation of blood vessel morphogenesis (Extended Data Fig. 2h–j). This analysis highlights the diversity of human-specific functions among motor cortex cell types.

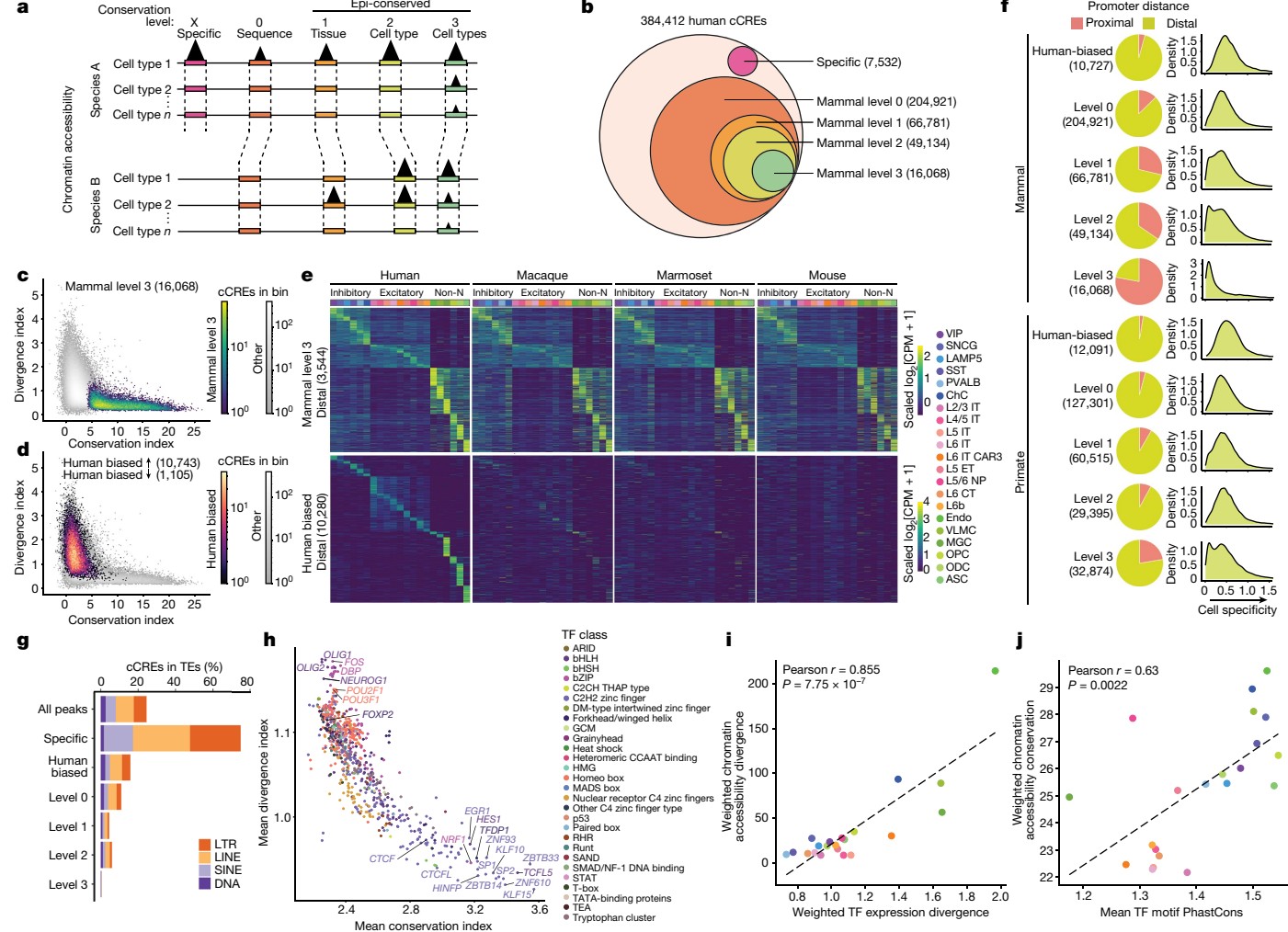

**Fig. 2 | Comparative analysis of chromatin accessibility across species.**
**a**, The levels of conservation for ATAC–seq peaks. **b**, Human cCREs from
ATAC–seq peaks for each indicated group for human-specific, level 0 (sequence
conserved), level 1 (tissue conserved), level 2 (cell type conserved) and level 3
(matched patterns across all the cell types) across mammals. **c**, The relationship
between the average conservation index (*x* axis) and the average divergence
index across species (*y* axis). The density of all mammal-conserved gene cCREs
is highlighted. **d**, The relationship between the average conservation index
across species (*x* axis) and the average divergence index across species (*y* axis) for
each level 0 (sequence conserved) peak. Human-biased peaks are highlighted.
**e**, Heat maps ordered by cell type with highest signal for mammal level 3 distal
cCREs (top) and human-biased distal cCREs (bottom). For visualization of cell
type patterns of accessibility, $\log_2$[counts per million (CPM) + 1] values are row

scaled. Non-N, non-neuronal. **f**, The proportion of promoter-proximal (≤1 kb from
a TSS) or promoter distal (>1 kb from a TSS) cCREs for the indicated group (left).
The density plots show the cell specificity scores (Methods) for cCREs in each
group. **g**, The percentage of human cCREs in TEs for different conservation
groups. **h**, The average conservation and divergence index for all cCREs
containing a given TF motif from JASPAR CORE. Motifs are coloured by TF class.
**i**, The weighted cell type chromatin accessibility divergence across distal
cCREs as a function of weighted cell type TF divergence for each cell type.
Distal cCREs and TF genes were weighted by CPM. **j**, The weighted chromatin
accessibility conservation in distal peaks as a function of weighted cell type
PhastCons among distal peaks. For **i** and **j**, *P* values were calculated using
two-sided Pearson correlation.

## Comparative analysis of chromatin accessibility across species

To identify cCREs on the basis of chromatin accessibility data, we
determined the open chromatin regions in each motor cortex cell
type in each species using MACS2[29] and identified 384,412 human,
336,463 macaque, 281,297 marmoset and 333,814 mouse cCREs that
display accessibility in one or more cell types (Extended Data Fig. 1b,c
and Supplementary Tables 4–7). We used liftOver[30] to classify human
cCREs as human-specific, mammalian sequence conserved (from
human to mouse) or primate sequence conserved (from human to
marmoset) (Extended Data Fig. 3a–c and Supplementary table 8). For
our comparative epigenomic analysis, we defined levels of conserva-
tion on the basis of sequence and activity (Fig. 2a and Extended Data

Fig. 3a–c). Species-specific cCREs contain sequences that have not been
identified in the other three species. Level 0 elements are cCREs with
orthologous sequences across all four mammals (mammal level 0) or
all three primates (primate level 0) (Fig. 2a). Moreover, we performed
epigenetic conservation analysis of the level 0 cCREs and defined three
levels of epigenetic conservation (epi-conservation) (Fig. 2a): level 1
are tissue-conserved cCREs with a peak called across species regardless
of which cell type; level 2 are cCREs displaying accessibility in at least
one of the same cell types across species; and level 3 are cCREs with
matching patterns of chromatin accessibility across all the cell types, as
measured to be significant (Benjamini–Yekutieli-adjusted[31] *P* < 0.05) by
GLS, in all of the species (Methods). Analogous to species-biased genes,
species-biased cCREs are defined as peaks with differential accessibility
that are consistently higher in one species compared with in the three

other species in the same cell type as identified through differential accessibility analysis performed using EdgeR[26] (Methods).

Of the 384,412 human cCREs, 7,532 cCREs (~2%) represented sequences found only in human, while 204,921 (~53%) shared orthologous sequences in the three other species (mammal level 0) (Fig. 2b and Extended Data Fig. 3d). An additional 127,301 human cCREs (~33%) shared orthologous sequences only among the three primates (primate level 0) (Extended Data Fig. 3d and Supplementary Table 8). Of the 204,921 mammal level 0 cCREs, 66,781 (32.5%) were classified as level 1, 49,134 (24.0%) as level 2 and 16,068 (7.8%) as level 3 epi-conservation (Fig. 2b,c and Extended Data Fig. 3d–g). Considering conservation among just primates, we identified an additional 127,301 cCREs that are primate level 0, 60,515 primate level 1, 29,395 primate level 2 and 32,874 primate level 3 that are not conserved in mice (Extended Data Fig. 3d,e,h and Supplementary Tables 8 and 9). Finally, 10,743 cCREs showed human-biased accessibility in one or more cortical cell type compared with all three other mammals (Fig. 2d and Extended Data Fig. 3f,g), and an additional 12,091 human-biased cCREs compared with only two other primates (Extended Data Fig. 3h and Supplementary Table 10a–f).

A large proportion of epi-conserved cCREs was found to be promoter proximal (<1 kb from a TSS), and this number increased along with the conservation level for both mammal and primate comparisons (Fig. 2f and Supplementary Table 8). Notably, mammal level 3 conserved cCREs have a considerably higher fraction of promoter-proximal cCREs compared with the primate level 3 conserved cCREs. This suggests that the turnover rates are different between the promoter-proximal and distal cCREs during evolution, whereby distal cCREs have lower evolutionary constraint than proximal cCREs. Furthermore, level 3 conserved cCREs that are promoter distal have reduced cell type specificity compared with level 0 cCREs ($P < 2 \times 10^{-16}$, $t$-test) (Fig. 2e,f), suggesting that distal cCREs with broad cell type activity have higher constraint than cell-type-specific distal cCREs.

TEs have been proposed to be a driver of genomic diversity as they can contain CREs[11,32]. To characterize the extent of TE contribution to epi-divergence, we calculated the percentage of cCREs within TEs from different conservation levels (Fig. 2g and Supplementary Table 8). We found that 75% of human-specific cCREs and 16% of human-biased cCREs are located within TEs. By contrast, less than 1% of mammal level 3 cCREs are located within TEs. Particularly, LINE-1 and LINE-2 are the most common TEs containing cCREs, which are most active in excitatory neurons (Extended Data Fig. 4a). However, human-specific cCREs in different cell types are enriched in different types of TEs. Human-specific cCREs from IT excitatory neurons had the highest overlap with LINE-1, while glial cells (microglia cells (MGCs), OPCs, oligodendrocytes (ODCs) and astrocytes (ASCs)) had the highest overlap with endogenous retrovirus-1 (ERV1) and endogenous retrovirus-K (ERVK) long terminal repeats (LTRs) (Extended Data Fig. 4a,b). Similar to human, mouse-specific cCREs were highly enriched in TEs and were depleted as epi-conservation increased (Extended Data Fig. 4c). Mouse cCREs were enriched for different types of LTRs and short interspersed nuclear elements (SINEs) compared with those observed in humans (Extended Data Fig. 4d). Our results provide further evidence that organisms may co-opt TEs to achieve species-specific and cell-type-specific gene regulation.

We next performed transcription factor (TF) motif analysis to identify factors contributing to conserved and divergent human cCREs. We calculated the mean conservation and divergent indexes for all promoter distal cCREs containing each non-redundant TF motif from the JASPAR CORE vertebrate database ($n = 791$ motifs). We found that motifs belonging to the C2H2 zinc-finger class of TFs were present in distal cCREs with the highest epi-conservation. This included Krüppel-like factor family members, ZBTB33 and ZNF610, and insulator-binding factor CTCF (Fig. 2h). Despite mostly belonging to the same TF class, the top epi-conserved TF motifs come from diverse families and subfamilies with non-redundant binding sequences. By contrast, distal cCREs with bHLH, bZIP and homeodomain-containing TF motifs had the highest epi-divergence in human distal cCREs. Among them were the lineage-specific TFs OLIG1, OLIG2, NEUROG1, POU2F1 and POU3F1, which are known to have a role in neural development[33–35]. Although the homeodomain TF motifs have low epi-conservation in adult cortex, their epi-conservation may be high during earlier stages of development, given their important roles in embryogenesis[36]. Consistent with this, they possessed high sequence conservation despite low epi-conservation (Extended Data Fig. 3i).

To assess the degree to which divergent expression of TFs contributes to divergent epigenomic landscapes, we checked for a correlation between epigenome divergence with weighted TF expression divergence across cell types (Fig. 2i). From this analysis, we observed a high degree of correlation (Pearson $r = 0.855$, $P = 7.75 \times 10^{-7}$), suggesting that TF expression divergence may greatly influence the evolution of gene regulatory landscapes. Moreover, sequence divergence can result in the loss or gain of gene regulatory elements during evolution. To test whether the conservation of TF motifs in cCREs correlates with epigenome conservation, we checked for a correlation between the mean PhastCons score for every TF motif in human cCREs and the mean epi-conservation index for all human cCREs across cell types (Fig. 2j), for which we observed a significant correlation (Pearson $r = 0.63$, $P = 0.0022$). A significant positive correlation was also observed for individual elements (Extended Data Fig. 3j). Taken together, these results imply that changes in the expression of TFs and TF-motif sequences during evolution have a role in shaping species-specific cCRE use.

## Comparison of DNA methylomes across species

We further investigated epigenetic evolution by examining differentially methylated regions (DMRs) across all four species. Previous studies have indicated that DMRs are enriched for cCREs[21,37]. For each species, we called differentially methylated CG regions between cell types using methylpy[38], identifying 1,361,958 human, 1,661,598 macaque, 1,066,980 marmoset and 1,748,945 mouse DMRs (Extended Data Fig. 1b,c and Supplementary Tables 11–14). We next identified conserved DMRs across species by repeating the same analysis described for chromatin-accessible cCREs (Fig. 2a and Extended Data Fig. 3a,b).

In terms of conserved sequences, we identified 54,829 human-specific DMR sequences (4.0%), 579,026 (42%) sequences were orthologous in all four mammalian species (mammal level 0) and 519,456 (38%) sequences were found in all three primates but not in mouse (Fig. 3a, Extended Data Fig. 5a and Supplementary Table 15). Of the mammal level 0 DMRs, we found 195,435 (14.3%) as level 1, 144,156 (10.6%) as level 2 and 23,414 (1.72%) as level 3 in which patterns of cell type CG hypomethylation are highly conserved (Fig. 3a and Extended Data Fig. 5a,b,f). Conservation of CG methylation (mCG) showed strong correlation with conservation of chromatin accessibility (Fig. 3b). Primate epi-conserved elements were identified from the mammal and primate level 0 cCREs (Extended Data Fig. 5a,b). We found an additional 201,415 DMRs (14.8%) as level 1, 199,555 (14.65%) as level 2 and 64,138 (4.7%) as level 3 that were epi-conserved in primates but not in mice (Extended Data Fig. 5a,b,f and Supplementary Tables 15 and 16).

Compared with chromatin-accessible cCREs, DMRs are much less promoter proximal (Extended Data Fig. 5g), probably due to ubiquitous promoter hypomethylation across cell types. However, they still displayed an increasing enrichment for transcription start site (TSS) proximity with higher levels of conservation (Fisher's exact test, $P < 2 \times 10^{-16}$) (Extended Data Fig. 5c,g). Much like chromatin-accessible cCREs, promoter-distal mammal and primate level 3 DMRs showed lower cell type specificity compared with level 0 DMRs ($P < 2 \times 10^{-16}$, $t$-test) (Extended Data Fig. 5g).

We again evaluated the contribution of TEs to DNA methylome evolution by evaluating the proportion of DMRs in TEs across different

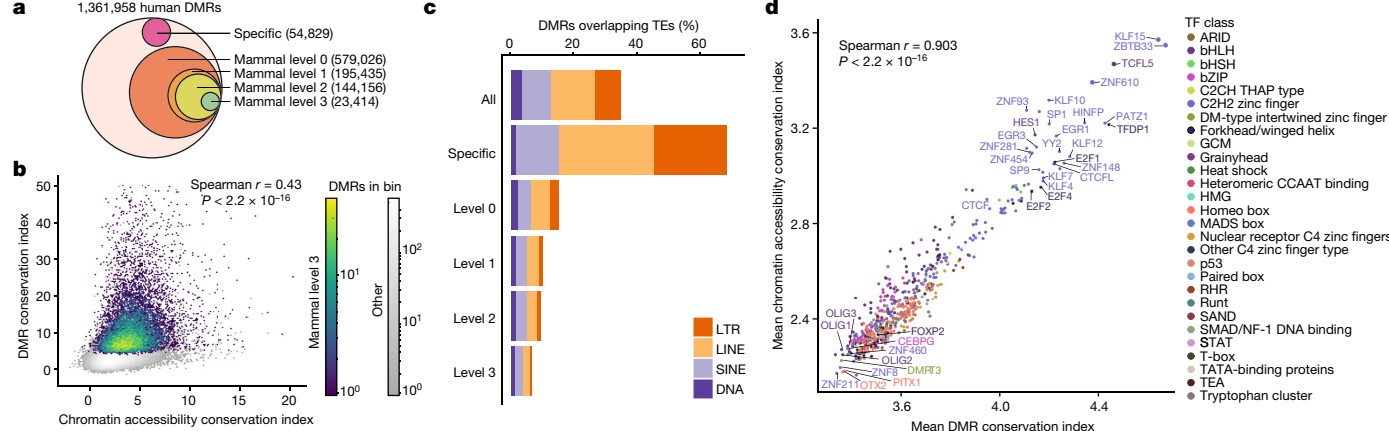

**Fig. 3 | Comparative analysis of DNA methylomes across species.**
**a**, The conservation levels of human DMRs, including human-specific (sequence divergent), level 0 (sequence conserved), level 1 (tissue conserved), level 2 (cell type conserved) and level 3 (matched patterns across all cell types) across mammals. **b**, The relationship between the conservation index (mean GLS *T*-statistic across comparisons) of chromatin accessibility, and the conservation index of DNA methylation for intersecting ATAC–seq peaks and DMRs. *P* values were calculated using two-sided Spearman correlation. **c**, The percentage of human DMRs in TEs for different conservation groups. **d**, Comparison of the average conservation index of DMRs and ATAC–seq peaks containing a particular motif from the JASPAR CORE database. Motifs are coloured by TF class annotation.

levels of conservation. DMRs showed the same pattern of decreasing TE enrichment with increasing conservation as chromatin-accessible cCREs, with human-specific sequences showing enrichment for TEs (69%), and mammal level 3 conserved sequences showing depletion (6.5%) (Fig. 3c and Supplementary Table 15). Similar trends were observed for primate-conserved DMRs (Extended Data Fig. 5d). To evaluate the contribution of TEs to DMRs across cell types we identified the proportion of various TE classes to hypomethylated DMRs in each cell type (Extended Data Fig. 5e). While both chromatin-accessible cCREs and DMRs showed enrichment of LINE-1 elements, the proportion of hypomethylated DMRs overlapping LINE-1 elements was relatively consistent across cell types. By contrast, DMRs in Alu elements had modest enrichment among excitatory neurons, which was absent among chromatin-accessible cCREs. Human-specific hypomethylated DMRs in Alu elements showed a preference for neurons (Extended Data Fig. 5e). Like chromatin accessibility, human-specific DMRs showed depletion in certain classes of elements, and a notable increase in ERVK elements.

We again performed TF-motif analysis to identify factors contributing to conserved DMRs. We calculated the mean conservation indexes for all promoter distal DMRs containing each non-redundant TF motif from the JASPAR CORE vertebrate database. Comparing the conservation of motifs in DMRs to their conservation in chromatin-accessible cCREs showed a marked identity between the sequence drivers of conservation between the two modalities (Fig. 3d). Lastly, we intersected DMRs with chromatin-accessible cCREs and found that DNA methylation was relatively high at human-specific cCREs and decreased as chromatin accessibility conservation increased (Extended Data Fig. 5h).

## Comparing 3D genomes across species

The genome is organized into topologically associating domains (TADs) that influence gene expression by constraining chromatin contacts between promoters and CREs[39]. To characterize the conservation of 3D genome organization, we compared TAD boundary elements across species (Fig. 4a and Supplementary Tables 17–20). Most human boundaries had a conserved sequence in at least one other species, with 12,641 identified as conserved in all four species (mammal level 0) and a further 12,960 having conserved a sequence among primates (primate level 0) (Extended Data Fig. 6a). Of the mammal level 0 (sequenced

conserved) boundaries, we identified 40% (5,118) found in all four species from any cell type (mammal level 1, tissue-level conserved), and 10% (1,290) conserved in any of the same cell types in all four species (mammal level 2) (Fig. 4b, Extended Data Fig. 6a,b and Supplementary Table 21). In total, 859 human boundaries were sequence specific in human (Fig. 4b and Extended Data Figs. 6a) and 1,653 were called only in human (human biased) (Fig. 4c). Across cell types, we observed differences between the ratio of divergent to conserved TAD boundaries. For example, all neurons contained more mammal level 2 boundaries than human-biased boundaries, with cortical layer IT excitatory neurons having the highest (Fig. 4c). The reverse was true for non-neurons (with the exception of OPCs), with MGCs containing the highest proportion of human-biased boundaries (Fig. 4c). Genes associated with divergent boundaries have increased divergent expression (Extended Data Fig. 6c), which could be a consequence of or contribution to divergent boundary insulation.

It has been demonstrated that CTCF binding is necessary for maintaining and establishing TAD boundary insulation in a dosage-dependent manner[40,41]. We observed that the number of CTCF motifs and CTCF chromatin immunoprecipitation–sequencing peaks from human cortex ENCODE data were higher in mammal level 2 human boundaries than in human-biased boundaries (Extended Data Fig. 6d,e). However, there was no reduction in CTCF motifs in macaque, marmoset or mouse in orthologous regions of human-biased boundaries to explain the human-biased insulation. It has also been demonstrated that CTCF binding is interrupted at methylated binding sites[42]. We found that the regions near CTCF binding sites in human-biased boundaries have a higher proportion of methylated CGs in non-human species compared with in human; however, the average proportion at the precise CTCF motifs was low in all species (Extended Data Fig. 6f), suggesting alternative mechanisms for divergent TAD boundary formation.

To determine whether TEs are differentially enriched between divergent and conserved boundary elements, we calculated the percentage of boundaries containing TEs for each conservation group. Notably, 67% of human-specific boundary elements contain TEs; by contrast, 32% of mammal level 2 boundaries contain TEs (Fig. 4d). Particularly, LINE-1 and Alu elements had the highest enrichment in human boundaries, and human-specific boundaries were most enriched for LINE-1, Alu and ERV1 TEs (Extended Data Fig. 6g). It has previously been reported that TEs, especially Alu and SINE elements containing CTCF-binding sites,

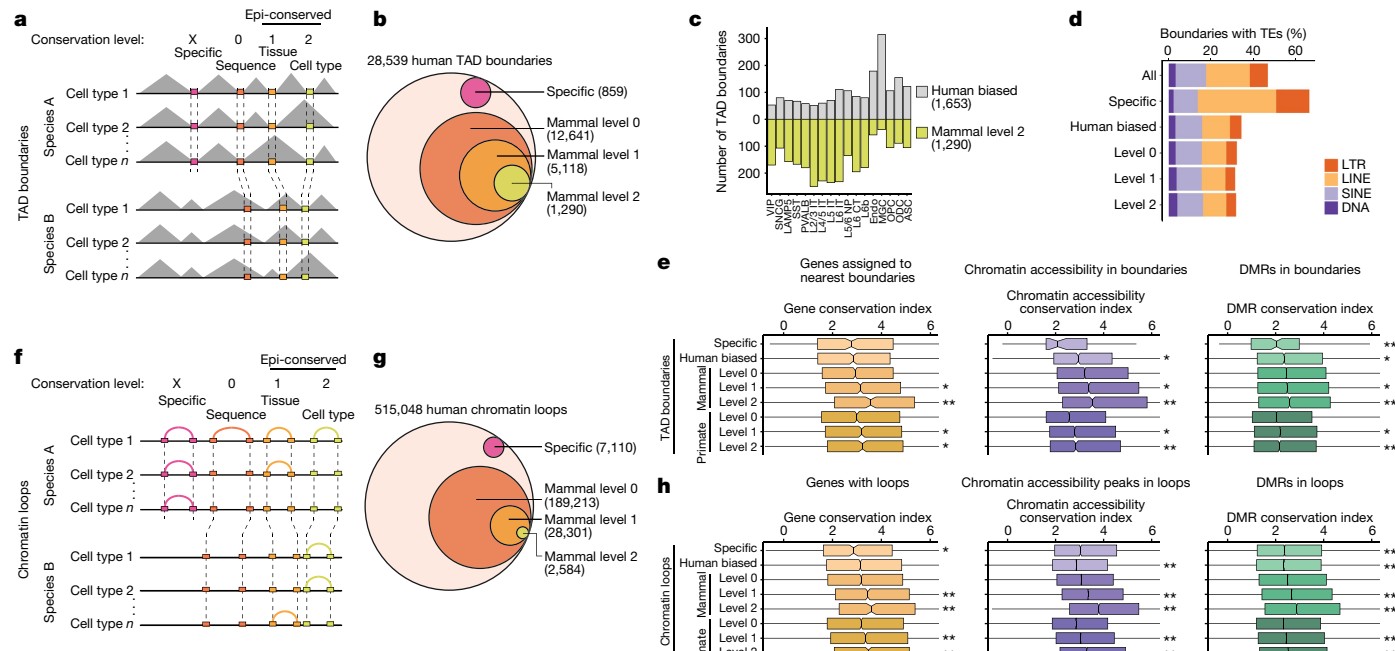

**Fig. 4 | Comparative analysis of TAD boundaries and chromatin loops across species. a**, The levels of conservation for TAD boundaries. **b**, The conservation levels of human TAD boundaries, including human-specific (sequence divergent), level 0 (sequence conserved), level 1 (tissue conserved) and level 2 (cell type conserved), across mammals **c**, The number of human-biased (conserved sequence, called only in human) and mammal level 2 (cell type conserved in all four species) TAD boundaries in each cell type. **d**, The percentage of boundaries overlapping TEs for different conservation groups. **e**, The conservation index of gene expression, ATAC–seq peaks and DMRs associated with boundaries of the indicated conservation level. 'Primate conserved' excludes mammal conserved. *P* values were calculated using two-sided unpaired Wilcoxon rank-sum tests comparing with mammal level 0, except for primate level 1 and 2, which were compared with primate level 0; *$P < 0.05$, **$P < 0.001$. Sample sizes are reported in Supplementary Table 34. **f**, The levels of conservation of chromatin loops. **g**, The conservation level of human chromatin loops, highlighting human-specific, level 0 (sequence conserved), level 1 (tissue conserved) and level 2 (cell type conserved), across mammals. **h**, The conservation index of gene expression, ATAC–seq peaks and DMRs overlapping at least one anchor of loops for each indicated conservation level. 'Primate conserved' excludes mammal conserved. *P* values were calculated as described for **e**. Sample sizes are reported in Supplementary Table 34. For the box plots in **e** and **h**, the centre line represents the median, the box limits encompass the 25th to 75th percentiles and the whiskers represent 1.5× the interquartile range.

may be a mechanism for the evolution of chromatin organization in different species[43]. More recently, several studies showed that the LTR family of TEs also induces the formation of TAD boundaries in specific cell types and developmental stages[44,45].

We also checked for a connection between conservation of the 3D genome with conservation of gene expression and the epigenome. Importantly, we found that genes near conserved boundaries have more conserved expression compared with genes near divergent boundaries (Fig. 4e). Chromatin-accessible cCREs and DMRs within conserved boundaries similarly have more conserved epigenetic states compared with those within divergent boundaries (Fig. 4e). Our results suggest a co-evolution of the 3D genome along with epigenome and gene expression, whereby evolutionary constraints are placed onto the 3D genome to preserve gene expression.

We next classified chromatin loops by conservation levels (Fig. 4f and Supplementary Tables 22–25). Compared with boundary elements, a lower fraction of loops was identified as conserved across mammals and primates (Fig. 4g, Extended Data Fig. 7a,b and Supplementary Table 26). We observed high correspondence of overlap between chromatin-accessible cCREs and loops between cell types (Extended Data Fig. 7c). Conserved loops were more likely to contain promoters with conserved expression, and chromatin-accessible cCREs and DMRs with conserved activity (Fig. 4h), suggesting that conserved 3D chromatin interactions maintain the conservation of gene regulatory functions.

We further characterized loops in each conservation group by calculating the percentage overlap with TSSs and boundaries. Conserved loops had a higher percentage containing TSSs and a higher percentage containing boundaries that correlated with conservation level (Extended Data Fig. 7d,e). Moreover, the anchor-to-anchor loop distance decreased with increasing conservation group (Extended Data Fig. 7f), suggesting that shorter-distance loops are more likely to be preserved through evolution, potentially due to a greater chance that both anchors will be retained in the same syntenic region. Our analysis demonstrates the marked concordance between the conservation of the 3D genome with gene regulatory programs, suggesting that selective pressure on genome organization maintains conserved gene regulation throughout mammalian evolution.

## Divergent cCREs and evolution of expression

We next examined how epigenetic divergence at cCREs correlates with the evolution of gene expression programs in different species. We first predicted putative enhancers and their target genes for cCREs using the activity-by-contact (ABC) model[46], using our chromatin accessibility and chromatin contact data for each cell type (Extended Data Fig. 8a and Supplementary Tables 27–30). We found that human-specific cCREs were greatly depleted of putative enhancers (Extended Data Fig. 8b). However, we also noticed that the sequences at human-specific cCREs had a considerable drop in read mappability due to the presence of repetitive elements (Extended Data Fig. 9a). As mappability correlated with read counts (Extended Data Fig. 9b,c), reduced read mappability can negatively impact ABC scoring. We addressed this by normalizing chromatin accessibility for mappability[47] at every cCRE (Extended Data

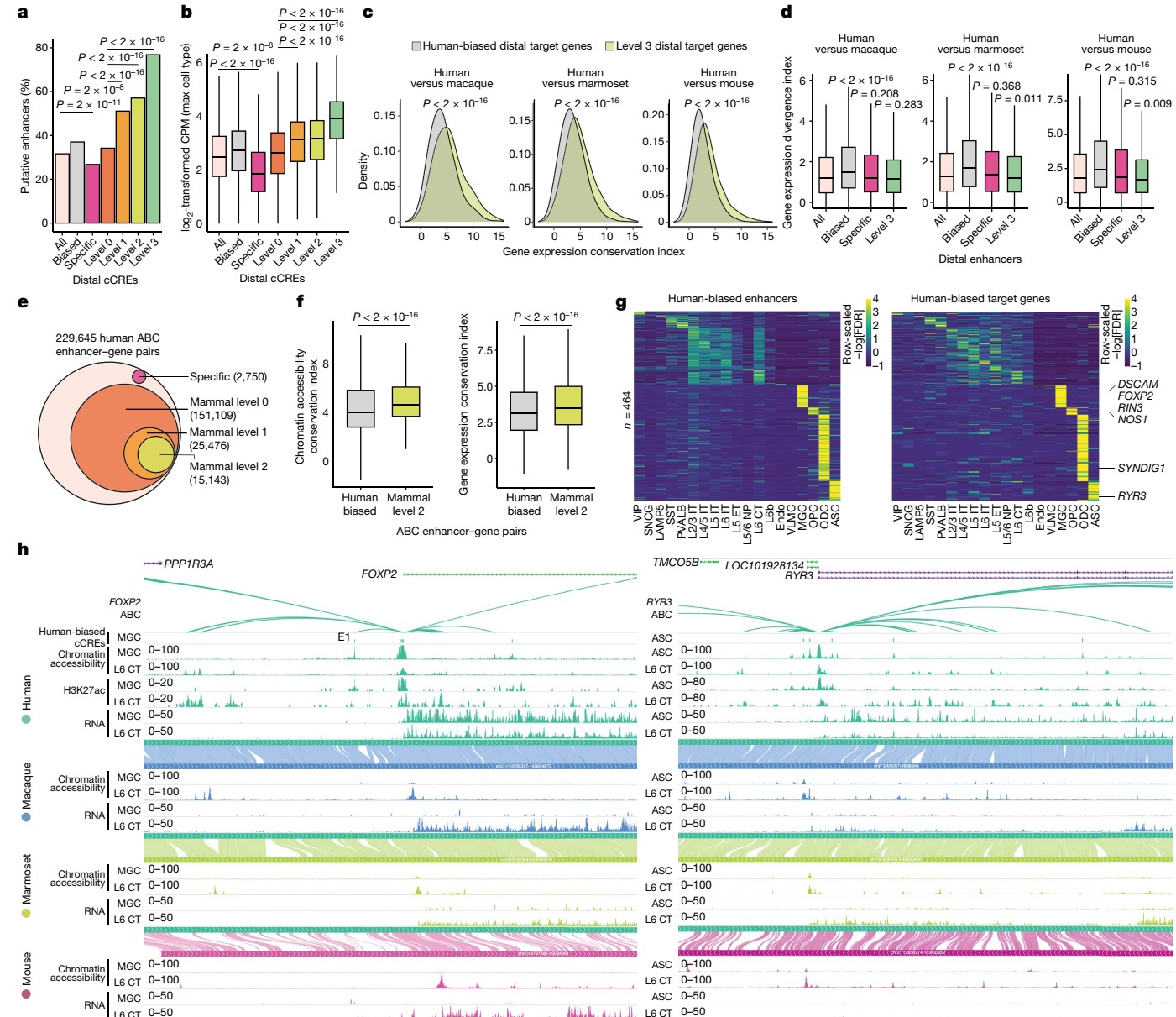

**Fig. 5 | Epigenetic conservation at cCREs is correlated with conservation in expression of their putative target genes. a**, The percentage of peaks predicted to be enhancers (ABC score ≥ 0.02) with mappability-normalized chromatin accessibility. $P$ values were calculated using $\chi^2$ tests. **b**, The mappability-normalized H3K27ac $\log_2$-transformed CPM within ±2 kb of cCRE centres for the specified groups. $P$ values were calculated using two-sided unpaired Wilcoxon rank-sum tests. Sample sizes are reported in Supplementary Table 34. **c**, Density plots for gene expression conservation index values from each indicated comparison. Target genes are categorized as either human-biased distal cCRE targets or mammal level 3 distal cCRE targets. $P$ values were calculated using two-sided unpaired Wilcoxon rank-sum tests. **d**, Box plots of ABC putative target genes for each distal cCRE from the indicated conservation group. $P$ values were calculated using two-sided unpaired Wilcoxon rank-sum tests. Sample sizes are reported in Supplementary Table 34. **e**, The conservation levels of human TAD boundaries, including human-specific,

level 0 (sequence conserved), level 1 (tissue conserved) and level 2 (cell type conserved), across mammals. **f**, The conservation levels of genes and cCREs in the indicated conservation groups. **g**, Heat maps for pairs of human-biased cCREs targeting human-biased genes in the same cell type. The values represent the smallest $-\log_{10}$-transformed false-discovery rate (FDR) for any comparison between human and another species. Rows are scaled to visualize relative differences across cell types. **h**, WashU browser snapshots of *FOXP2* (left) and *RYR3* (right) showing chromatin accessibility, H3K27ac and RNA signals in human and chromatin accessibility and RNA for macaque, marmoset and mouse for MGC (left) or ASC (right) and L6 CT. The tracks display concordance of genome alignment from human (hg38) to the indicated species. For the box plots in **b**, **d** and **f**, the centre line represents the median, the box limits encompass the 25th to 75th percentiles and the whiskers represent 1.5× the interquartile range.

Fig. 9d,e). As expected, mappability-normalized values increased the number of human-specific distal cCREs acting as putative enhancers identified by the ABC model, increasing from 16.8% to 26.8% (Fig. 5a and Extended Data Fig. 8b). However, distal human-specific cCREs remained significantly depleted of putative enhancers compared with

all distal cCREs (31.61%, $P = 2.4 \times 10^{-11}$), while the percentage of distal human-biased putative enhancers (37.0%) was significantly higher than all level 0 cCREs ($P = 1.9 \times 10^{-8}$) (Fig. 5a). We observed a correlation between ABC scores and epi-conservation (Extended Data Fig. 8c,d). In general, cCREs located in TEs were less often predicted to function

as enhancers (Extended Data Fig. 8e). Notably, 76.9% of mammal level 3 distal cCREs were identified as putative enhancers (Fig. 5a), again suggesting that epi-conserved cCREs are more likely to function as enhancers.

Acetylation of histone H3 at lysine 27 (H3K27ac) is a known marker for active enhancers[48]. To examine H3K27ac levels of the putative enhancers identified using the ABC model, we performed droplet paired-tag[49] analysis in human M1 tissue to generate H3K27ac profiles in each cell type (Extended Data Fig. 10a). To annotate the cell types in the droplet paired-tag data, we integrated RNA profiles with 10x multiome RNA data for both human M1 and previously published mouse frontal cortex droplet paired-tag[49] data (Extended Data Fig. 10a). We observed a strong agreement between the signals of H3K27ac and chromatin accessibility across cell types (Extended Data Fig. 10b). Epi-conservation of H3K27ac correlated with epi-conservation of chromatin accessibility and was strongest at promoter-distal cCREs (Extended Data Fig. 10c,d and Supplementary Table 32). However, this correlation disappeared at the mammal level 3 chromatin-accessible cCREs (Extended Data Fig. 10c,d). For distal cCREs, H3K27ac was higher at those predicted as putative enhancers (Extended Data Fig. 10e). The levels of H3K27ac at distal cCREs for each conservation group had exact correspondence to the relative number of putative enhancers (Fig. 5a,b), further supporting the prediction for epi-conserved cCREs to possess enhancer activity.

Expression of mammal level 3 distal cCRE target genes was more conserved than human-biased distal cCRE target genes (Fig. 5c), providing evidence for conservation of the enhancer landscape to promote conservation of gene expression during evolution. Although human-biased cCREs were less often predicted as enhancers compared with epi-conserved cCREs (Fig. 5a), their putative target genes were enriched for divergent expression (Fig. 5d), giving evidence for their function as enhancers for human-biased gene expression.

We classified ABC-predicted human enhancer–gene pairs into conservation levels and identified 25,472 mammal level 1 pairs and 15,142 mammal level 2 pairs (Fig. 5e, Extended Data Fig. 8f,g and Supplementary Table 31). In general, mammal level 2 pairs had more conserved putative enhancers and target genes compared with those in human-biased pairs (Fig. 5f). Consistent with our gene expression conservation analysis, we found that human-biased cCREs target genes that are enriched for extracellular matrix organization, mammal level 3 cCREs target genes that are involved in transcriptional regulation and primate level 3 cCREs target genes that are enriched for nervous system and neuronal functions (Extended Data Fig. 8h–j).

To characterize human divergent gene regulatory programs, we identified all human-biased cCREs predicted to target a human-biased expressed gene in the same cell type (Supplementary Table 33). Most of these human divergent enhancer–gene pairs were found in four glia cell types, MGCs, OPCs, ODCs and ASCs (Fig. 5g). The genes in MGC divergent enhancer–gene pairs were significantly enriched for GO terms such as negative regulation of viral entry into the host cell and amyloid fibril formation (Extended Data Fig. 8k). Consistent with a previous report[50], we noticed *DSCAM* and *FOXP2* in the MGC human divergent enhancer–gene pairs—genes that have been implicated in neurodevelopmental disorders[51,52] (Fig. 5g,h). Also present was *RIN3*, which was previously implicated in Alzheimer's disease[53] (Fig. 5g and Extended Data Fig. 8l). Genes involved in synaptic signalling were found in other glia cell types such as *NOS1* in OPCs, *SYNDIG1* in ODCs and *RYR3* in ASCs (Fig. 5g,h). Although *FOXP2* was expressed in neurons across all the species, its expression in MGCs was restricted to human and is probably activated by the putative enhancer E1[50] (Fig. 5h (left)). This putative enhancer was linked to *FOXP2* by ABC and displayed MGC-specific and human-biased accessibility and H3K27ac (Fig. 5h (left)). Similarly, ryanodine receptor-3 (*RYR3*) was expressed in L6 CT (corticothalamic) neurons in all species, but had human-biased expression in ASCs. *RYR3* was targeted by three ASC human-biased putative enhancers with ASC-specific and human-biased accessibility

and H3K27ac (Fig. 5h (right)). By integrating comparative analyses of the epigenome and the transcriptome with 3D genome data, we identified putative human divergent gene regulatory programs across human cortical cells.

## Predicting cCRE use from DNA sequence

Previous studies have suggested a conserved regulatory grammar and syntax at CREs in the genomes of mammalian species[54], but how the genome encodes the gene regulatory program remains unclear. To understand how differences in species chromatin accessibility are driven by sequence changes, we trained a neural network model to predict the chromatin accessibility in each cell type from the DNA sequence alone (Fig. 6a). We adapted the neural network framework basenji[55] to this task. We first constructed testing and validation sets on chromosomes with conserved sequence identity from all four species (Fig. 6b). This mitigates the risk of data leakage[56,57] caused by an orthologous region trained on another species appearing in the test set. In such a case, one might greatly overestimate the model's understanding and predictive ability when applied to unseen DNA.

For each species, we evaluated how different training data contribute to the model accuracy. For each species, we trained a single modality baseline, a bimodal (CG DNA methylation and chromatin accessibility) and a four-species bimodal model, using the same parameters for each (Methods). We evaluated the accuracy of the model by comparing the Spearman correlation of all test set predictions in each cell type to the true values. Both non-human primates showed a significant increase in accuracy when trained using both chromatin accessibility and mCG datasets, and all species demonstrated a significant increase in accuracy when expanding to a multi-species multimodality dataset (one-sided paired *t*-test) (Fig. 6c).

We also evaluated each model's ability to predict differences among cell types. For each peak in the test dataset, we correlated the model predictions to the true accessibility across cell types (Methods). Including additional species demonstrated improved the prediction ability except for mice (Fig. 6d).

Given the superior predictive ability of the four-species model, we generated tracks for each species in the unseen test dataset. Our model effectively predicts chromatin accessibility at *SLC4A4*, successfully identifying not only L2/3 IT neuron activity in all cell types, but a human-specific increase in ODC chromatin accessibility (Fig. 6e). Moreover, at the huntingtin protein locus, the model again identifies L2/3 IT neuron activity, along with promoter activity conserved across cell types (Extended Data Fig. 11h).

To evaluate the ability of the model to predict epigenomes in unseen species, we trained four bimodal models, one excluding each species (for example, to evaluate unseen prediction accuracy in mouse, we trained a model using human, macaque and marmoset data). We then evaluated the model using the test dataset of the unseen species. To demonstrate the model performance under evolutionary dissimilarity, we performed predictions using the least accurate species-specific predictor for each model. For each held-out species, accuracy across regions within a cell type remained high (Extended Data Fig. 11f); however, when evaluating model accuracy in peak regions across cell types there is a considerable drop (Extended Data Fig. 11g), suggesting the necessity of species-specific information to inform patterns of cell type specificity.

We further evaluated how variable levels of genomic conservation impacted model accuracy across human test-set peaks. Notably, the single-species and the four-species models demonstrated increasing accuracy patterns in increasingly conserved regions (Extended Data Fig. 11a,b,d). Notably, both models showed high accuracy in predicting human-specific sequence cCREs relative to peaks with sequence orthology in all other species (Extended Data Fig. 11a,d). By contrast, sequences with human-biased activity were difficult to predict for

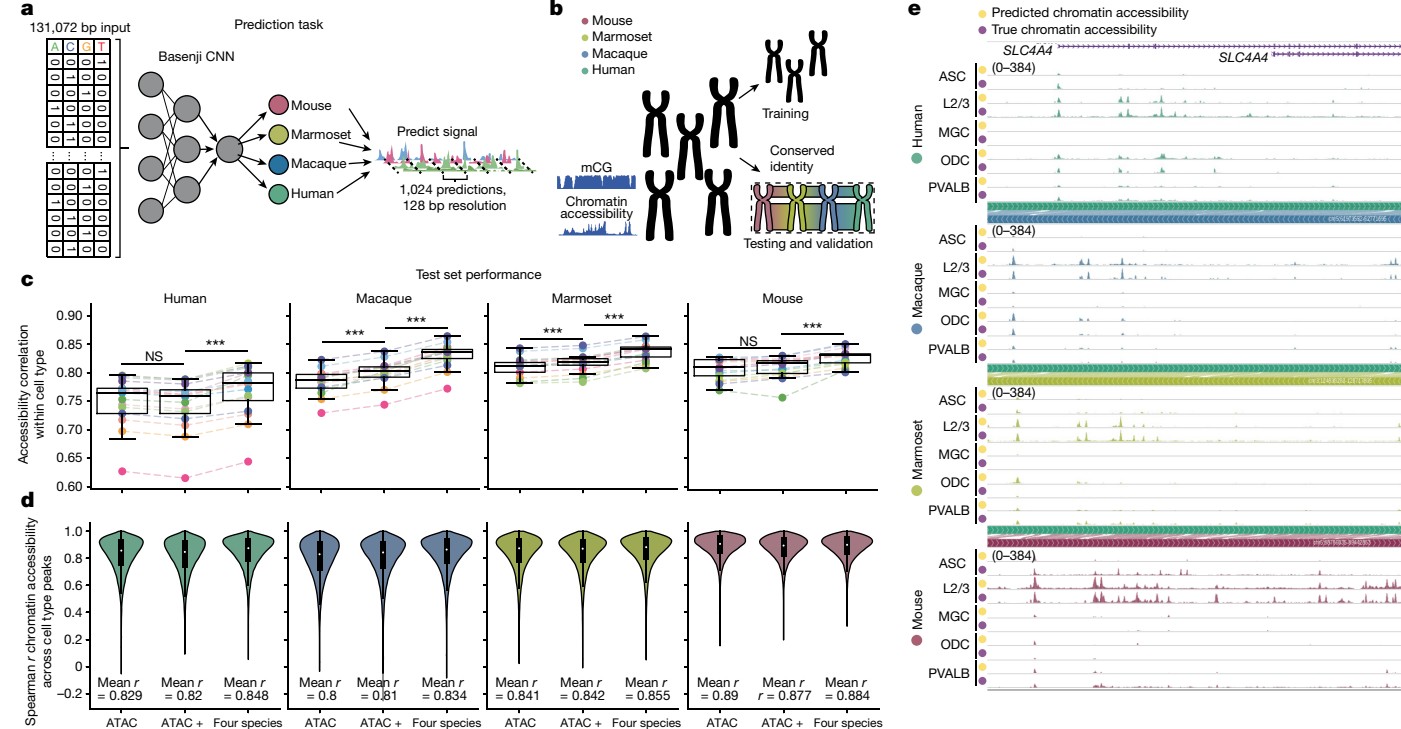

**Fig. 6 | Deep learning models predict cell-type-specific chromatin accessibility from the DNA sequence alone. a**, Schematic of the prediction task used to predict cCREs from the DNA sequence. CNN, convolutional neural network. **b**, Schematic of the dataset design for the model. Chromatin accessibility and DNA methylation datasets from human, macaque, mouse and marmoset are divided by chromosome. Chromosomes with a conserved sequence identity across species are identified as a testing and validation dataset. **c**, The prediction accuracy of the model chromatin accessibility prediction within each cell type in unseen test regions. The panels from left to right correspond to accuracy in human, macaque, marmoset and mouse datasets. For each species, three predictions were evaluated—a chromatin accessibility model; chromatin accessibility and DNA methylation; and a four-species model combining both modalities. The dots represent cell types. Statistical analysis was performed using one-sided paired-sample $t$-tests; ***$P < 1 \times 10^{-3}$. $n = 16$ cell types. **d**, Correlation across cell types in regions with a peak. Correlation was evaluated for each model type for each species as described in **c**. $n = 39,236, 44,311, 32,484$ and $41,605$ test set peaks for human, macaque, marmoset and mouse, respectively. **e**, True and predicted chromatin accessibility near *SLC4A4* in ASCs, Layer 2/3 IT neurons, microglia, ODCs and parvalbumin interneurons in human, macaque, marmoset and mouse. For the box plots in **c** and **d**, the centre line (**c**) and white dots (**d**) represent the median, the box limits encompass the 25th to 75th percentiles and the whiskers represent 1.5× the interquartile range.

both models (Extended Data Fig. 11a,c,d). Evaluating poorly predicted regions, we identified that both models have greater failure rates at intergenic peaks (Extended Data Fig. 11e).

## Epigenome conservation helps to interpret non-coding risk variants

Genome-wide association studies (GWASs) have identified common genetic variants linked to various traits and diseases, yet most GWAS variants reside in non-coding regions of the genome and their influence on gene expression remains unresolved[58]. A growing body of evidence suggests that the non-coding disease risk variants may contribute to disease by disrupting CREs and affecting gene expression in cell types relevant to disease pathogenesis[59-61]. As human cCREs with elevated epigenetic conservation levels are more likely to be predicted as active enhancers, we hypothesize that evidence of epigenetic conservation may improve our ability to interpret non-coding disease-risk-associated variants. To test this, we performed stratified linkage disequilibrium score regression (LDSC)[62]. Performing this analysis with all cCREs, we observed high enrichment for variants implicated in neurological traits within cCREs identified from various neuronal and glial cell types, as expected (Fig. 7a). However, when this analysis was performed with the human epi-divergent cCREs (specific and human-biased cCREs), the enrichment for GWAS variants associated with neurological traits

was almost entirely eliminated (Fig. 7a). By contrast, the enrichment improved when the analysis was performed with the mammal level 2 epi-conserved cCREs (Fig. 7a,b). For example, variants associated with multiple sclerosis (MS) are highly enriched for epi-conserved MGC cCREs, but not significantly enriched in any cell type when considering the full set of cCREs (Fig. 7a,c). Two other examples include anorexia nervosa and tobacco-use disorder, which show significant enrichments only in neuronal epi-conserved cCREs (Fig. 7a,c). Moreover, cCREs containing fine-mapped risk variants for Alzheimer's disease, bipolar disorder and schizophrenia have significantly higher epi-conservation compared with all cCREs (Extended Data Fig. 12a).

As our list of epi-conserved cCREs specifically linked MGC regulatory elements to MS, we used our enhancer–gene predictions to determine whether we could interpret potential gene regulatory effects of MS-risk-associated variants and their relation to microglial functions. Using a list of 233 MS-risk-associated variants[63], we identified 38 overlapping human cCREs with 32 predicted target genes. The target genes of enhancers containing MS-risk-associated variants are enriched for functions related to immune response pathways (Fig. 7d). For example, MS-risk-associated intronic variant rs60600003 resides in a mammal level 2 cCRE in *ELMO1*, a gene that is involved in phagocytosis (Fig. 7e). *ELMO1* is expressed in both MGCs and ODCs; however, the cCRE containing the risk variant is accessible exclusively in MGCs with matching H3K27ac signals (Fig. 7e and Extended Data Fig. 12b). This MGC-specific

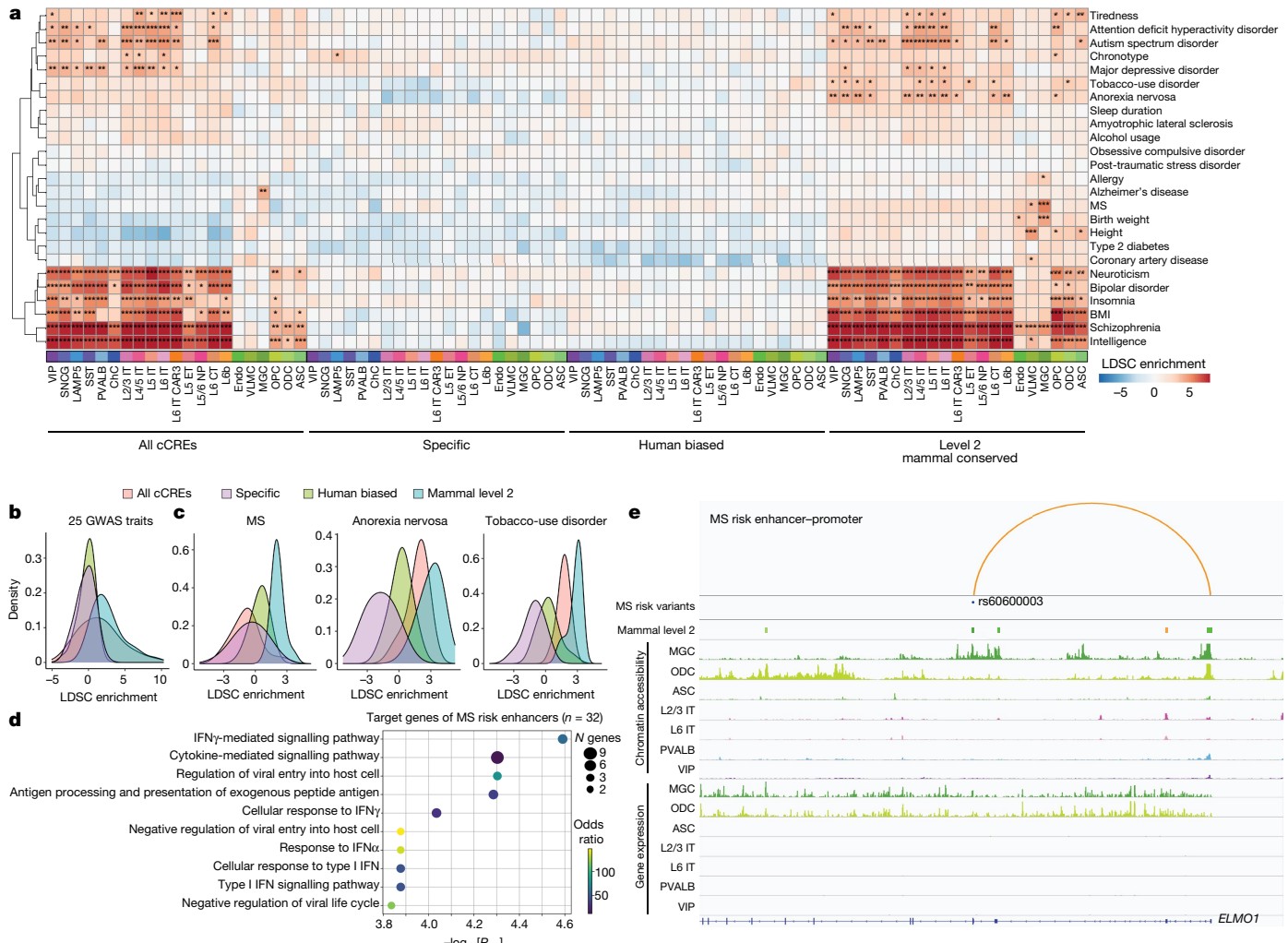

**Fig. 7 | Taking advantage of epigenetic conservation to interpret non-coding risk-associated variants of neurological disease and traits. a**, Linkage disequilibrium score regression analysis to identify GWAS enrichments in cCREs of each cell type for different conservation sets. *FDR-adjusted $P < 0.001$, **FDR-adjusted $P < 0.0001$, ***FDR-adjusted $P < 0.00001$. **b**, The distribution of LDSC enrichments across cells for each of the 25 traits from **a. c**, The distribution of LDSC enrichments across cells for MS, anorexia nervosa and tobacco-use disorder. **d**, The top significant GO biological process terms for ABC target genes of enhancers containing a MS-associated risk variant[63]. **e**, Example locus of a mammal level 2 predicted enhancer of *ELMO1* overlapping a MS-associated risk variant in a microglia-specific chromatin-accessible region.

cCRE is predicted to be an enhancer of *ELMO1*, indicating that *ELMO1* expression is selectively affected in MGCs by this MS-risk-associated variant. Our analysis provides examples of how comparative epigenomics can help with interpreting disease-risk genetic variants for neurological diseases such as MS.

## Discussion

Our comparative analysis of the transcriptome, epigenome and 3D genome features of 21 cortical cell types from four species provides a perspective on the evolution of gene regulatory programs in rodents and primates. By integrating four molecular modalities (gene expression, chromatin accessibility, DNA methylation and chromatin conformation) across 21 cell types in four species, we characterized conserved and divergent gene regulatory features focusing on three evolutionary times scales: mammal conserved (from human to mouse, around 90 million years), primate conserved (from human to marmoset, around 43 million years) and human divergent (human from macaque, about 25 million years ago). Although cCREs show non-neutral sequence constraint, the majority exhibit an unconserved epigenetic state. Their

selective constraint is dependent on the type of CRE. For example, epigenetic conservation of distal cCREs is generally lower than that of promoters or promoter-proximal cCREs, and epigenetic conservation of cCREs with a high cell type specificity is lower than those with broad cell type activities, consistent with previous findings[4–7].

By quantifying epi-conservation of all TF motifs, we found potential factors contributing to the evolution of the mammalian epigenome (Figs. 2h and 3d). We found evidence of divergence of TF expression and TF motif sequences in promoting species-specific epigenome landscapes (Fig. 2i,j). Compared with epi-divergent distal cCREs, we show that epi-conserved distal cCREs are more often predicted to act as enhancers of target genes (Fig. 5a), have stronger H3K27ac signals (Fig. 5b) and are more enriched for genetic variants associated with neurological disease/traits (Fig. 7a,b). Moreover, our data provide evidence that selective pressure on 3D genome organization maintains conserved gene regulatory programs (Fig. 4e,h). Taken together, this provides evidence that comparative epigenomics can assist in identifying functional enhancers.

We provide evidence that TEs may be a major source of species-specific chromatin-accessible cCREs (Fig. 2g), DMRs (Fig. 3c) and

TAD boundaries (Fig. 4d). Notably, different types of TEs contribute to the establishment of different categories of cCREs. For example, ERVK contributes highly to human-specific chromatin accessibility (Extended Data Fig. 4a,b) but much less to human-specific boundaries (Extended Data Fig. 6g). TE contribution is also cell type dependent. The human-specific cCREs in TEs from IT excitatory neurons occur most often in LINE-1 elements, whereas, in glia, they occur most often in ERV1 and ERVK (Extended Data Fig. 4a,b).

We highlight the power of machine learning approaches to learn the gene regulatory grammar from single-cell multiomic datasets across mammalian species and cell types (Fig. 6 and Extended Data Fig. 11). The resulting sequence-based predictors demonstrate a highly predictive ability in unseen species, suggesting a general conservation of regulatory grammar across mammalian species. These results also suggest differences in regulatory grammar that establishes patterns across cell types (Extended Data Fig. 11f,g). While neural networks have shown promise in predicting epigenetic features and gene expression levels from the DNA sequence[64–66], there is still a gap between model predictions and experiment-level observations. Despite recent advances, research in neural network scaling suggests improvements in model accuracy follow a power law, requiring an exponential increase in both model and dataset size[67]. Expanding epigenome datasets to diverse species enables us to overcome limited sequence diversity and dataset size as well as link sequence changes to diverse phenotypes[68].

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

# Methods

## Nucleus preparation from frozen brain tissue for Chromium single-cell multiome ATAC and gene expression analysis

M1 tissue was obtained from three human donors (male, aged 42, 29 and 58 years), three macaque donors (male, aged 6 (*Macaca mulatta*), 6 (*M. mulatta*) and 14 (*Macaca fascicularis*) years), three marmoset (*Callithrix jacchus*) donors (aged 5 (male), 4 (male) and 6 (female) years) and mouse primary motor cortex (MOp) from eight P56 C57BL/6J male mice (*Mus musculus*). Mouse MOp was dissected into four subregions (2C, 3C, 4B, 5D) as described previously[19]. Each subregion was pooled from four mice for each replicate, and a total of two replicates was performed for each subregion. C57Bl/6J mice, purchased from Jackson Laboratories, were kept for up to 10 days in the Salk animal barrier facility under a 12 h–12 h dark–light cycle, under a controlled temperature (between 20–22 °C) and food ad libitum. All samples were effectively controls, therefore randomization was not used and all samples were included in the same experimental group. Samples were labelled with IDs with no identifying donor information within species, however researches were not blind to the species for each sample. Sample size was not predetermined.

Brain tissue was pulverized using a mortar and pestle on dry ice and pre-chilled with liquid nitrogen. Pulverized brain tissue was resuspended in 1 ml of chilled NIM-DP-L buffer (0.25 M sucrose, 25 mM KCl, 5 mM MgCl₂, 10 mM Tris-HCl pH 7.5, 1 mM DTT, 1× protease inhibitor (Pierce), 1 U µl⁻¹ recombinant RNase inhibitor (Promega, PAN2515) and 0.1% Triton X-100). Tissue was Dounce homogenized with a loose pestle (5–10 strokes) followed by a tight pestle (15–25 strokes) or until the solution was uniform. Nuclei were filtered using a 30 µm CellTrics filter (Sysmex, 04-0042-2316) into a LoBind tube (Eppendorf, 22431021) and pelleted (at 1,000 rcf, for 10 min at 4 °C; Eppendorf, 5920R). The pellet was resuspended in 1 ml NIM-DP buffer (0.25 M sucrose, 25 mM KCl, 5 mM MgCl₂, 10 mM Tris-HCl pH 7.5, 1 mM DTT, 1× protease inhibitor, 1 U µl⁻¹ recombinant RNase inhibitor) and pelleted (1000 rcf, 10 min at 4 °C). Pelleted nuclei were resuspended in 400 µl 2 µM 7-AAD (Invitrogen, A1310) in sort buffer (1 mM EDTA, 1 U µl⁻¹ recombinant RNase inhibitor, 1× protease inhibitor, 1% fatty acid-free BSA in PBS). A total of 120,000 nuclei was sorted (Sony, SH800S) into a LoBind tube containing collection buffer (5 U µl⁻¹ recombinant RNase inhibitor, 1× protease inhibitor, 5% fatty acid-free BSA in PBS). Then, 5× permeabilization buffer (50 mM Tris-HCl pH 7.4, 50 mM NaCl, 15 mM MgCl₂, 0.05% Tween-20, 0.05% IGEPAL, 0.005% Digitonin, 5% fatty acid-free BSA in PBS, 5 mM DTT, 1 U µl⁻¹ recombinant RNase inhibitor, 5× protease inhibitor) was added for a final concentration of 1×. Nuclei were incubated on ice for 1 min, then centrifuged (500 rcf, 5 min at 4 °C). The supernatant was discarded and 650 µl of wash buffer (10 mM Tris-HCl pH 7.4, 10 mM NaCl, 3 mM MgCl₂, 0.1%.Tween-20, 1% fatty acid-free BSA in PBS, 1 mM DTT, 1 U µl⁻¹ recombinant RNase inhibitor, 1× protease inhibitor) was added without disturbing the pellet followed by centrifuging (500 rcf, 5 min at 4 °C). The supernatant was removed, and the pellet was resuspended in 7 µl of 1× nucleus buffer (nucleus buffer (10x Genomics), 1 mM DTT, 1 U µl⁻¹ recombinant RNase inhibitor). Nuclei (1 µl) were diluted in 1× nucleus buffer, stained with Trypan Blue (Invitrogen, T10282) and counted. In total, 16,000–20,000 nuclei were used for the tagmentation reaction and controller loading and libraries were generated according to the manufacturer's recommended protocol (https://www.10xgenomics.com/support/single-cell-multiome-atac-plus-gene-expression). 10x multiome ATAC–seq and RNA-sequencing (RNA-seq) libraries were paired-end sequenced on the NextSeq 500 and NovaSeq 6000 systems to a depth of around 50,000 reads per cell for each modality.

## Genome assemblies and annotations

*Homo sapiens* (human) assembly: hg38, GRCh38 annotation: hg38 Gencode v33; *M. musculus* (mouse) assembly: mm10, GRCm38 annotation: mm10 Gencode vM22; *M. mulatta* (rhesus monkey) assembly: Mmul_10 (rheMac10), annotation: Ensembl release 104 (and Refseq GCF_003339765.1 for 10x multiome (see below)); *Callithrix jacchus* (white-tufted-ear marmoset) assembly: cj1700_1.1 (calJac4), annotation: GCA_009663435.2.

To maximize the number of orthologous protein-coding genes quantified in macaque 10x multiome RNA data, we supplemented any missing protein-coding genes in GCF_003339765.1 gtf with annotations present in Ensembl release 104.

## 10x multiome sequence data processing and clustering

Raw sequencing data were processed using cellranger-arc (10x Genomics), generating single-nucleus RNA-seq (snRNA-seq) UMI count matrices for intronic and exonic reads mapping in the sense direction of a gene. We performed unsupervised clustering with RNA UMI counts using the Seurat (v.4)[71] standard analysis pipeline. First, cells were filtered for low-quality nuclei by requiring ≥1,000 ATAC fragments and ≥500 genes detected per nuclei. Counts were normalized using SCTransform identifying 3,000 variable genes used for principal component analysis (PCA). Putative multiplets were predicted using DoubletFinder[72] and 10% of cells were removed from each sample that had the highest doublet score. Batch correction across donors was performed using Harmony[73] on SCTransformed PCs. A $k$-nearest neighbour graph was built using the first 20 PCs and clusters were identified using Louvain clustering. To visualize clusters, we performed the UMAP non-linear dimension reduction technique[70]. We annotated subclass-level cell types for mouse, marmoset and human cells by reference mapping to published M1 snRNA-seq datasets[17,18] using Seurat. We integrated datasets from all four species using reciprocal PCA, which projects each species datasets into the PCA space of other species and identifies anchors by the same mutual neighbourhood requirement. For integration anchors, we considered only genes that are orthologous across all four species. Reads from 21 annotated cell types were combined to generate pseudo-bulk datasets used for downstream analyses.

## ATAC–seq peak calling and filtering

We used MACS2 for ATAC–seq peak calling on pseudo bulk ATAC–seq fragments using the MACS2 command callpeak with the parameters --shift −75 --ext 150 --bdg -q 0.1 -B --SPMR --call-summits -f BAMPE. We extended the peak summit by 249 bp upstream and 250 bp downstream to achieve 500 bp width for every peak. As the number of peaks called in each cell type is related to the sequence depth, which is highly variable due to differences in cell type abundance, we converted MACS2 peak scores (−log₁₀[$q$]) to score per million[74]. Peaks with a score per million of ≥2 were retained for each cell type. We further filtered human and mouse peaks by removing those with ENCODE blacklist regions (https://mitra.stanford.edu/kundaje/akundaje/release/blacklists/) of hg38 and mm10. For comparative analysis of human ATAC–seq peaks, we first removed peaks that were mapping to a region in any of the four species that had low read mappability. To identify regions with low mappability in our ATAC–seq data, we counted all reads in 1 kb bins across each genome. We took 1 kb bins with 0 reads and, for the remaining bins, we took the 0.02 quantile for the number of reads mapped and extended by 1 kb in both directions giving us 3 kb low-mappability bins. Finally, low-mappability bins within 5 kb were stitched together, providing our final list of low ATAC–seq mappability regions. Peaks or orthologous elements falling within any of these regions in any species were excluded from the comparative analysis.

## Nucleus isolation and FANS

For all snm3C-seq samples, in situ 3C treatment was performed during the nucleus preparation, enabling the capture of the chromatin conformation modality as described previously[22]. These steps were performed using the Arima-3C BETA Kit (Arima Genomics). The nuclei were isolated and sorted into 384-well plates using previous described methods[21]. In brief, single nuclei were stained with Alexa-Fluor488-conjugated anti-NeuN antibodies (MAB377X, Millipore) and

Hoechst 33342 (62249, Thermo Fisher Scientific) and then processed for fluorescence-activated nucleus sorting (FANS) using the BD Influx sorter with single-cell (1 drop single) mode.

## Library preparation and Illumina sequencing

The snm3C-seq samples were prepared according to a previously described library preparation protocol[21,22]. This protocol has been automated using the Beckman Biomek i7 instrument to facilitate large-scale applications. The snm3C-seq libraries were sequenced on the Illumina NovaSeq 6000 instrument, using one S4 flow cell per 16 384-well plates and using 150 bp paired-end mode.

## Data preprocessing

**Mapping and quality control of snm3C-seq data.** The snm3C-seq mapping was conducted using the YAP pipeline (cemba-data, v.1.6.8) as previously described[21]. Specifically, the main mapping steps include (1) demultiplexing FASTQ files into single cells (cutadapt, v.2.10); (2) read-level quality control; (3) mapping (one-pass mapping for snmC, two-pass mapping for snm3C) (bismark v.0.20, bowtie2 v.2.3); (4) BAM file processing and quality control (samtools v.1.9, picard v.3.0.0); (5) methylome profile generation (ALLCools v.1.0.8); and (6) chromatin contact calling. All reads from human, macaque, marmoset and mouse were mapped to the hg38, Mmul_10, calJac4 and mm10 genomes, respectively.

Pre-analysis quality control for DNA methylome cells was (1) overall mCCC level < 0.05; (2) overall mCH level < 0.2; (3) overall mCG level < 0.5; (4) total final reads of >500,000 and <10,000,000; and (5) Bismarck mapping rate > 0.5. Note that the mCCC level serves as an estimation of the upper bound of the cell-level bisulfite non-conversion rate. Moreover, we calculated lambda DNA spike-in methylation levels to estimate the non-conversion rate for each sample. To prevent any meaningful cell or cluster loss, we chose loose cut-offs for the pre-analysis filtering. The potential doublets and low-quality cells were accessed in the clustering-based quality control described below. For the 3C modality in snm3C-seq cells, we also required cis-long-range contacts (two anchors > 2,500 bp apart) > 50,000.

## Methylome clustering analysis

After mapping, single-cell DNA methylome profiles of the snm3C-seq datasets are stored in the 'all cytosine' (ALLC) format, which is a tab-separated table compressed and indexed by bgzip/tabix. The generate-dataset command in the ALLCools package can help to generate a methylome cell-by-feature tensor dataset (MCDS), stored in Zarr format. We used non-overlapping chromosome 100 kb (chrom100k) bins of the corresponding reference genome to perform clustering analysis, gene body regions ±2 kb for clustering annotation and integration with the companion 10x multiome dataset. Details of the integration analysis are described in the next section.

## Methylome clustering

We next performed clustering on the chrom100k matrices, as described previously[21]. In summary, the clustering process includes the following main steps:
(1) Basic feature filtering based on coverage and ENCODE blacklist.
(2) Highly variable feature (HVF) selection.
(3) Generation of posterior chrom100k mCH and mCG fraction matrices, as used in the previous study and initially introduced previously[75].
(4) Clustering with HVF and calculating cluster enriched features (CEF) of the HVF clusters. This framework is adapted from cytograph2[76]. We first perform clustering based on variable features and then use these clusters to select CEFs with stronger marker gene signatures of potential clusters. The concept of CEF was introduced previously[77]. The calculation and permutation-based statistical tests for calling CEFs are implemented in ALLCools.clustering.

cluster_enriched_features, in which we select for hypo-methylated genes (corresponding to highly-expressed genes) in methylome clustering.
(5) Calculation of PCs in the selected cell-by-CEF matrices and generation of the t-SNE and UMAP[70] embedding for visualization. t-SNE was performed using the openTSNE[78] package according to previously described procedures[79].

## Cluster-level DNA methylome analysis

After the clustering analysis, we merged the single-cell ALLC files into pseudo-bulk level using the ALLCools merge-allc command. We next performed DMR calling as previously described[80] using methylpy. In brief, we first calculated CpG differentially methylated sites using a permutation-based root mean square test[80]. The base calls of each pair of CpG sites were added before analysis. We then merged the differentially methylated sites into DMR if they are (1) within 500 bp and (2) the minimum methylation difference was greater than or equal to 0.3 across samples. We applied the DMR calling framework across the cell clusters in each species.

## Cell- and cluster-level 3D genome analysis

**Generating the chromatin contact matrix and imputation.** After snm3C-seq mapping, we used the cis long-range contacts (contact anchors distance > 2,500 bp) and trans contacts to generate single-cell raw chromatin contact matrices at three genome resolutions: chromosome 100 kb resolution for the chromatin compartment analysis; 25 kb bin resolution for the chromatin domain boundary analysis; and 10 kb resolution for the chromatin loop or dot analysis. The raw cell-level contact matrices are stored in HDF5-based scool format. We then used the scHiCluster package (v.1.3.2) to perform contact matrix imputation. In brief, the scHiCluster imputes the sparse single-cell matrix in two steps: the first step is Gaussian convolution (pad = 1); the second step is to apply a random walk with restart algorithm on the convoluted matrix. The imputation is performed on each cis matrix (intrachromosomal matrix) of each cell. For 100 kb matrices, the whole chromosome is imputed; for 25 kb matrices, we imputed contacts within 10.05 Mb; for 10 kb matrices, we imputed contacts within 5.05 Mb. The imputed matrices for each cell were stored in cool format. For most of the following analyses, cell matrices were aggregated into cell groups identified in the previous section. These pseudo-bulk matrices are concatenated into a tensor called CoolDS, and stored in the Zarr format.

## Domain boundary analysis

We used the imputed cell-level contact matrices at 10 kb resolution to identify the domain boundaries within each cell using the TopDom algorithm[81]. We first filter out the boundaries that overlap with ENCODE blacklist v2.

We used cooltools (v.0.5.1) to call cluster-level boundaries and domains with 10 kb resolution matrices. A sliding window of 500 kb was used to compute the insulation score of each bin, and the bins with boundary strength > 0.1 were selected as domain boundaries.

## Loop analysis

We called the cluster-level loops with 10 kb resolution matrices using the call_loop function in the scHiCluster package.

## Droplet paired-tag

Nuclei were extracted from frozen human M1 tissue using Dounce homogenization according to the method described in the 'Nucleus preparation from frozen brain tissue for Chromium single-cell multiome ATAC and gene expression analysis' section above. Single nuclei were subsequently stained with AlexaFluor488-conjugated anti-NeuN (MAB377X, 603 Millipore) antibodies and Hoechst 33342 (62249, Thermo Fisher Scientific). The stained nuclei were sorted to

split NeuN+ and NeuN– using the BD Influx sorter. Finally, the NeuN+ and NeuN– nuclei were combined at a 2:8 ratio.

The experimental protocol for droplet paired-tag was adopted from a previous study[49]. In brief, pA–Tn5 and H3K27ac (Abcam, ab4729) primary antibodies were pre-conjugated at room temperature during nucleus extraction at 1 µg per 500,000 nuclei, and subsequently incubated with 0.50 million permeabilized nuclei at 4 °C overnight. After the overnight incubation, the nuclei were washed twice to remove excess antibodies and PA-Tn5, and then tagmented by PA-Tn5 at 37 °C for 1 h on a ThermoMixer (Eppendorf).

The tagmentation reaction was terminated by adding stop buffer. We aliquoted 40,000 nuclei into two tubes for loading onto the Chromium Next GEM Chip J system and carried out droplet generation using the Chromium X microfluidic system (10x Genomics). Reverse transcription and cell barcoding were performed inside the 10x GEM system. Both DNA and RNA library construction were performed according to the Chromium Next GEM Single Cell Multiome ATAC + Gene Expression kit manual except that we used 13 amplification cycles for histone modality libraries.

**Identification of orthologous sequence elements across species (level 0).** We identified orthologous sequences for each human *cis*-regulatory region in all other species using liftOver[30]. For each human ATAC peak and DMR, we first performed liftOver to each other species' genome with a requirement of 50% retained sequence identity (minMatch = 0.5). For loop anchors and boundaries, lifted-over required only 30% of retained sequence identity (minMatch = 0.3) to account for the difficulty of lifting over a longer (10 kb) region. Any region that could not be lifted to any of the other profiled species was identified as human specific. For ATAC peaks (500 bp), we retained only orthologous elements that are 1 kb or less to the lifted-over genome. We next performed liftOver from the identified orthologous sequence back to the human sequence. We retained all sequences that mapped back to the same peak identity as 'level 0 conserved' between human and the respective species. We then further identified sequences that are level 0 across all mammals and level 0 across primates.

**Identification of human level 1 (tissue conserved) and level 2 (cell type conserved) CREs.** For each human feature (DMR, ATAC peak, loop, boundary and ABC enhancer pair), we determined whether the feature was also present across species. For each non-human species, we used each feature's orthologous coordinates in hg38 and performed bedtools[82] intersect[82], counting each human element with an overlapping element as having level 1 conservation between human and that species independent of cell type. We further identified elements that are level 1 across all mammals (mammal level 1), as well as elements that are level 1 across primates and not in mouse (primate level 1). Elements were identified as level 2 conserved if the intersection, as described above, existed in any of the same cell types across species. This procedure was modified for DMRs, with DMRs being split into hypomethylated and hypermethylated DMRs and performing the described procedure for each; the results of both were aggregated. For loops, this procedure was modified by requiring intersection at both anchor bins. For ABC enhancer pairs, we required that the orthologous cCRE targeted the orthologous gene across species.

**Identification of level 3 conserved peaks and DMRs.** For each species pair, we identified ATAC peaks and DMRs with conserved patterns of activity across cell types. We first normalized peak accessibility in each cluster to $\log_2[CPM]$ quantified for level 0 mammal peaks or the combined set of mammal and primate level 0 peaks (when comparing primates). For DMRs, we transformed quantifications to 1 – the mCG level in each cell type. We then considered the effect size (*T*-statistic) of a GLS regression[25] between the species as the effect size of conservation. This procedure controls for the effects of dependence between cell

types. A key step in GLS is to estimate the covariance matrix. For each species pair, we computed a covariance matrix between cell types by first taking the covariance between cell types for each species across all peaks or DMRs. We then formed a covariance matrix for the regression by taking the mean of both species' covariance. Given the GLS *T*-statistic for each species pair, we next identified conserved genes between each species with a false discovery rate of 0.05 adjusted using the Benjamini–Yekulti method[31] to account for dependency among cCREs.

We further identified two categories for peaks and DMRs: those conserved among mammals that were identified as conserved between each pair of species, and those conserved among primates that were identified as conserved among all three primates but not among all species.

**Cell type specificity of genes, ATAC peaks and DMRs.** For each gene, ATAC peak and DMR, we computed its cell type specificity using an information-theoretic criteria[83]. We identified ubiquitously expressed genes as those with a specificity of less than 0.01. For DMRs, we transformed quantifications to 1 – the methylation level in each cell type.

For distal ATAC peaks and distal DMRs we compared each increasing pair of conservation levels for changes in mean conservation using a two-sided *t*-test for independent samples.

**Annotation of TEs and TSS proximity.** For each human element in each category (DMR, peak, loop, boundary), we annotated its TE association and identified its TSS proximity using annotatePeaks.pl with hg38 from HOMER[84]. This analysis was repeated for mouse ATAC peaks using mm10 to identify their TE association.

We compared each increasing conservation level of ATAC peaks and DMRs to determine enrichment for TSS proximity. Between each pair of levels, we performed a two-sided Fisher's exact test.

**GO enrichment analysis.** We performed GO enrichment analysis using the Enrichr[85] module in GSEApy[86]. For each gene set, we used GO biological process 2021. We performed such analyses using the most appropriate background set, for example, the background set for level 3 genes was all genes expressed in each species using the default minimal expression criteria as edgeR (v.3.36.0)[26]. For ABC target genes, the background set was all human genes called as having an ABC enhancer. For evaluating human-biased genes in specific cell types, the background set was all genes tested for differential expression in the cell type. When evaluating human-biased gene links, the background set was all tested genes in that cell type that had an ABC link.

**Identification of species-biased gene activity.** Starting with a list of one-to-one orthologous genes across all four species, we performed differential expression analysis on pseudo bulk count profiles for each cell type using edgeR (v.3.36.0)[26]. We performed analysis using previous recommendations[87]. Each pseudo bulk profile was normalized for sequencing depth using trimmed mean of *M*-value normalization[88], after which tagwise dispersion was estimated using locfit. We fit a single model to predict the expression of a cell type based on species identity using glmFit, after which differential expression was evaluated on between-species contrasts for each species pair. We used stringent criteria to identify whether a gene is differentially expressed between a species pair. To account for multiple comparisons we nominated an FDR of 0.001, which we further lowered to $8.33 \times 10^{-6}$ by dividing by the number of pairs of species (6), multiplied by the number of cell types (20). In addition to this FDR threshold, we required our differentially expressed genes to meet a minimum fold change of 2, as well as be expressed in at least 15% of the cells in the upregulated species cell type.

After applying these criteria, we further identified biased genes for each species. For each cell type in each species, we identified biased genes as a gene that was significantly upregulated in that cell type compared with in each other species.

**Identification of peaks with species-biased chromatin accessibility.** Starting from the sets of human peaks with orthologues in all four species, we used edgeR to identify differential chromatin accessibility across species. We used the same parameters as used for identifying species-biased gene activity to estimate fold changes and $P$ values for each orthologous peak region. When identifying significantly differentially accessible peaks, we made some modifications. We used the same FDR cut-off ($8.33 \times 10^{-6}$); however, to account for the sparseness of peaks, we no longer placed a threshold on the number of cells where a peak was detected. To compensate, we require a minimum fold change between species of at least 4.

After applying these significance criteria, we further identified biased peaks for each cell type in each species. For each cell type in each species, we identified a biased peak as a peak that was significantly upregulated in that cell type compared with in each other species.

**Identification of genes with conserved patterns of activity.** For each species pair, we identified genes with conserved patterns of activity across cell types. We first normalized gene expression in each cluster to $\log_2[\text{CPM}]$ quantified in orthologous genes. We next considered the effect size ($T$-statistic) of a GLS regression[25] between the species as the effect size of conservation. This procedure controls for the effects of dependence between genes. A key step in GLS is to estimate the covariance matrix. For each species pair, we computed a covariance matrix between cell types by first taking the covariance between cell types for each species across all genes. We then formed a covariance matrix for the regression by taking the mean of both species' covariance. Given the GLS $T$-statistic for each species pair, we then identified conserved genes between each species with an FDR of 0.05, adjusted using the Benjamini–Yekulti method[31] to account for dependence among genes.

We further identified two categories for genes: those conserved among mammals that were identified as conserved between each pair of species, and those conserved among primates that were identified as conserved among all three primates but not among all species.

## TF motif scanning

For each TF motif from the JASPAR[89] CORE vertebrate database, we used FIMO (v.5.5.3)[90] to scan for all occurrences in the hg38 sequences of every cCRE and DMR. For all elements containing a given motif, we calculated the average conservation index, divergence index or PhastCons score at the TF motifs. To classify the TF class for each motif, we used the annotations in the TFClass database[91].

**Annotation of TF families.** We annotated TF families to visualize the conservation and divergence of TF motifs. Annotated TF families were identified from TF class[91]. The HTML text document summarizing TF families was downloaded (http://www.edgar-wingender.de/huTF_classification.html) and parsed to identify the family of each motif analysed.

## Comparison of conservation index to sequence conservation

We compared our measured conservation index (defined as the mean GLS $T$-statistic across all species pairs) to sequence conservation as defined by PhastCons[92] for genes and chromatin-accessible cCREs.

For each cCRE, we computed two sequence conservation values: one as the average PhastCons of nucleotides in the cCRE, and a second as the average PhastCons of the previously identified motif sequences in the cCRE.

For each gene, we measured sequence conservation as the average PhastCons across all the gene exons.

For both genes and cCREs, we compared the conservation index to PhastCons using two-sided Spearman correlation.

## Analysis of paired-tag RNA and H3K27ac

Droplet paired-tag fastq files were demultiplexed using cellranger-arc (v.2.0.0) using the command 'cellranger-arc mkfastq'; however, DNA and RNA data were preprocessed using cellranger-atac (v.2.0.0) and cellranger (v.6.1.2), respectively, and barcodes were manually paired using the related barcodes connecting each modality[49].

## Integration with human or mouse motor cortex 10x multiome RNA.

For human M1, nuclei with RNA from less than 500 detected genes were removed. Counts were log normalized, identifying 3,000 variable genes used for PCA. Putative multiplets were predicted using DoubletFinder[72] and 10% of cells were removed from each reaction ($n = 2$) that had the highest doublet score. Cells were clustered using Seurat, and clusters were annotated by reference mapping to the 10x multiome RNA generated in this study.

Published mouse frontal cortex data[49] was re-annotated by reference mapping to mouse MOp 10x multiome RNA data from this study using Seurat. Cells that were not found in the MOp were removed (D12MSN, OBGA, OBGL CLAGL and STRGA).

DNA fragments were combined from each annotated cluster to generate H3K27ac pseudobulk files. H3K27ac counts were quantified for each cCRE in human and mouse for the region ±2 kb from the centre. These counts were used for downstream analyses.

## Conservation and divergence of human cell type molecular identity

We first scored activity for each peak in each cell type. To do so, we first subset to distal peaks, as promoter elements demonstrate much greater sequence and epi conservation, and promoter peak proportion may be representative of reduced relative sequencing depth rather than increased promoter activity. We next further subset to only peaks with orthologous sequences across mammals. Next, we normalized peak activity in each cell type as the CPM among all peaks called in that cell type.

Given the peak activity of each cell type, we calculated that cell's weighted peak conservation as the conservation index of each peak in that cell type, multiplied by each peak's activity score.

We then calculated the sequence conservation of each cell type. First, we first identified the sequence conservation of each peak. Using the previously identified motif coordinates from FIMO v.5.5.3, we considered the sequence conservation score of a peak to be the mean PhastCons[92] of the motifs in each peak. For each cell type, we consider the weighted sequence conservation score to be the sequence conservation of each peak multiplied by the cell type activity of each peak.

To compute normalized TF divergence, we first calculated normalized TF expression for each cell type. We subsetted gene expression in each cell type to a list of transcript factors, and then normalized expression to CPM (among TFs). Given this relative TF activity for each cell type, we next calculated the TF divergence of a cell type by multiplying the relative expression of each TF by the average divergence index of the same TF between human and all other species.

We computed the weighted epigenome divergence of each cell type by multiplying the activity score of each of each cell type's peaks, and the absolute $\log_2$-transformed fold change of that cell type's peaks compared with all other species.

## Analysis of region mappability

UMAP mappability scores for $K = 100$ were downloaded for hg38 from a previous study[47]. Mappability scores were converted from wig to bigwig using UCSC wigToBigWig and averaged over peak regions using bigWigAvgOverBed[93]. To assess the impact of mappability on read counts, total counts for each peak and each 4 kb region centred on each peak were counted using bedtools multicov[82].

To normalize for mappability, the counts of each region were divided by the regions mappability score. The regions were then normalized to mappability-normalized CPM for downstream analysis.

## Identifying putative enhancer–gene pairs with the ABC model

We used the ABC model[46] to identify putative enhancer gene links in each species. In brief, the ABC model uses normalized contact frequencies from Hi-C data, along with a measure of enhancer activity, to predict putative enhancer–gene pairs. For each cell type, we ran the ABC model using the default parameters, providing normalized Hi-C matrixes at 10 kb resolution, ATAC chromatin accessibility BAM files and a list of ATAC peaks identified in that same cell type. Predictions with an ABC score greater or equal to 0.02 were considered positive and used for downstream analysis.

For each cCRE conservation level, we quantified the proportion of cCREs involved in an ABC-predicted enhancer–gene pair.

We performed ABC again using mappability-normalized counts to account for mappability differences impacting the ABC links identified among different enhancer classes. To do so, we computed the average mappability in each peak for each cell type, and replaced the values 'activity_base' columns in enhancer_list.txt with mappability-normalized CPM values.

We again quantified the proportion of cCREs involved in ABC-predicted enhancer–gene pairs and, while there was an increase in predicted ABC enhancers among the groups most affected by mappability, most ABC pair categories were unaffected.

Because UMAP mappability scores are unavailable for marmoset and macaque genomes, we proceeded with the ABC links identified using un-normalized values.

## Identification of conserved enhancer–gene pairs

Human ABC pairs were classified as mammal level 0 (sequence conserved) if the pair contained a cCRE that was orthologous across all four species and targeted a one-to-one orthologous gene in all four species. For those not mammal level 0, they were tested for primate level 0, which includes the same criteria but across only three primates. Pairs that were not mammal or primate level 0 were classified as human-specific. For all mammal level 0 pairs, we classified those as epi-conserved if the orthologous element in all four species was predicted to target the same gene. If the same ABC enhancer–gene pair was called across species, regardless of which cell type, it was categorized as mammal level 1 conserved. If the same ABC enhancer–gene pair was identified in at least one of the same cell types across species, it was categorized as level 2. We performed the same analysis across primates for primate level 0 pairs to classify as primate level 1 and primate level 2. For level 0 pairs that were not identified as mammal or primate level 1, they were categorized as human biased.

## Identification of human divergent enhancer–gene pairs

We identified CREs that are likely to regulate human-biased patterns of gene expression. For each cell type, a human divergent enhancer–gene pair was defined as a human-biased enhancer with an ABC link to a gene that was human-biased in the same cell type. In this case, we considered ABC links identified in any human cell type, as missed links may be reflective of lower chromatin contact coverage in a cell type rather than true cell type differences.

## Cross-species sequence-based model of epigenome activity

We trained a deep learning model to predict open chromatin, using a Basenji[55,66] neural network architecture. We used the same layer construction as in a previous study[66], with minor modifications. Namely, the standard convolutional tower was replaced using residual convolutional blocks, which have been shown to improve learning speed and accuracy[64,94]. For multi-species modelling, we added an output prediction head for each species.

**Dataset construction.** We selected test and validation datasets by identifying chromosomes with high degrees of sequence similarity across species to minimize data leakage[56]. Chromosomes were selected by visualizing region correspondence in the NIH National Library of Medicine Comparative Genome Viewer. We removed low-coverage cell types from training and evaluation, namely endothelial cells, Chandelier interneurons, L5 extra-telencephalic neurons and vascular leptomeningeal cells.

**Training.** For each species we trained three models, one model that predicts chromatin accessibility alone, one that predicts both chromatin accessibility and DNA methylation, and one model including all other species with the species held out for evaluation trained on both DNA methylation and ATAC data.

Models were trained on a single NVIDIA A6000 GPU with 48 GB of VRAM. Training datasets were augmented using reverse complements as well as a 3 bp sequence shift. Each model was trained for at least 10 epochs, with training continuing for as long as validation loss had improved within the past 8 epochs. Training parameters were as follows: batch size = 4, loss = poisson, ADAM optimizer, learning rate = 0.01, momentum = 0.99, clip_norm = 2. We saved the model with the lowest validation loss across training and used it for evaluation. In the case of multi-species models, we saved the lowest validation loss model for each species, and used that model for later predictions.

**Model evaluation.** The evaluate within-cell-type predictive accuracy, for each model, we evaluated the predictive ability to rank activities within cell types as the Spearman correlation between the predicted accessibility within a cell type, to the true accessibility of that cell type for all peaks.

To evaluate the improvement of different datasets on model accuracy, we performed a one-sided paired sample $t$-test between the chromatin accessibility alone and the bimodal model, as well as between the bimodal modal and the multi-species bimodal model.

**Cross-cell-type evaluation.** To evaluate the model's ability to predict patterns of chromatin accessibility across cell types, we subset to regions in the testing dataset overlapping all peaks called in each species genome. For each peak in the testing dataset, we calculated the correlation of the predicted cell type chromatin accessibility with the true chromatin accessibility at the same locus. We also measured the normalized error for each peak. The normalized error was computed as the L1 norm between the true and predicted accessibility divided by the mean true accessibility at that locus.

**Evaluation of model generalization to unseen species.** For each species, we evaluated our ability to predict its accessibility when excluded from training. After training a three-species model excluding each species, we predicted accessibility for that species in the testing dataset. We first evaluated the accuracy of these predictions by calculating the within-cell type correlation of predictions for each species-specific prediction made by the three-species model. We then report the accuracy of the predictions using the poorest predictor of the three training species. We further evaluated the predictive accuracy of the held-out model for predicting changes in chromatin accessibility across cell types as described previously.

## GWAS variant enrichment

We obtained GWAS summary statistics for quantitative traits related to neuropsychiatric and neurological traits and disorders as described previously[19]. We prepared summary statistics to the standard format for linkage disequilibrium score regression. We used a subset of chromatin-accessible peak cCREs from the indicated conservation group for each cell type as a binary annotation and, as the background control set, we used all cCREs from the indicated cell type. For each trait, we used cell-type-specific linkage disequilibrium score regression

(https://github.com/bulik/ldsc) to estimate the enrichment coefficient of each annotation jointly with the background control[62].

## External datasets

PhastCons[92] conserved elements were downloaded from the UCSC genome browser (http://hgdownload.cse.ucsc.edu/goldenpath/mm10/phastCons60way/). A list of annotated TF genes was downloaded from (https://ars.els-cdn.com/content/image/1-s2.0-S0092867418301065-mmc2.xlsx)[95].

Mouse frontal cortex H3K27ac droplet paired-tag data were downloaded from the Gene Expression Omnibus (GEO: GSE224560).

## Statistics

No statistical methods were used to predetermine sample sizes. There was no randomization of the samples, and investigators were not blinded to the specimens being investigated. Low-quality nuclei and potential barcode collisions were excluded from downstream analysis as outlined above.

## Ethical compliance

Permission was obtained from the decedent next of kin. Postmortem tissue collection was performed in accordance with the provisions of the United States Uniform Anatomical Gift Act of 2006 described in the California Health and Safety Code section 7150 (effective 1/1/2008) and other applicable state and federal laws and regulations. The Western Institutional Review Board reviewed tissue collection processes and determined that they did not constitute human subjects research requiring institutional review board (IRB) review. Mouse experiments were approved by the SALK Institute Animal Care and Use Committee under protocol number 18-00006. Marmoset experiments were approved by and performed in accordance with the Massachusetts Institute of Technology IACUC protocol number 05170520. Macaque experiment protocols were approved by the University of Washington Institutional Animal care and Use Committee.

## Reporting summary

Further information on research design is available in the Nature Portfolio Reporting Summary linked to this article.

## Data availability

Data produced in this study are available at the NCBI GEO under accession number GSE229169 (10x multiome), GSE240297 (sn-m3C-seq) and GSE246760 (droplet paired-tag). Data have been uploaded for viewing on the WashU Comparative Epigenome Browser data hub (https://epigenome.wustl.edu/BrainComparativeEpigenome/).

## Code availability

Code to perform analyses in this study is accessible at GitHub (https://github.com/ejarmand/comparative_epigenomic_motor_cortex).

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

**Acknowledgements** We thank all of the other members of the Ren and Ecker laboratory for their input. This study was supported by NIH grants U19MH11483 to J.R.E. and E.M.C., U19MH114831-04s1 to J.R.E. and B.R., 5U01MH121282 to J.R.E. and M.M.B., NIH grants P510D01425 and U420D011123 supporting the WaNPRC and UM1HG011585 to B.R. J.R.E. is an investigator of the Howard Hughes Medical Institute. Work at the Center for Epigenomics was also supported by the UC San Diego School of Medicine. This publication includes data generated at the UC San Diego IGM Genomics Center using an Illumina NovaSeq 6000 system that was purchased with funding from a National Institutes of Health SIG grant (S10 OD026929).

**Author contributions** Study supervision: B.R. and J.R.E. Contribution to data analysis: N.R.Z., E.J.A., W.W., J.Z., S.L., H.L., Y.E.L., W.T. and Y.X. Contribution to data generation: N.R.Z., J.R.N., R.G.C., A.B., M.M., L.C., K.D., H.S.I. and J.A.R. Contribution to data interpretation: N.R.Z., E.J.A., W.W., B.R. and J.R.E. Contribution to writing the manuscript: N.R.Z., E.J.A., B.R. and W.W. All of the authors edited and approved the manuscript.

**Competing interests** B.R. is a co-founder and consultant of Arima Genomics and co-founder of Epigenome Technologies. J.R.E. is on the scientific advisory board of Zymo Research.

**Additional information**
**Correspondence and requests for materials** should be addressed to Joseph R. Ecker or Bing Ren.

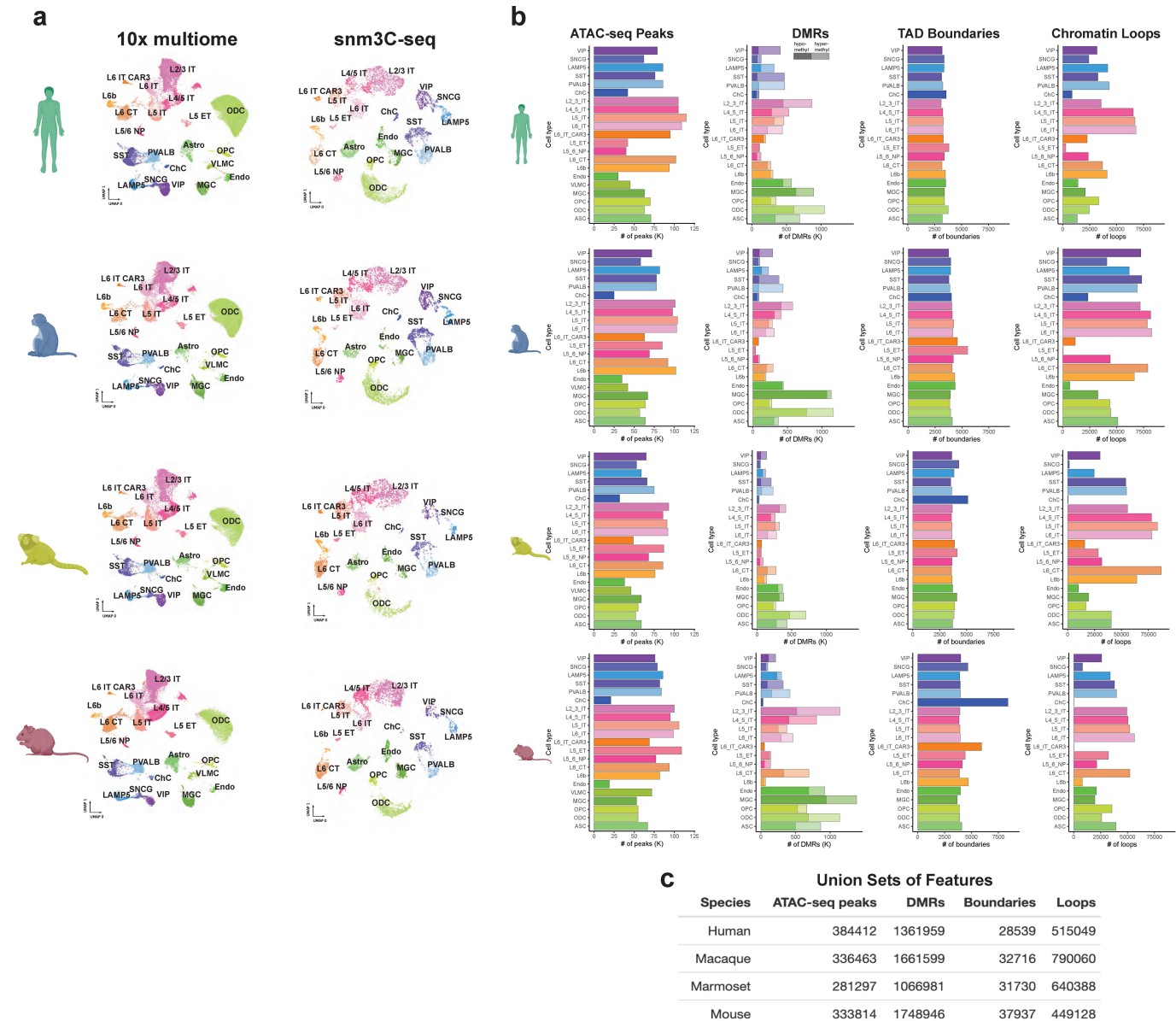

**Extended Data Fig. 1 | Cell type quantification in each species. a.** Uniform manifold approximation and projection (UMAP)[70] embeddings of 10x multiome RNA (left) and snm3C-seq DNA methylation (right) clusters for human, macaque, marmoset, and mouse separately. **b.** Number of indicated features (ATAC-seq peaks, DMRs, TAD boundaries, or chromatin loops) identified for each cell type for each species. **c.** Numbers of unique features found in each species, i.e. union set of features. Species silhouettes in **a** and **b** created in BioRender.

## c | Union Sets of Features

| Species | ATAC-seq peaks | DMRs | Boundaries | Loops |
|---|---|---|---|---|
| Human | 384412 | 1361959 | 28539 | 515049 |
| Macaque | 336463 | 1661599 | 32716 | 790060 |
| Marmoset | 281297 | 1066981 | 31730 | 640388 |
| Mouse | 333814 | 1748946 | 37937 | 449128 |

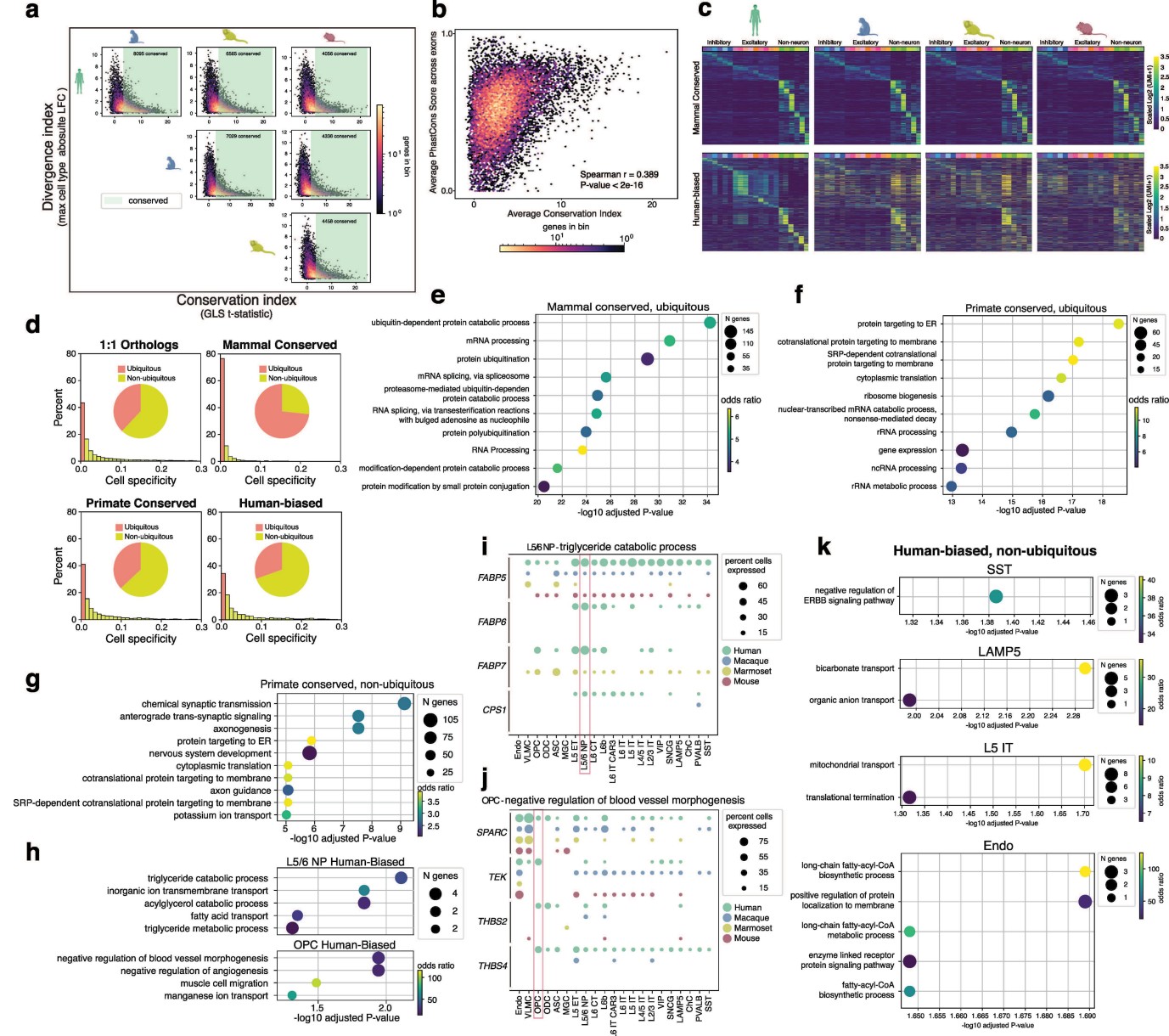

**Extended Data Fig. 2 | Patterns of gene expression conservation and divergence. a**. Pairwise divergence vs conservation index of gene expression for each species pair. **b**. A scatter plot highlighting the correspondence of gene expression conservation index to average PhastCons score across exonic sequences of each gene. **c**. Heatmaps in each species highlighting conserved (top) and human-biased (bottom) genes in each cell type. Genes are ordered by the highest expressed cell type in the human data. **d**. Histograms highlighting the distribution of entropy-based cell type specificity measures for each human for each indicated category. Pie charts summarizing the proportion of ubiquitously expressed (specificity ≤ 0.01) genes in each indicated category. **e**. Dot plot displaying the GO terms enriched in ubiquitously expressed mammal conserved genes. **f**. Dot plot displaying the top significant GO terms enriched in ubiquitously expressed primate conserved genes. **g**. Top significant GO

analysis terms for non-ubiquitous primate conserved genes. **h**. Top significant GO analysis terms for non-ubiquitous human biased genes in L5/6 NP neurons and oligodendrocyte precursor cells. **i**. Dot plot displaying human-biased L5/6 NP neuron genes involved in triglyceride catabolic processes. The size of each point represents the percent of cells with a transcript detect. Each point is coloured by species. **j**. Dot plot displaying human-biased OPC genes involved in the negative regulation of blood vessel morphogenesis. The size of each point represents the percent of cells with a transcript detected. Each point is coloured by species. **k**. Dot plots displaying the top significant GO terms identified in non-ubiquitously expressed human-biased genes, for each cell type where a significant enrichment was identified. Species silhouettes in **a** and **c** created in BioRender.

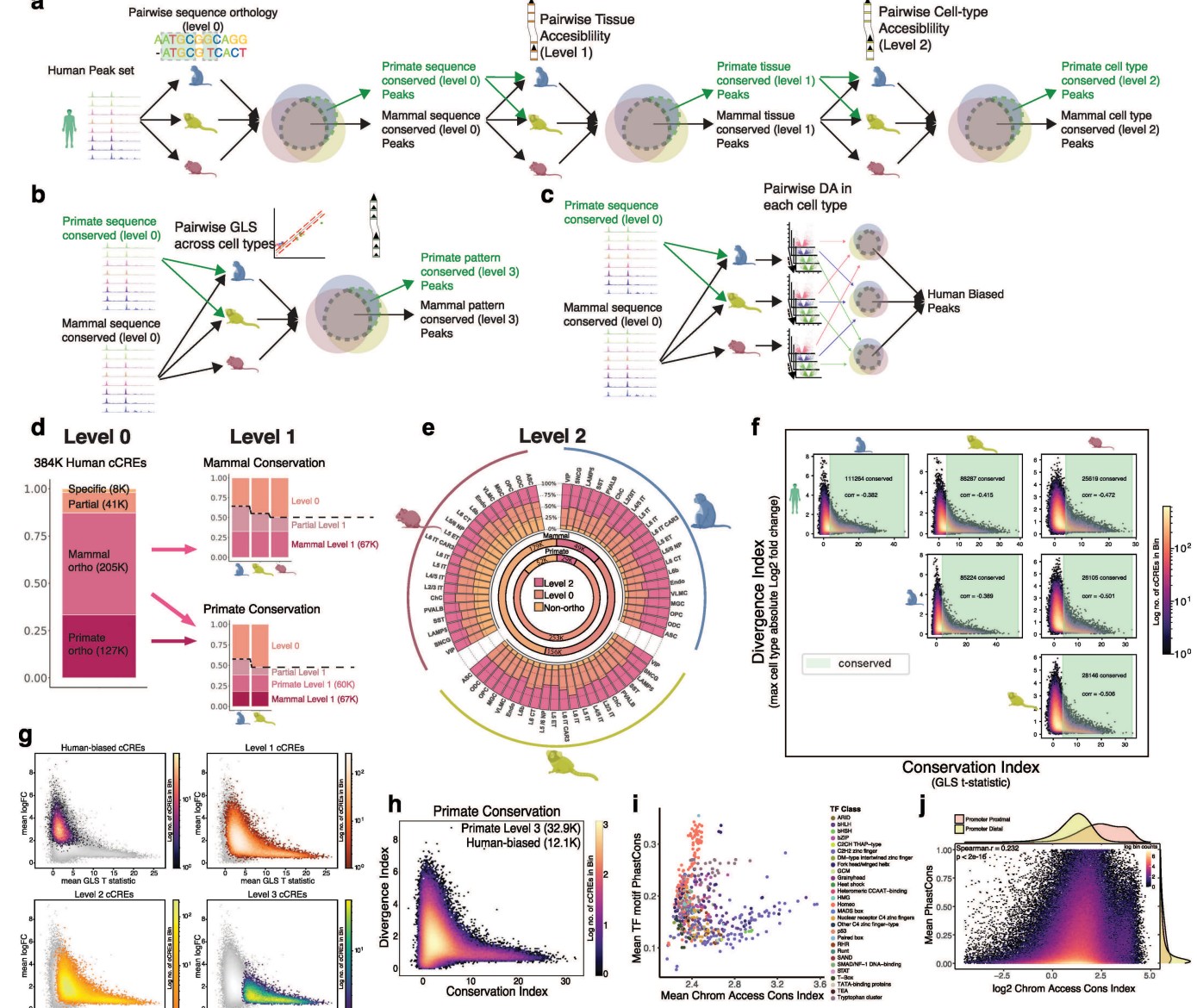

**Extended Data Fig. 3 | Comparative chromatin accessibility. a**. A workflow schematic for classifying level 0 (sequence conserved), level 1 (tissue conserved), and level 2 (cell type conserved), epigenome elements across both mammals and primates. **b**. A workflow schematic for classifying level 3 (matched patterns across all cell types) conserved elements in both mammals and primates. **c**. A schematic illustrating the workflow for identifying human-biased cCREs. **d**. Stacked bar plots representing the breakdown of human cCREs from ATAC-seq peaks for each indicated group for Level 0 and Level 1 conservation. **e**. Level 2 conservation of human cCREs from ATAC-seq peaks showing the overlap between each species for the same cell type (outer circle stacked bars). Inner circles show the breakdown for mammal and primate comparisons for all human ATAC-seq cCREs **f**. Scatter plots highlighting the pairwise divergence vs. conservation index of human ATAC-seq peaks for each species pair. **g**. Scatter plots comparing the conservation index and divergence index of all mammal level 0 peaks highlighting human biased (top left), level 1 (top right), level 2 (bottom left) or level 3 (bottom right). **h**. Scatter plot displaying the relationship between the conservation index (mean GLS T-statistic across comparisons) and divergence index (maximum absolute fold change across cell types) for primate level 0 cCREs. **i**. A scatter plot showing the relationship between conservation of epigenome signals (open chromatin conservation index), and conservation of motif sequence (PhastCons) averaged over all motifs of each transcription factor found in peaks. **j**. A scatterplot showing the relationship between sequence conservation (PhastCons) and ATAC conservation index among mammal level 0 cCREs. Density plots highlight the difference in ATAC conservation index (top) and PhastCons (right) between promoters and distal elements. Species silhouettes in **a**, **b**, **c**, **e** and **f** created in BioRender.

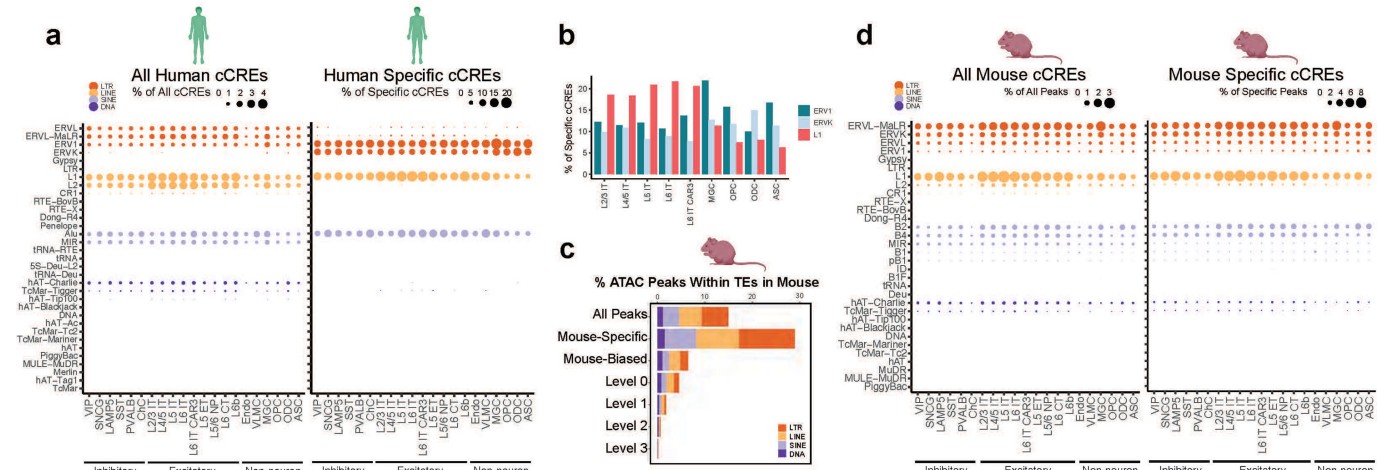

**Extended Data Fig. 4 | cCRE enrichment in TEs. a**. Dot plots showing the percentage of all (left) or human-specific (right) cCREs in different subclasses of TEs for each cell type. **b**. Bar plots showing the percentage of human-specific cCREs that overlap ERV1, ERVK, or LINE-1 for IT neurons and glia. **c**. Stacked bar plots showing percentage of mouse cCREs in TEs for different conservation groups. **d**. Dot plots showing the percentage of all (left) or mouse-specific (right) cCREs in different subclasses of TEs for each cell type. Species silhouettes in **a**, **c** and **d** created in BioRender.

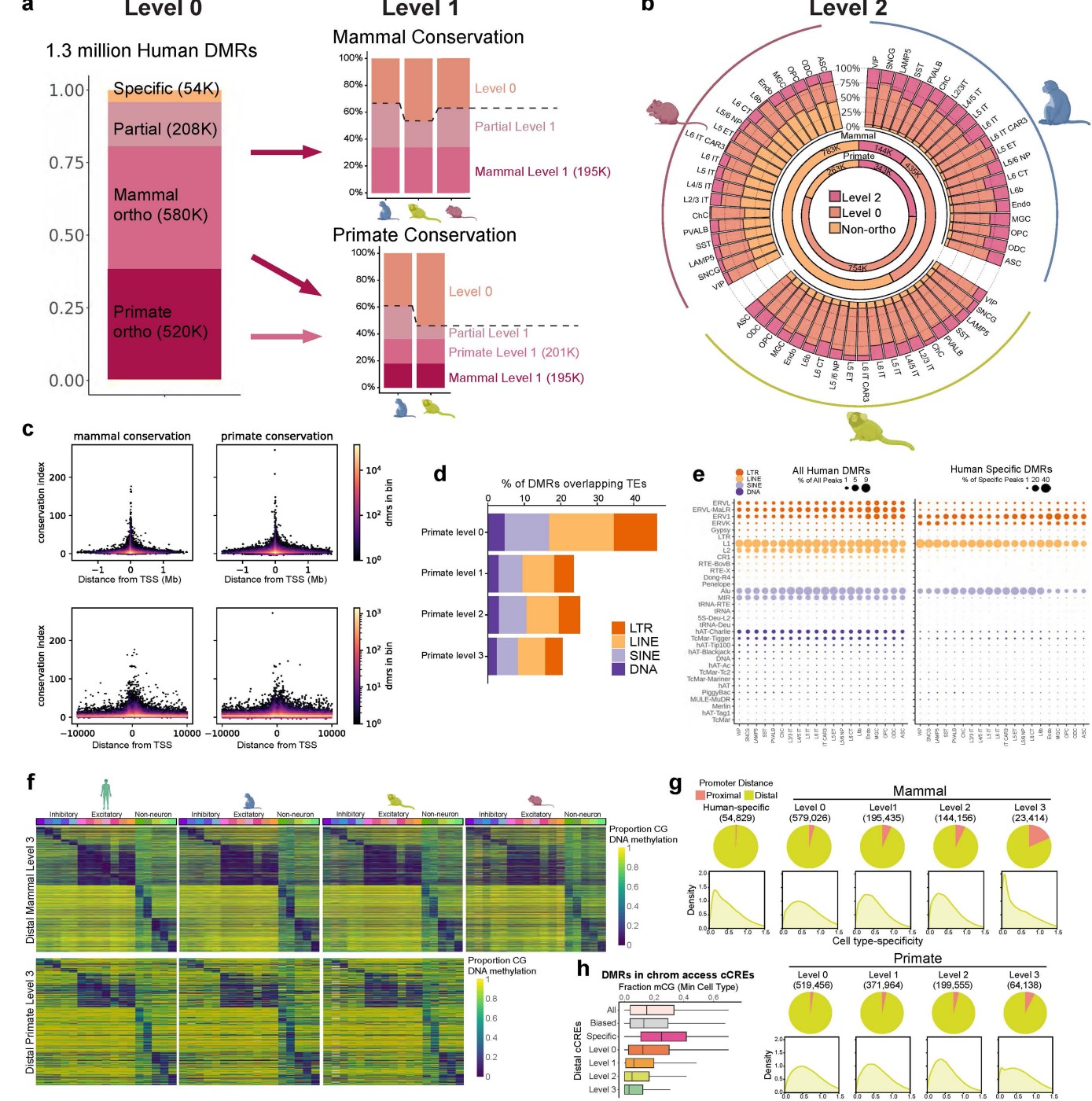

**Extended Data Fig. 5 | Patterns of DNA methylation conservation.**
**a**. Proportions of level 0 (sequence conserved) and level 1 (tissue conserved) DMRs across mammals and primates. **b**. Level 2 conservation of human DMRs showing the overlap between each species for the same cell type (outer circle stacked bars). Inner circles show the breakdown for mammal and primate comparisons for all human DMRs. **c**. Distance from nearest TSS with conservation level for all mammal conserved (left) or primate conserved (right) DMRs. Upper and lower plots display different genomic scales on the X-axis. **d**. Proportion of TEs in different levels of primate conserved DMRs. **e**. Dot plots showing the percentage of all (left) or human-specific (right) DMRs overlapping different subclasses of TEs for each cell type. N = 519,456, 371,964, 343,711, 64,138 DMRs in each category. **f**. Heatmaps in each species highlighting distal mammal level

three DMRs (above) and primate level three DMRs (below). Each DMR is ordered by the cell type with the lowest methylation level in the human data. **g**. Pie charts showing the proportion of promoter-proximal (≤1 kb from a TSS) and promoter distal (>1 kb from a TSS) elements for each level of mammal conservation (above) and primate conservation (below). Density plots show distribution of cell specificity scores (methods) for DMRs in each conservation group. **h**. Box plots showing the fraction of methylated CGs at all DMRs that intersect indicated group of promoter distal chromatin accessible cCREs. Box plots encompass 25th to 75th percentiles; central lines represent medians; whiskers represent 1.5 times the interquartile interval. Species silhouettes in **a**, **b** and **f** created in BioRender.

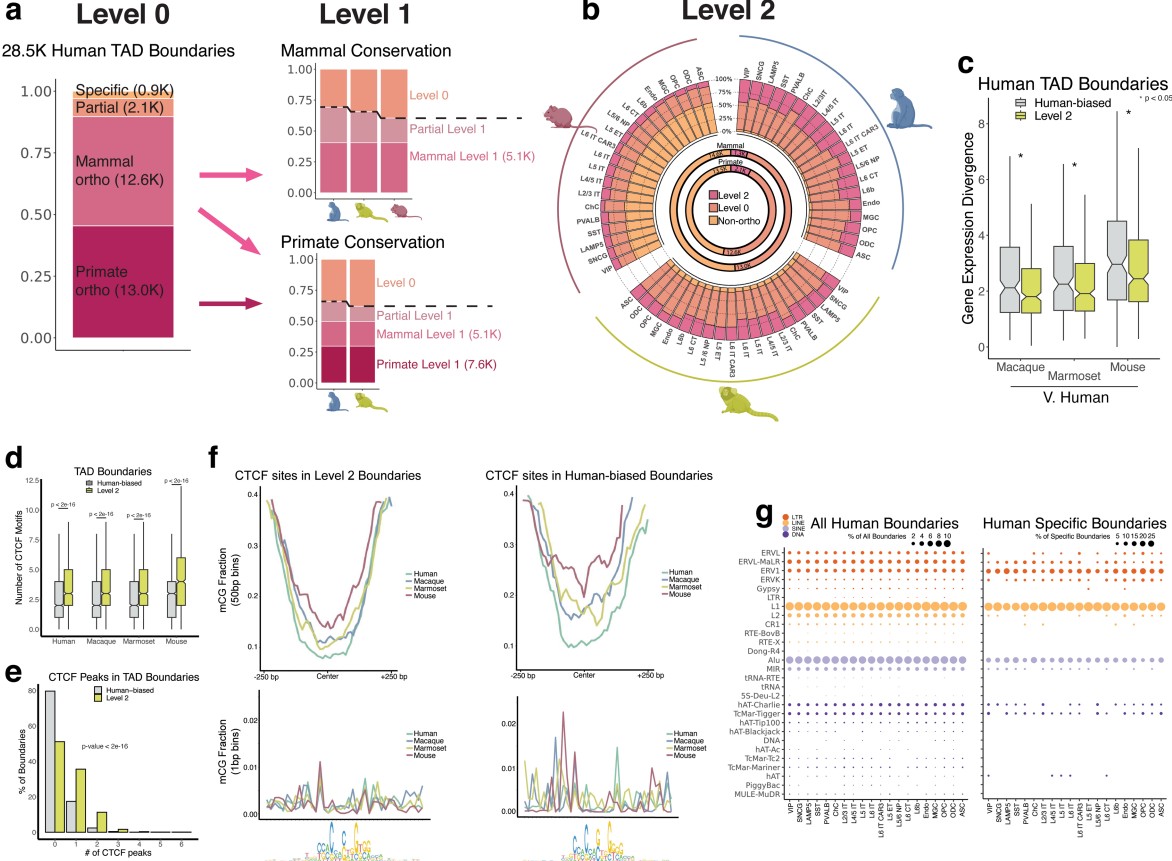

**Extended Data Fig. 6 | Comparative evaluation of boundary elements. a**. Stacked bar plots representing the breakdown of human boundaries for each indicated group for Level 0 and Level 1 conservation. **b**. Level 2 conservation of human boundaries showing the overlap between each species for the same cell type (outer circle stacked bars). Inner circles show the breakdown for mammal and primate comparisons for all human boundaries. **c**. Box plots comparing gene expression divergence index for genes overlapping human-biased (only humans have a TAD boundary) or level 2 (conserved in the same cell type across all species) TAD boundaries in human. Box plots encompass 25th to 75th percentiles; central lines represent medians; whiskers represent 1.5 times the interquartile interval. * p < 0.05 from two-sided, unpaired Wilcoxon rank sum. N = 198 (Human-biased), 291 (level 2). **d**. Box plots comparing CTCF motif number across species between human-biased and level 2 TAD boundaries.

P-values from two-sided, unpaired Wilcoxon rank sum and intervals same as in **c**. N = 1,653 (Human-biased), 1,290 (level 2). **e**. Paired histograms of number of CTCF peaks overlapping human-biased and level 2 human boundaries. P-values from two-sided, unpaired Wilcoxon rank sum. **f**. Average CG DNA methylation levels for cells containing a sequence orthologous human cCRE with a CTCF motif in level 2 or human-biased human boundaries. Signal is averaged for 50 bp bins in a 500 bp window centred around the CTCF motifs (top). And average per base CG DNA methylation levels within the CTCF motifs (bottom). Consensus motif sequence of all identified motifs below. **g**. dot plots showing the percentage of all (left) or human-specific (right) boundaries in different subclasses of TEs for each cell type. Species silhouettes in **a** and **b** created in BioRender.

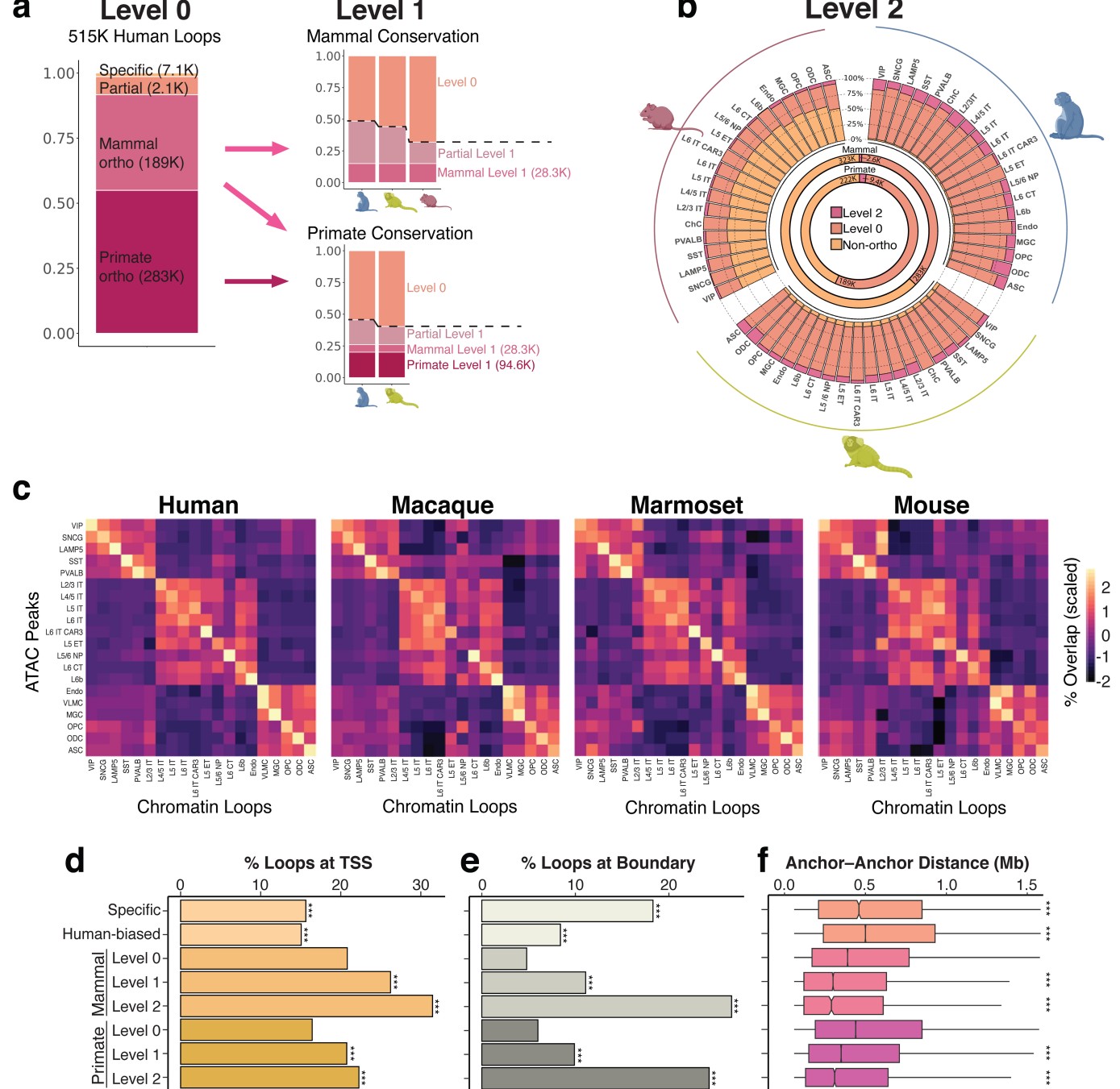

**Extended Data Fig. 7 | Chromatin Loop enrichment in TEs. a.** Stacked bar plots representing the breakdown of human boundaries for each indicated group for level 0 and level 1 conservation. **b.** Level 2 conservation of human boundaries showing the overlap between each species for the same cell type (outer circle stacked bars). Inner circles showing the breakdown for mammal and primate comparisons for all human boundaries. **c.** Heatmaps for each species showing the scaled percentage overlap of peaks and loops across cell types. **d.** Barplot showing the percent of loops with a TSS overlapping at least one anchor bin. *** p < 2e-16 from Fisher's exact test compared to mammal level 0, except primate level 1 and 2 were compared to primate level 0. **e.** Barplot showing the percent of loops with a boundary overlapping at least one anchor bin. P-values same as in **d**. **f.** Boxplots of anchor to anchor distance for loops of indicated conservation level. *** p < 2e-16 from two-sided, unpaired Wilcoxon rank sum. Box plots encompass 25th to 75th percentiles; central lines represent medians; whiskers represent 1.5 times the interquartile interval. Sample sizes reported in Supplementary Table 34. Species silhouettes in **a** and **b** created in BioRender.

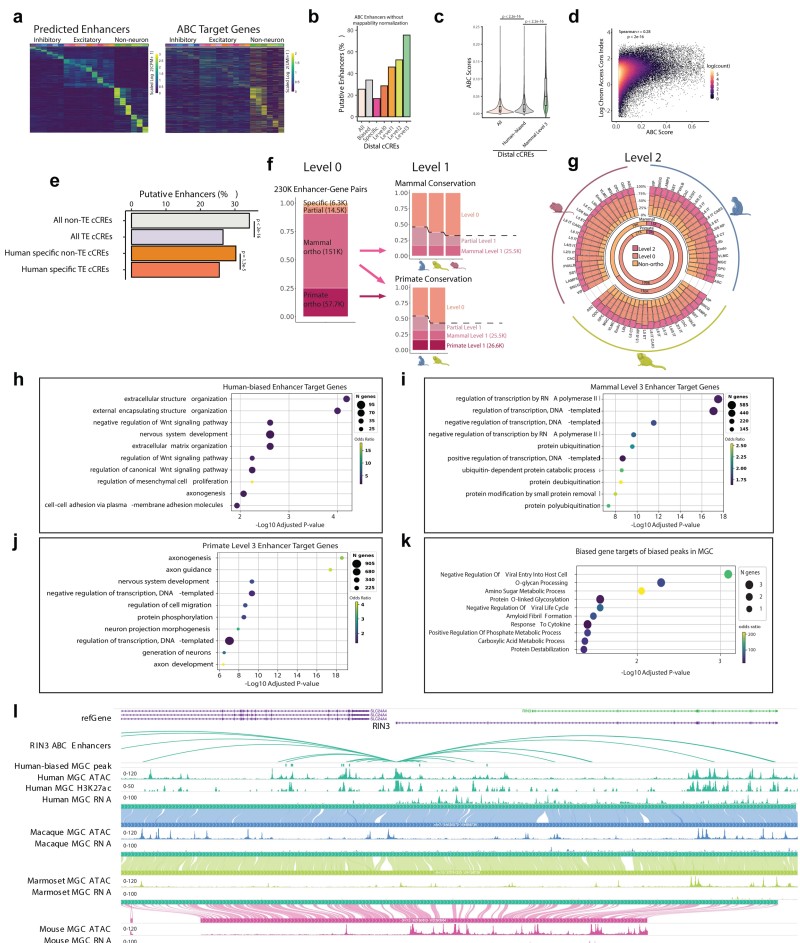

**Extended Data Fig. 8 | Properties of predicted ABC enhancers. a.** Heatmaps displaying correlation of activity between predicted enhancers with highest ABC score for each gene with prediction (n = 8,083), row scaled. **b.** Bar plots showing the proportion of cCREs predicted as a putative enhancer for each conservation level without normalizing for mappability. **c.** Violin plots showing the difference in ABC scores between all cCREs, human-biased cCREs and mammal level 3 cCREs. P-values from two-sided, unpaired Wilcoxon-rank sum test. N = 350,813 (all), 10,280 (Human-biased), 3,544 (mammal level 3). **d.** A scatter plot displaying the correlation between chromatin accessibility conservation index and ABC score. **e.** The percent of ABC predicted enhancers for indicated group. P-values from Chi-square test. **f.** Stacked bar plots representing the breakdown of human ABC predictions for each indicated group for level 0 and level 1 conservation. **g.** Level 2 conservation of human ABC predictions showing the overlap between each species for the same cell type (outer circle stacked bars). Inner circles show the breakdown for mammal and primate comparisons for all human boundaries. **h.** Dot plot showing top significant GO analysis Biological Process terms for human-biased enhancer ABC target genes. **i.** Dot plot showing top significant GO analysis Biological Process terms for mammal level 3 enhancer ABC target genes. **j.** Dot plot showing top significant GO analysis Biological Process terms for primate level 3 enhancer ABC target genes. **k.** Top significant GO analysis Biological Process terms for genes in a microglia human divergent enhancer-gene pair. **l.** WashU comparative epigenome browser snapshot highlighting the human-biased gene regulation of *RIN3* in microglia across species. Species silhouettes in **f** and **g** created in BioRender.

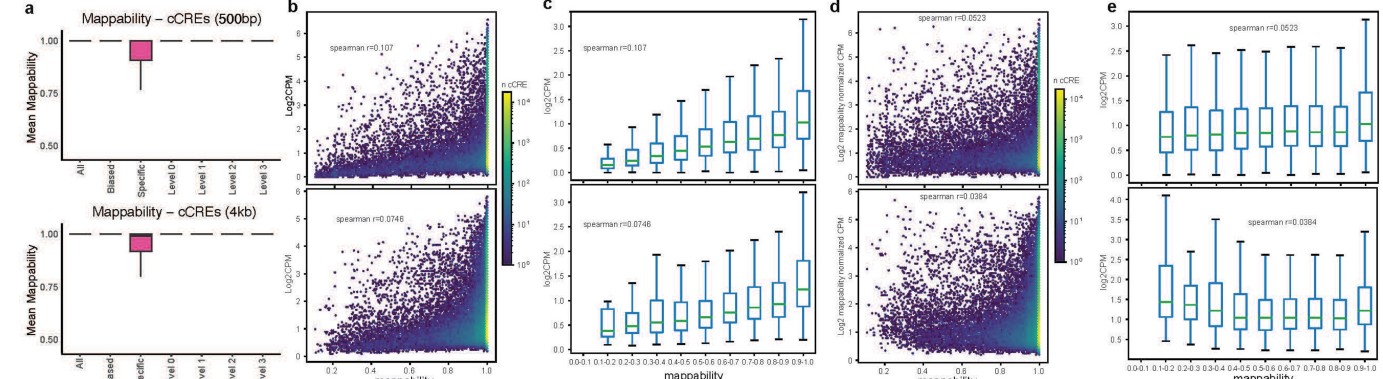

**Extended Data Fig. 9 | Mappability normalization of peak accessibility.**
**a**. 100 base-pair mappability rate[47] for each peak across conservation levels
(above), and the 4 kb centred around each peak (below). Box plots encompass
25th to 75th percentiles; central lines represent medians; whiskers represent
1.5 times the interquartile interval. **b**. A scatter plot highlighting the effect of
mappability on reads in each peak (N = 384,412), and the 4 kb region centred
around each peak. **c**. Box plots highlighting the change in number of reads as a
function of region mappability for each peak, and 4 kb region centred around
each peak. Box plots encompass 25th to 75th percentiles; central lines represent

medians; whiskers represent 1.5 times the interquartile interval. **d**. A scatter
plot highlighting the correspondence between mappability normalized
accessibility for each cell type peak and 4 kb region centred around each peak.
**e**. Box plots highlighting the change in number of mappability normalized
reads as a function of region mappability for each peak(N = 384,412), and 4 kb
region centred around each peak. Box plots encompass 25th to 75th percentiles;
central lines represent medians; whiskers represent 1.5 times the interquartile
interval.

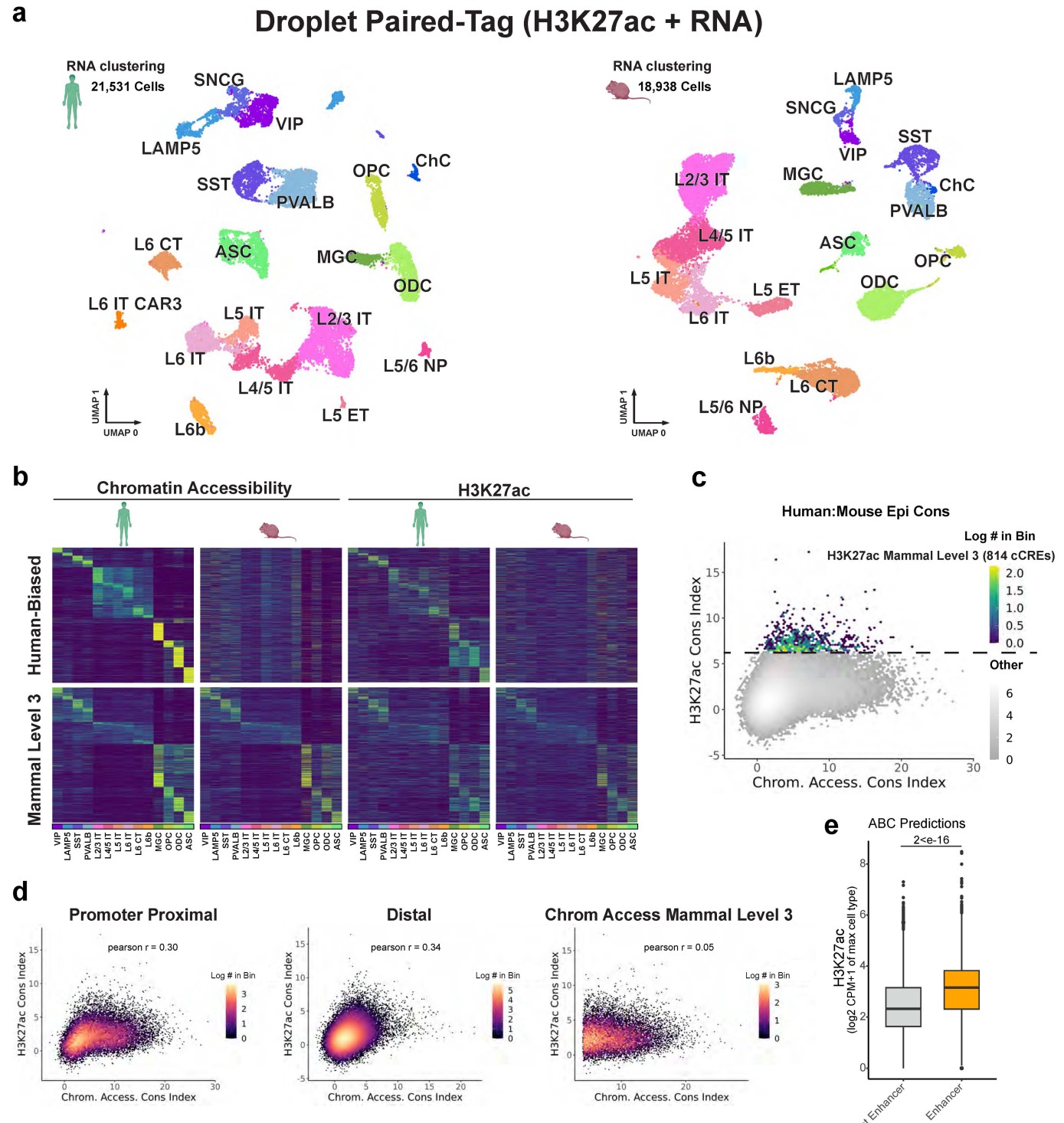

**Extended Data Fig. 10 | Conserved H3K27ac landscapes in human and mouse. a.** UMAP embedding of Droplet Paired-Tag RNA profiles coloured by cell type in human M1 and mouse frontal cortex[49]. **b.** Heatmaps of human-biased and mammal level 3 (conserved patterns across all cell types) cCREs in human and mouse ordered by the cell type with the highest accessibility in human. Cell types with low coverage for H3K27ac were removed. **c.** A scatter plot highlighting the relationship between chromatin accessibility conservation, and H3k27ac conservation for each cCRE. Level 3 conserved human-mouse conserved H3K27ac elements are highlighted (N = 814). **d.** Scatter plots highlighting the relationship between H3K27ac conservation, and chromatin accessibility

conservation at promoter-proximal elements (≤1 kb from a TSS, left), at promoter proximal-distal (>1 kb from a TSS, middle), and at chromatin accessibility mammal level 3 conserved cCREs (right). **e.** Box plots displaying H3K27ac signal (log2 CPM + 1, 4 kb genomic span) from the cell type with the highest signal for distal cCREs grouped by whether they are predicted to be enhancers or not by ABC model. H3K27ac counts were mappability normalized before converting to log2 CPM + 1. N = 281,840, 102,573; Box plots encompass 25th to 75th percentiles; central lines represent medians; whiskers represent 1.5 times the interquartile interval. Two-sided, unpaired Wilcoxon rank sum test for P-values. Species silhouettes in **a** and **b** created in BioRender.

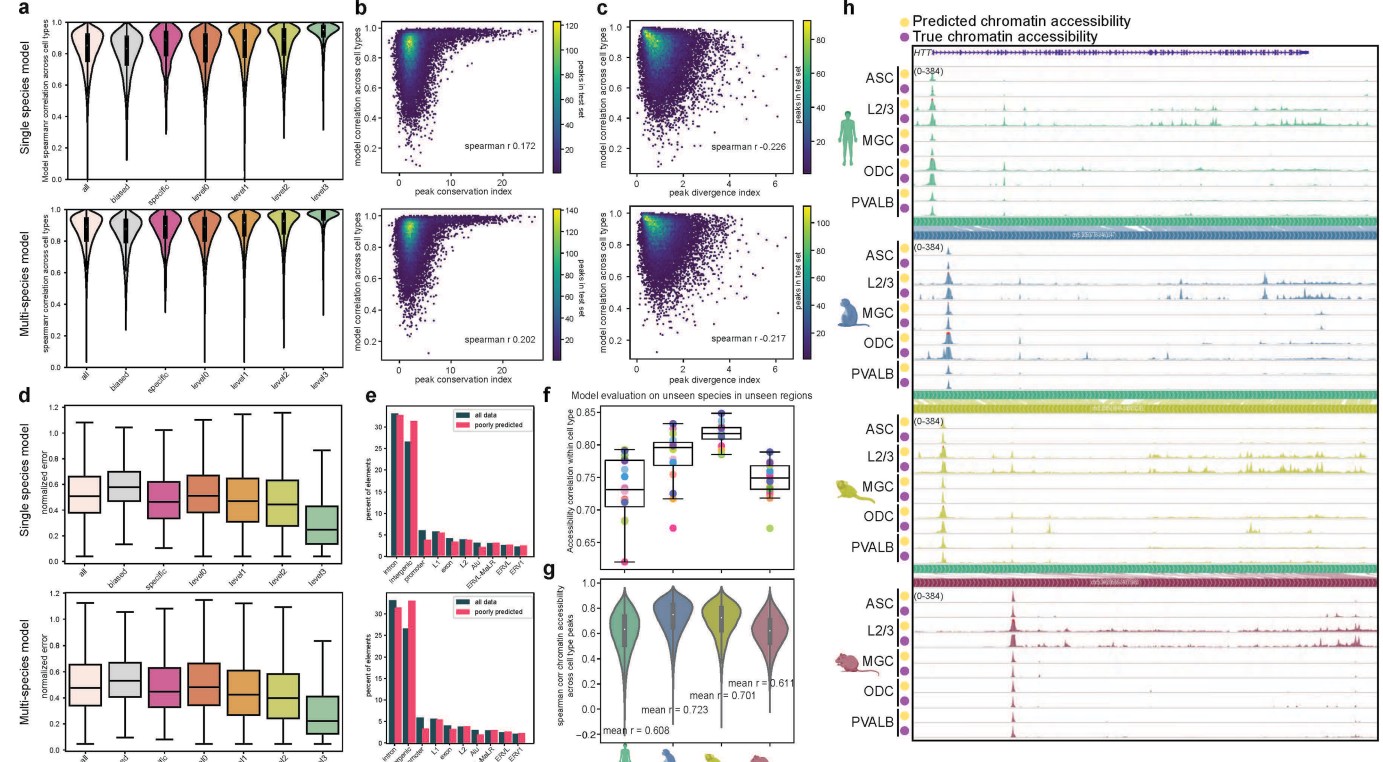

**Extended Data Fig. 11 | Species specificity of open chromatin deep learning.**
**a**. Correlation across cell types for peaks by conservation levels in human test
dataset for single and multi-species models. Violin plots represent the density
of data points. Box plots encompass 25th to 75th percentiles; white dots
represent medians; whiskers represent 1.5 times the interquartile interval.
N = 39,236, 777, 21,737, 6,605, 4,896, 1,493. **b**. Scatter compares the model's
ability to predict chromatin accessibility across cell types (spearman r) and
conservation index in the test set. **c**. Scatter plot compares the model's ability
to predict chromatin accessibility across cell types (spearman r) and divergence
index in the test set. **d**. Box plots show relationship between model accuracy
(mean L1 norm between predictions and true data) and conservation level in
the test dataset. Box plots encompass 25th to 75th percentiles; central lines
represent medians; whiskers represent 1.5 times the interquartile interval.

N as in **a**. **e**. Barplot comparing poorly predicted peaks in the top 10 peak
annotations from Homer to each peak annotation in the entire test dataset.
Shown for human only model (top), and multispecies model (bottom).
N = 39,236 peaks. **f**. Accuracy of a three-species model across cell types with
each species as an outgroup. Spearman correlation of model predictions and
measured chromatin accessibility for each cell type, each represented as a dot.
Plotted intervals are the same as in **a**. N = 16 for each. **g**. Correlation of test set
peaks predictions to measured chromatin accessibility across cell types for
each species. Violin plots represent the density of data points. Plotted intervals
are the same as in **d**. N = 39,236 (human), 44,311 (macaque), 32,484 (marmoset),
and 41,605 (mouse) test set peaks. **h**. True and predicted chromatin accessibility
across the *huntingtin* locus in indicated cell types. Species silhouettes in **g** and
**h** created in BioRender.

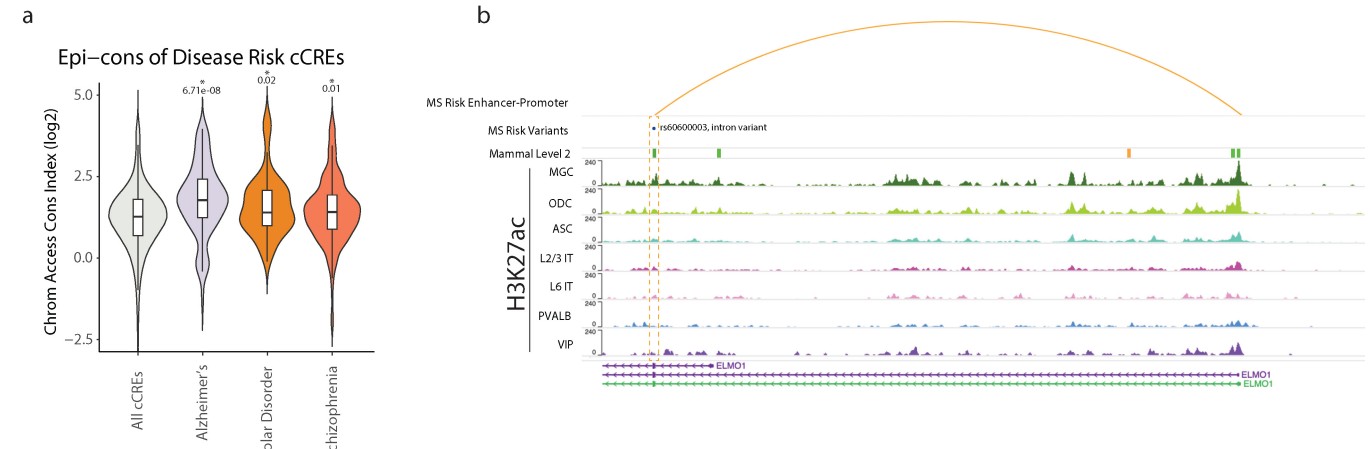

**Extended Data Fig. 12 | Conserved cis-regulatory landscape of disease risk.**
**a**. A violin plot showing conservation index of cCREs containing fine-mapped disease-risk variants in Alzheimer's disease, bipolar disorder, and schizophrenia. Width of violin plots represent the density of data points. Box plots encompass 25th to 75th percentiles; lines inside the boxes represent medians; whiskers represent 1.5 times the interquartile interval. N = 384,412 (All cCREs), 86 (Alzheimer's disease), 49 (Bipolar disorder), 251 (Schizophrenia). Two-sided, unpaired Wilcoxon rank sum test for P-values vs "All cCREs". **b**. Genome browser snapshot showing H3K27ac landscapes of a mammal level 2 predicted enhancer of *ELMO1* overlapping a multiple sclerosis risk variant across cell types.

# Reporting Summary

## Statistics

For all statistical analyses, confirm that the following items are present in the figure legend, table legend, main text, or Methods section.

| n/a | Confirmed | |
|---|---|---|
| ☐ | ☒ | The exact sample size (*n*) for each experimental group/condition, given as a discrete number and unit of measurement |
| ☒ | ☐ | A statement on whether measurements were taken from distinct samples or whether the same sample was measured repeatedly |
| ☐ | ☒ | The statistical test(s) used AND whether they are one- or two-sided<br>*Only common tests should be described solely by name; describe more complex techniques in the Methods section.* |
| ☒ | ☐ | A description of all covariates tested |
| ☐ | ☒ | A description of any assumptions or corrections, such as tests of normality and adjustment for multiple comparisons |
| ☐ | ☒ | A full description of the statistical parameters including central tendency (e.g. means) or other basic estimates (e.g. regression coefficient) AND variation (e.g. standard deviation) or associated estimates of uncertainty (e.g. confidence intervals) |
| ☐ | ☒ | For null hypothesis testing, the test statistic (e.g. $F$, $t$, $r$) with confidence intervals, effect sizes, degrees of freedom and $P$ value noted<br>*Give P values as exact values whenever suitable.* |
| ☒ | ☐ | For Bayesian analysis, information on the choice of priors and Markov chain Monte Carlo settings |
| ☒ | ☐ | For hierarchical and complex designs, identification of the appropriate level for tests and full reporting of outcomes |
| ☐ | ☒ | Estimates of effect sizes (e.g. Cohen's *d*, Pearson's *r*), indicating how they were calculated |

*Our web collection on statistics for biologists contains articles on many of the points above.*

## Software and code

Policy information about availability of computer code

| Data collection | No software was used for data collection. |
|---|---|
| Data analysis | cellranger-arc v2.0.0, cellranger-atac v2.0.0, cellranger v6.1.2, MACS2, YAP v1.6.8, cutadapt, v2.10, bismark v0.20, bowtie2 v2.3, allcools v1.0.8, methylpy, scHiCluster, cooltools v0.5.1, liftOver, HOMER, GSEApy, edgeR v3.36.0, FIMO v5.5.3, wigToBigWig, bigWigAvgOverBed, bedtools multicov, Seurat v4, DoubletFinder v2.0.3, Harmony v0.1.0, ABC: run.neighborhoods.py & predict.py , Basenji v0.6 |

For manuscripts utilizing custom algorithms or software that are central to the research but not yet described in published literature, software must be made available to editors and reviewers. We strongly encourage code deposition in a community repository (e.g. GitHub). See the Nature Portfolio guidelines for submitting code & software for further information.

## Data

Policy information about availability of data

All manuscripts must include a data availability statement. This statement should provide the following information, where applicable:

- Accession codes, unique identifiers, or web links for publicly available datasets
- A description of any restrictions on data availability
- For clinical datasets or third party data, please ensure that the statement adheres to our policy

Data produced in this study are available in the NCBI Gene Expression Omnibus (GEO) under accession number GSE229169 for 10x multiome, GSE240297 for sn-m3C-seq and GSE246760 for Droplet Paired-Tag. Data is uploaded for viewing on the WashU Comparative Epigenome Browser data hub: https://

epigenome.wustl.edu/BrainComparativeEpigenome/. Reference genomes used are hg38, mm10, Mmul_10, cj1700_1.1.

# Research involving human participants, their data, or biological material

Policy information about studies with human participants or human data. See also policy information about sex, gender (identity/presentation), and sexual orientation and race, ethnicity and racism.

| | |
|---|---|
| Reporting on sex and gender | Human subjects were male. |
| Reporting on race, ethnicity, or other socially relevant groupings | *Please specify the socially constructed or socially relevant categorization variable(s) used in your manuscript and explain why they were used. Please note that such variables should not be used as proxies for other socially constructed/relevant variables (for example, race or ethnicity should not be used as a proxy for socioeconomic status).*<br>*Provide clear definitions of the relevant terms used, how they were provided (by the participants/respondents, the researchers, or third parties), and the method(s) used to classify people into the different categories (e.g. self-report, census or administrative data, social media data, etc.)*<br>*Please provide details about how you controlled for confounding variables in your analyses.* |
| Population characteristics | Donor ID, Sex, Age, PMI, hemisphere, Cause of Death<br>H19.30.001, Male, 42, 8h, right, suicide<br>H19.30.002, Male, 29, 7.5h, right, pulmonary embolism<br>H19.30.004, Male, 58, 12h, right, witnessed cardiac arrest |
| Recruitment | Postmortem donors with no known neuropsychiatric or neurological conditions between ages 18 and 68 were considered for inclusion. |
| Ethics oversight | Permission was obtained from decedent next-of-kin. Postmortem tissue collection was performed in accordance with the provisions of the United States Uniform Anatomical Gift Act of 2006 described in the California Health and Safety Code section 7150 (effective 1/1/2008) and other applicable state and federal laws and regulations. The Western Institutional Review Board reviewed tissue collection processes and determined that they did not constitute human subjects research requiring institutional review board (IRB) review. |

Note that full information on the approval of the study protocol must also be provided in the manuscript.

# Field-specific reporting

Please select the one below that is the best fit for your research. If you are not sure, read the appropriate sections before making your selection.

☒ Life sciences ☐ Behavioural & social sciences ☐ Ecological, evolutionary & environmental sciences

For a reference copy of the document with all sections, see nature.com/documents/nr-reporting-summary-flat.pdf

# Life sciences study design

All studies must disclose on these points even when the disclosure is negative.

| | |
|---|---|
| Sample size | Sample size was not predetermined, and were limited to available samples. Number of subjects were n = 3 for human, n = 3 for macaque, n = 3 for marmoset, and n = 8 mice. |
| Data exclusions | Low quality nuclei were removed from datasets. After predicted doublets were removed we used quality metrics to further remove low quality cells. For 10x multiome nuclei were required to have ≥1000 ATAC fragments and ≥500 genes detected. For sn-m3c-seq nuclei were required to have overall mCCC level < 0.05, overall mCH level < 0.2, overall mCG level < 0.5, total final reads > 500,000, and < 10,000,000, Bismarck mapping rate > 0.5. For Droplet Paired-Tag nuclei were required to have ≥200 genes detected. |
| Replication | Sample quality metrics, data quality metrics, and cell type proportions were assessed for each individual sample and displayed high reproducibility, n = 3 for primates, n = 4 for mouse. To obtain adequate sample sizes for downstream analysis, data across samples were subsequently combined for each cell type within each species individually. |
| Randomization | All samples were effectively controls, therefore randomization was not used and all samples were included in the same experimental group. |
| Blinding | Researchers used samples that were labeled with IDs and no identifying donor information within species, however researches were not blind to the species for each sample to exercise appropriate safety protocols. |

# Reporting for specific materials, systems and methods

We require information from authors about some types of materials, experimental systems and methods used in many studies. Here, indicate whether each material, system or method listed is relevant to your study. If you are not sure if a list item applies to your research, read the appropriate section before selecting a response.

## Materials & experimental systems

| n/a | Involved in the study |
|-----|------------------------|
| ☐ | ☒ Antibodies |
| ☒ | ☐ Eukaryotic cell lines |
| ☒ | ☐ Palaeontology and archaeology |
| ☐ | ☒ Animals and other organisms |
| ☒ | ☐ Clinical data |
| ☒ | ☐ Dual use research of concern |
| ☒ | ☐ Plants |

## Methods

| n/a | Involved in the study |
|-----|------------------------|
| ☒ | ☐ ChIP-seq |
| ☒ | ☐ Flow cytometry |
| ☒ | ☐ MRI-based neuroimaging |

## Antibodies

| | |
|---|---|
| Antibodies used | anti-H3K27ac (Abcam, ab4729), anti-NeuN antibody (MAB377X, Millipore) |
| Validation | ab4729 validated by manufacturer for ICC/IF, WB, IHC-P, ChIP, PepArr. Validated for droplet paired-tag signal-to-noise through transcription start site enrichment analysis. MAB377X is validated by manufacturer for IHC and reactivity in human, mouse, and rat. |

## Animals and other research organisms

Policy information about studies involving animals; ARRIVE guidelines recommended for reporting animal research, and Sex and Gender in Research

| | |
|---|---|
| Laboratory animals | 3 macaque donors (6 y.o. male Macaca mulatta, 6 y.o. male Macaca mulatta, and 14 y.o. male Macaca fascicularis), 3 marmoset (Callithrix jacchus) donors (5 y.o. male, 4 y.o. male, and 6 y.o. female), and MOp from 8 P56 C57BL/6J male mice (Mus musculus). C57Bl/6J animals, purchased from Jackson Laboratories, were kept for up to 10 days in the Salk animal barrier facility on a 12-hour dark/light cycle, under controlled temperature (between 20-22 degrees Celsius) and food ad-libitum. |
| Wild animals | No wild animals were used in this study |
| Reporting on sex | 3 male macaque, 2 male and 1 female marmoset, 8 male mice |
| Field-collected samples | No field collected animals were used in this study |
| Ethics oversight | Mouse experiments were approved by the SALK Institute Animal Care and Use Committee under protocol number 18-00006. Marmoset experiments were approved by and in accordance with the Massachusetts Institute of Technology IACUC protocol number 05170520. Macaque experiment protocols were approved by the University of Washington Institutional Animal care and Use Committee. |

Note that full information on the approval of the study protocol must also be provided in the manuscript.

## Plants

| | |
|---|---|
| Seed stocks | Report on the source of all seed stocks or other plant material used. If applicable, state the seed stock centre and catalogue number. If plant specimens were collected from the field, describe the collection location, date and sampling procedures. |
| Novel plant genotypes | Describe the methods by which all novel plant genotypes were produced. This includes those generated by transgenic approaches, gene editing, chemical/radiation-based mutagenesis and hybridization. For transgenic lines, describe the transformation method, the number of independent lines analyzed and the generation upon which experiments were performed. For gene-edited lines, describe the editor used, the endogenous sequence targeted for editing, the targeting guide RNA sequence (if applicable) and how the editor was applied. |
| Authentication | Describe any authentication procedures for each seed stock used or novel genotype generated. Describe any experiments used to assess the effect of a mutation and, where applicable, how potential secondary effects (e.g. second site T-DNA insertions, mosiacism, off-target gene editing) were examined. |

