## [Peer Review File · Nature]

Manuscript Title: Comparative single cell epigenomic analysis of gene regulatory programs in the rodent and primate neocortex

Reviewer Comments & Author Rebuttals

Reviewer Reports on the Initial Version:

Referees' comments:

Referee #1:

Remarks to the Author:

This manuscript reports the gene regulatory landscape in 21 cortical cell types in four species, human, macaque, marmoset, and mouse. Four modalities related to gene expression (RNA-seq) or gene regulation (ATAC-seq, DNA methylation, and chromatin architecture inferred from contact frequencies) were ascertained in sequencing based assays in single cells. This approach allowed the assignment of the data for each epigenetic modality to a specific cortical cell type in a data-driven manner (clustering in reduced dimensions, e.g. by UMAP) and the use of marker genes for known cell types. By doing the same assays and analyses in the four species, one can examine changes in these aspects of the epigenetic across three evolutionary time frames, specifically the times since humans and macaques diverged, primate evolution, and euarchontoglires evolution (since the separation of primates and rodents). The results of this study are a wealth of valuable data. The analysis of epigenetic evolution across species is complicated by the additional dimension of cell types to the comparisons. This manuscript utilizes a set of informative categories, called conservation levels X, 0, 1, 2, and 3 (for some features). Other categories could be considered, but the chosen ones are informative when well defined (which is not the case for all features). Overall, the manuscript describes the data well and draws several informative conclusions from the wide range of observations. Featured conclusions include that regulatory elements and features associated with cell type-specific gene expression evolve more rapidly than those for widely expressed genes, that distal regulatory elements evolve more rapidly than proximal ones, and the transposable elements are a major source of elements regulating species-specific gene expression. While none of these conclusions is completely novel, the wealth and depth of data supporting them is a major contribution. The manuscript also shows the power of using the epigenetically conserved regulatory elements to improve the analysis and interpretation of trait-associated genetic variants and in inferring gene targets of regulatory elements.

While the data and major conclusions in this manuscript are important contributions, the presentation has several issues, some of which make it difficult to understand the observations and analyses. The following points should be addressed in revision. They start with two larger questions, and then there is a long list of specific issues.

1. Two multi-modal assays were employed in these experiments. One of the rationales for using multi-modal assays is so that two different aspects, e.g. RNA-seq and ATAC-seq, can be determined in the same cell, whereas that is not possible with single cell assays for a single modality. Was this aspect of multi-modal assays used in the analyses of the data? It seems like each modality was analyzed separately in this work.

2. The manuscript should discuss an additional category of epigenetic (non)conservation, which are candidate cis-regulatory elements (cCREs) that are conserved in sequence (level 0) across mammals or primates, but they show no evidence of epigenetic conservation. For mammal level 0, there are about 205,000 cCREs, of which about 132,000 were assigned to one of the three levels of epigenetic conservation. That would appear to leave about 73,000 with no evidence of epigenetic conservation. Is that a correct estimate? How does one interpret them? Perhaps it is an acquisition of a new regulatory function in human for a segment of DNA that pre-dates the mammal-rodent separation. This process has been called "exaptation" or "repurposing" in previous work. Others may argue that they are largely

"noise" in the establishment of the regulatory landscape. That is not my favorite notion, but it is an example of an interpretation espoused by some investigators.

3. L77-79 need to be clarified: "Additionally, not all functional elements are conserved, and some non-functional elements may appear to be conserved due to sequence similarity". The first clause is fine, but the second clause is hard to follow. Perhaps what is being referred to is that DNA sequences align across different species does not necessarily mean that they have a function. Even neutral DNA can align across a particular phylogenetic span.

4. L81-81 It is likely that this sentence should refer to altered expression across species "In other words, how sequence divergence leads to altered gene expression patterns across different cell types remains largely unexplored."

5. The graph of divergence index versus conservation index in Fig. 2c is currently colored by by numbers of genes in each bin. It would be highly informative to also include a graph colored by the deduced categories described on L171-175. Those category labels are listed in the graph but it is not shown where the genes in each category lie on the spectrum (i.e. divergence index versus conservation index).

6. The text L178-179 says that "the majority of mammal-conserved genes displayed broad gene expression patterns across cell types" but the pie charts in Ext. Data Fig. 2d shows Mammal Conserved with the smallest fraction of ubiquitous genes. This apparent discrepancy needs to be resolved.

7. L189: The reference to Extended data Fig. 5c,d may be an error, since that figure does not have the data described in the text. Perhaps it should be Extended data Fig. 2c,d.

8. It appears that Fig. 2d and 2e are not described or discussed in the text.

9. L213: The diagram in Fig. 3a indicates that the criterion for level 2 epiconserved elements was that they displayed chromatin accessibility in *at least one of the* same cell types across species (*..* demarcate a suggested addition). If this inference is correct, then the text should be revised accordingly.

10. The text in L219-230 is difficult to follow, especially with respect to the cited figure panels (Fig. 3b-d). This text and accompanying figures needs to be re-worked and clarified, with simpler figure panels to show the relevant data. The next 4 points are specific examples of this issue.

11. L223 states an informative conclusion, but the numbers of cCREs in each category listed in L223-228 cannot readily be deduced from the cited panels in Fig. 3.

12. Fig. 3b: It is not clear what the second set of bar graphs ("level 1") are showing. Is this figure supposed to support the observation on L224 that "66,781 (32.5%) were classified as level 1"? That number is not apparent in any of the three graphs. Why use 3 graphs to support one number? What are the partial levels?

13. Fig. 3c: This complex figure is difficult to interpret, and it is not clear what messages are being conveyed. Neither the Results nor the legend provide the needed information. Do the species images in the middle refer to the cell types on the outer circle? What is learned from the breakdown of the results by cell types? What conclusion can be drawn from the inner circles, and can it be shown in a less complicated figure panel? Is this figure supposed to support the observation on L224 that "49,135 (24.0%) as level 2"? That number is one value in very small print on one of the inner circles; this information could be conveyed in a MUCH more accessible manner.

14. Fig. 3d: This panel uses the divergence index versus conservation index spectrum again, apparently aiming to support the statement on L225 of "16,068 (7.8%) as level 3 epi-conservation". That number is provided as a inset to one of the spectra (graphs). The actual data shown are not for cCREs with level 3 epi-conservation, but rather it is for all the cCREs with level 0 sequence conservation. One cannot deduce which points in the spectra correspond to the cCREs in the category of interest (level 3 epi-conservation).

15. The data in Fig. 3h are shown at a high level of granularity, with a full matrix of TE classes versus cell types. However, the results as presented are difficult to align with the statements in the text (L249-252). For example, the text says that "Human-specific cCREs from excitatory neurons (except for L5/6 NP and L6 CT) are most enriched in L1", but the image in Fig. 3h shows a strong enrichment for these TEs across all the cell types. The more general groups of excitatory neurons and glial cells are not explicitly labeled in the figure panel. A simpler figure focused on bringing out the key result would be more effective.
16. L228-230: The criteria for categorizing cCREs as having "human-biased" accessibility are not clear. The level 0 cCREs have a peak of accessibility only in human despite having orthologous sequences in other species. How do human-biased accessibility peaks differ from these?
17. Pages 8-9 and Fig 4: Many of the problems of unclear and overly complex figure panels described for the analysis of ATAC-seq peaks apply to this analysis of differentially methylated regions (DMRs).
18. It is not clear how panels e and f for Fig. 4 support the statement on L280, "elevated levels of conservation correlate with lower cell-type specificity".
19. In several places, the "cell-type specificity" is compared for a particular modality across categories, such as levels of epigenetic conservation. The relevant data are shown as a distribution over a range of entropy values, such as in Fig. 3e and Fig. 4d. Given that there are many values for the specificity-related entropy, the manuscript should clarify how the distributions were compared to reach conclusions such as "the level 3 conserved cCREs that are promoter distal have reduced cell type specificity" (L238) and "elevated levels of conservation correlate with lower cell-type specificity" (L280).
20. Pages 9-10 and Fig 5: Many of the problems of unclear and overly complex figure panels described for the analysis of ATAC-seq peaks apply to this analysis of TAD boundaries and loops.
21. The text in L305-308 implies that rules similar to those used for ATAC-peaks and DMRs were used for TAD boundaries and chromatin loops. However, Fig. 5a and 5g introduce new categories, which may be "tissue conserved" and "cell type conserved" (the text is pixelated and difficult to read even at high magnification). The criteria for those categories are not explained, and the schematic figure panels are confusing. It is not apparent what distinguishes "tissue conserved" from "cell type conserved". L308 says that mammal level 2 in this context refers to TADs that are conserved in the same cell types, but Fig. 5a shows the TADs in this category at the same place in DIFFERENT cell types. Fig. 5 and accompanying text needs to be corrected and revised extensively to provide clear evidence to support the claims, and a legible revised figure needs to be provided.
22. L352 suggests that cCREs assigned to a target gene by ABC are more likely to function as enhancers than are cCREs that are not assigned. However, the cCREs not assigned to a target gene by ABC could still act as enhancers, but they could be regulating one or more gene targets that are not revealed by the current chromatin conformation capture data. A more accurate conclusion is that "conserved cCREs are more readily assigned as enhancers for specific target genes". Similarly, the y-axis on Fig. 6d is the "Percent predicted to regulate a target gene".
23. "Syntenic" is not used correctly. Syntenic genes are on the same chromosome ("same string"). Genes that are syntenic in one species may be on the same chromosome in a second species, in which case synteny is conserved. Beyond that, gene order and even arrangement can be conserved. The manuscript should clarify what is meant by "syntenic".
24. The clause on L389-390 is hard to follow, because the training set consisted of orthologous regions in other species. What exactly is the risk that is being avoided? Also, in L387, "data leakage" is not a concept used widely by experimental scientists, and it would be helpful to explain what it means.
25. In Fig. 7, panels d-g are not described or discussed in the text. These panels have helpful

information, showing an overall correspondence between predicted and actual chromatin accessibility signals, BUT with some notable false positives and false negatives. These rich landscapes are reduced to a single number, a correlation across bins, in panel h, which loses the variance in predicting accuracy. Was there anything distinctive about the regions that the model missed for chromatin accessibility or that it over-predicted for accessibility?

26. The "correlations across cell types" in Fig. 7 panels i and j (text L394-398) are not explained sufficiently for the results to be understood. If this correlation is related to the correlation across bins shown in panel h, how the values for the correlation become so much smaller? It is not clear what values are being correlated across cell types.

27. Ref. 2 should be journal article.

To boost transparency, I sign this reviewer's report:
Ross Hardison

Referee #2:

Remarks to the Author:

Comparative single cell epigenomic analysis of gene regulatory programs in the rodent and primate neocortex.

Zemke et al.

Review

In Zemke et al, the authors seek to identify the evolutionary impact on gene regulatory organization in the neocortex. To do so, they apply multiomic measurements of transcription and chromatin state, methylation, and physical chromatin contacts in the mouse, marmoset, macaque, and human primary motor cortex. This is a monumental undertaking and one certainly worth doing. Not surprisingly, this is a well-executed genomic investigation, as the senior authors are experienced and prominent members of the genomics community. The authors identify differences in cellular composition between species, differences in cell-type specific enhancer usage and methylation state, and corresponding changes in nuclear architecture.

Unfortunately, the biological insights of this study fall short of a significant advance. A tremendous amount of interesting data has been collected in this study and certainly can serve as a useful resource article, but it is unclear if Nature is the proper journal for publication. The manuscript drifts between related genomic measurements but is not effective in harmonizing these data sets to identify key biological insights relevant to the evolution of gene regulatory strategies and grammar across species. Instead, a string of observations is made with no overarching biological conclusion or outcome.

In the interest of improving this manuscript, the authors should consider addressing the below issues.

1) In Figure 1, the authors show harmonization between 10X multiome RNA and snm3C-seq data. It is difficult to get a handle on what the authors seek to demonstrate in this figure. Compositional differences are observed, with little commentary on what this actually means in context of the structure and function of motor circuitry. Is there a key take home regarding patterns of methylation, accessibility and transcriptional activation that can explain the variance observed across species? Presumably this is handled in Figure 2, but we are left with gene ontologies and little explanation about how enhancer differences can explain variability in transcriptional activation. The authors show gene ontologies of conserved groups, but can they offer insight into species specific ontological processes and the molecular etiology associated with functional divergence. More broadly, is there any indication that errors in peak-gene association (often done in multiome data) to identify enhancers may simply account for some differences that are observed across species? Can data from these sequencing modalities be integrated effectively to address that concern?

2) I find the categorization and nomenclature of conservation levels is difficult to handle in this manuscript. What are the thresholds for this categorization? How “epi-conserved” does a site need to be to be placed in a given conservation level? The authors might want to consider clarifying in the text the significance of these conservation levels and integrate other sequencing modalities (methylome and chromatin contact) into the interpretation of these results. In figure 3b, the authors show “Level 0 conserved elements, indicating 8K human specific elements, can they offer gene ontology for these categories? Figure 3d is labeled with “Level 3” but the figure caption refers to a “Level 0” comparison between mammalian conservation and primate conservation. If the result is for “Level 3” conservation, then the authors should explain how strong “epi-conservation” could lead to such divergent results. Figure 3g and 3h are the most interesting observations in this manuscript. That said, mammalian “rewiring” by TEs been observed in Choudhary et al (Nature Communications 14, 634 (2023)). Basically, there doesn’t appear to be an overwhelming amount of variability in TEs in cell types shown in figure 3h. Perhaps the authors would uncover exciting biology in figure 3h by showing how cell types change in %cCREs in TEs across species? Unfortunately, an exciting result is diminished by a lack of substantive variation in TEs across cell types, bringing into question what insights have been made beyond what could be gained from bulk experiments.

3) In figure 4, the authors profile the methylome across species. How do these results relate to the findings of figure 3? For example, are there patterns of methylation that might explain genomic silencing that buffers evolutionary differences and results in conservation across species? There appears to be a trend where chromatin accessibility in figure 3 is more conserved in proximal elements (shown in level 3 conserved elements) and the trend is similar in the methylome, albeit to a lesser extent. What is the interpretation of these findings? How does it relate to regulatory convergence and divergence across species? In figure 4, can the authors offer insight into human biased methylation and mammalian conserved patterns similar to figure 3f? Again, are there instances where methylation abrogates the contribution of human specific accessibility? Ultimately, the authors need to integrate these sequencing modalities more effectively with one another to explain the impact of the differences they observe across species. Again, the authors share insight about the % of DMRs overlapping TEs, but there appears to be little variability across cell types in the human. Can they show differences across species and show cell type specific changes in DMRs? How do these data, alongside differences in accessibility, yield differences and similarities in transcriptional activation or lack thereof?

4) The authors continue their work looking at conserved and epi-conserved TAD boundaries across species. An interesting observation is that conserved boundaries yield conserved transcriptional profiles. Can the authors expand their interpretation of these results. CTCF is known to be sensitive to methylation. Can the authors rationalize any of their observed differences in TAD formation, with changes in the methylome binding? Granted, within a given species ~1.5% of CTCF sites are influenced by methylation. Are there differences of these effects across species and could that explain differences in TAD boundaries and subsequent transcriptional differences?

5) In Figure 6, the authors identify epigenetic conservation at cCREs and identify predicted conserved enhancers and their influence on transcriptional conservation. The authors have published single cell methods that can offer significant value in these studies, profiling chromatin state (enhancers) and transcriptional activation simultaneously and in single cells (Nature Methods, 2021). I think it is essential that the authors show agreement between their predicted enhancer classes and their agreement with experimental proof that there are bona fide enhancers that activate transcription across species. I recognize this is an undertaking but given that the method was pioneered by the senior investigator, a modest attempt at validation is reasonable. Speaking more generally, throughout the manuscript, the authors appear to avoid a key question: what is the significance of species specific TEs that are not enhancers? What functionality do they offer and how might this impact the evolution of transcriptional regulation?

6) In Figure 7, the authors modify a deep learning approach to predict cell type specific patterns of chromatin accessibility. Can the authors comment on this approach in context of other comparative studies of enhancer identification and impact, notably TACIT (Kaplow et. al. Science, vol 380, no 6643). This section of the manuscript, frankly, feels like a distraction. It is unclear how this relates to the

broader study and, furthermore, the authors should consider identifying chromatin accessibility using this deep learning model in an out group and then then conduct a simple snATAC-Seq experiment to demonstrate that this approach is general. If the idea is that this resource offers value towards the interrogation of other species, with predictive power, then an out-group study is essential.

7) The authors end this study with an important statement that these comparative studies could offer insights into noncoding genetic variation related to neurological disease and disorders. The authors show that a conserved regulatory element upstream of ELMO1 intersects with risk variant rs606000003, which is exclusively accessible in microglia. There is the suggestion that expression of ELMO1 may be impacted in a cell type specific manner that contributes to MS. While this may offer some rationale for prioritization of noncoding variation, the authors have tools to test this through a single cell CUT&Tag measurement of differential binding of the regulator that binds at this impacted cognate binding site. Furthermore, it remains unclear how epigenetic conservation offered more value than a simple multiomic measurement.

Summary

There are several claims made in the abstract of this manuscript. Namely: 1) Cell type specific gene expression evolves more rapidly than broadly expressed genes, 2) Epigenetic status at distal candidate cCREs evolves faster than promoters and 3) That transposable elements contribute to 80% of human specific cCREs in the primary motor cortex. I find that claims 1 and 2 are weakly supported and claim 3, which is interesting, is difficult to biologically interpret given the authors statement that "In general, cCREs located in TEs are less often predicted to function as enhancers." The authors should consider toning down language regarding "evolutionary speed" given that it is not addressed formally within the manuscript. Finally, the authors should be commended for pointing out that "Previous bulk sequencing assays have revealed general principles concerning the conservation of cis-regulatory elements and tissue-specific gene expression patterns. For example, enhancers exhibit rapid turnover during mammalian evolution and those conserved have lower cell type specific activity. In contrast sequence divergent enhancers play a significant role in establishing tissue and species-specific traits. Such divergent enhancers are often mediated by de novo insertion of transposable elements carrying clusters of transcription factor binding sites." The authors follow up with references supporting the aforementioned. Unfortunately, the bulk studies demonstrate biological phenomena that are seen in this study as well, bringing into question the biological novelty of this study.

Basically, I struggled with the question of whether the biological significance of this study is sufficient to justify publication as a Nature article, or should be published as a data resource. I am happy to opine on a revised manuscript that addresses my concerns or to be convinced that my concerns are unreasonable. I would hope that with all of the intellectual firepower in the authors list, more compelling, and novel biological insights could be provided based on this extraordinary data set.

Author Rebuttals to Initial Comments:

We are grateful to both reviewers for their constructive comments on our previous submission! We fully agree with their concerns and address them in the revision, by including new analyses and discussions. We have also added several new dimensions to the study. The major changes are briefly highlighted below, followed by point-by-point responses to the reviewers' comments. The reviewer comments are pasted below in *blue italic* font followed by our responses in black regular font. We also use the **red** font to indicate new texts added to the revised manuscript.

List of major updates:

- We carried out new integrative analyses to explore the relationships between the evolution of gene expression of transcription factors and the epi-conservation of candidate cis regulatory elements (cCREs) in different cell types. We showed that divergent transcription factor gene expression likely promotes species-biased gene expression patterns in corresponding cell types.
- We adopted a new framework of deep learning algorithm, Basenji, to develop sequence-based predictors of chromatin accessibility in different cell types. We found that use of multiple epigenetic modalities, namely ATAC-seq and DNA methylation signals from multiple species as training data resulted in substantial improvement in predictive performance. The newly trained deep learning can accurately predict cell-type-specific open chromatin regions in the four species.
 - We further carried out Droplet Paired-Tag experiments to profile jointly histone H327ac and gene expression at single cell resolution from human M1 (Extended Data Fig. 10). Together with our recently published mouse Droplet Paired-Tag cortical data, we demonstrated that epigenetic conserved cCREs are more likely to be active enhancers than the rest of the cCREs, thereby confirming the hypothesis that epigenetic conservation marks functional enhancers.
 - We also substantially simplified our figures to clearly show the relationship between conservation levels (Fig. 2b, Fig. 3a, Fig. 4b,g, 5e), and added a detailed workflow into Extended Data fig. 3a.

Beyond these updates we have significantly restructured and expanded the figures in our manuscript. Below we describe the major changes for each figure:

Figure 1:

- Combined figures 1 and 2.
- Original Fig 2c has been split into two panels: Fig 1h and 1i to separately highlight mammal level 3 and human-biased genes.
- Added gene ontology analysis for human-biased genes in Fig 1l and 1m.

Figure 2:

- Original Fig 3 is now Fig 2.
- Moved conservation level bar plots to Extended Data Fig 3d,e.
- Added Venn diagram of conservation levels for 2b.

- Original Fig 3d is now Fig 2c,d to separately highlight mammal level 3 and human-biased elements.
- Added 3h showing TF motif epi-conservation analysis
- Added 3i showing correlation between epi-divergence with TF expression divergence.
- Added 3j showing correlation between epi-conservation with TF motif sequence conservation.

Figure 3:

- Original Fig 4 is now Fig 3.
- Moved conservation level bar plots to Extended Data Fig 5a,b.
- Added Venn diagram of conservation levels for 2b.
- Added a scatter plot comparing chromatin accessibility conservation and DMR conservation as Fig 3b.
- Added TF motif epi-conservation analysis of DMRs as Fig 3f.

Figure 4:

- Original Fig 5 is now Fig 4.
- Moved conservation bar plots, panels b, c, h, and i, to Extended Data Fig 6a,b,7a,b.
- Added Venn diagrams of conservation levels for 4b,g.
- Added bar plots of numbers of conserved and divergent boundaries for each cell type.

Figure 5:

- Original Fig 6 is now Fig 5
- The new Figure 5 has undergone significant reorganization
- The majority of panels from old Fig 6 are now in Extended Data Fig. 8.
- New panels highlight enhancer activity measured through H3K27ac across levels
- simplified summary of ABC levels,
- correspondence between changes in enhancer activity and gene expression in human divergent Enhancer-Gene pairs
- human divergent gene regulatory networks for *FOXP2* in microglia and *RYR3* in astrocytes.

Figure 6:

- Original Fig 7 is now Fig 6
- undergone a significant reorganization and all panels have been replaced or modified.

Figure 7:

- Original Fig 8 is now Fig 7

Extended Data Fig 1:

- Combined Extended Data Fig 1 and 3.

Extended Data Fig 2:

- Added gene expression conserved and divergent heatmaps originally Fig 2d.
- Improved visual clarity of cell specificity density plots and fixed mislabeled pie charts in panel d.
- Added gene ontology analysis of conserved genes as panels e and f.
- Added gene ontology of human-biased genes for individual cell types as panel g, and highlighted the genes in specific terms in panel h and i.

Extended Data Fig 3:

- Added new schematics precisely outlining the workflow for identification of each level across species. New schematic outlining the identification of human biased cCREs.
- Moved pairwise comparisons of level 0, 1, 2 from Old Fig. 3.
- Added Highlights of each conservation level to the conservation/divergence index plot.

Added comparison of sequence conservation to peak epi-conservation and motif epi-conservation.

Extended Data Fig 4:

- Moved pairwise conserved/divergent comparisons to Extended Data Fig. 3 added human cell type level TE enrichment analysis from old Fig 3.
- Added pairwise level comparisons to Extended Data 5 from old Fig 4.
- Added Extended data Fig. 6 which contains pairwise conservation of TADs from old Fig. 5, and has new analysis highlighting the relationship between CTCF peaks, and CTCF motifs in conserved and divergent boundaries across species. Also highlights the changes in methylation at CTCF motifs between conserved and divergent boundaries.

Added Extended Data Fig. 7:

- which recapitulates pairwise conservation of loops across species from old figure 5, TE enrichment in loops (Old Extended Data Fig 6, and highlights within species correspondence between called loops and cCREs across cells.

Added Extended Data Fig. 8:

- Recapitulates most of the old Fig. 5 and expands it to highlight GO biological processes in human divergent microglia gene regulatory networks, the relationship between conservation and ABC score, and a human divergent microglia gene-regulatory network at *RIN3*.

Added Extended Data Fig. 9:

- Addresses potential issues in sequence mappability found dis-proportionately in cCREs with human specific sequence.

Added Extended data Fig. 10:

- New analysis highlighting new human H3K27ac droplet-paired tag data.
- H3K27ac activity at conserved and human-biased elements between human and mouse, human/mouse conservation index and cCREs with level 3 conserved activity, and the relationship between chromatin accessibility conservation and H3k27ac conservation.

Added Extended Data Fig. 11:

- Replaces old Extended Data Fig 8 with completely redone analysis highlighting the predictive power of sequence-epigenome deep learning models across conservation levels, and showing predictive power when applied to an out group.

Added Extended Data Fig. 12:

- corroborates the microglia ELMO1 enhancer with H3K27ac, and shows fine-mapped variants have enriched epi-conservation compared to other cCREs .

Referees' comments:

Referee #1 (Remarks to the Author):

This manuscript reports the gene regulatory landscape in 21 cortical cell types in four species, human, macaque, marmoset, and mouse. Four modalities related to gene expression (RNA-seq) or gene regulation (ATAC-seq, DNA methylation, and chromatin architecture inferred from contact frequencies) were ascertained in sequencing based assays in single cells. This approach allowed the assignment of the data for each epigenetic modality to a specific cortical cell type in a data-driven manner (clustering in reduced dimensions, e.g. by UMAP) and the use of marker genes for known cell types. By doing the same assays and analyses in the four species, one can examine changes in these aspects of the epigenetic across three evolutionary time frames, specifically the times since humans and macaques diverged, primate evolution, and euarchontoglires evolution (since the separation of primates and rodents). The results of this study are a wealth of valuable data. The analysis of epigenetic evolution across species is complicated by the additional dimension of cell types to the comparisons. This manuscript utilizes a set of informative categories, called conservation levels X, 0, 1, 2, and 3 (for some features). Other categories could be considered, but the chosen ones are informative when well defined (which is not the case for all features). Overall, the manuscript describes the data well and draws several informative conclusions from the wide range of observations. Featured conclusions include that regulatory elements and features associated with cell type-specific gene expression evolve more rapidly than those for widely expressed genes, that distal regulatory elements evolve more rapidly than proximal ones, and the transposable elements are a major source of elements regulating species-specific gene expression. While none of these conclusions is completely novel, the wealth and depth of data supporting them is a major contribution. The manuscript also shows the power of using the epigenetically conserved regulatory elements to improve the analysis and interpretation of trait-associated genetic variants and in inferring gene targets of regulatory elements.

While the data and major conclusions in this manuscript are important contributions, the presentation has several issues, some of which make it difficult to understand the observations and analyses. The following points should be addressed in revision. They start with two larger questions, and then there is a long list of specific issues.

1. Two multi-modal assays were employed in these experiments. One of the rationales for using multi-modal assays is so that two different aspects, e.g. RNA-seq and ATAC-seq, can be determined in the same cell, whereas that is not possible with single cell assays for a single modality. Was this aspect of multi-modal assays used in the analyses of the data? It seems like each modality was analyzed separately in this work.

Response: We are extremely thankful for this important critique, which has led us to substantially improve the manuscript. We agree that we did not take full advantage of the multimodality of our data in the first submission. The integrative analysis was limited to Figure 5, where we linked evolution of TADs and loops to conservation of gene expression patterns and open chromatin, and in Figure 6, where we correlated the different levels of conservation of chromatin accessibility in cCREs to that of expression levels of potential target genes. The reviewer's critiques prompted us to significantly expand the multi-modal data integration in the revision. We have carried out several new integrative analyses that take advantage of the single-cell multimodal nature of our datasets. These include:

- (a) we integrated gene expression data with chromatin accessibility data to examine the relationships between the gene expression patterns of transcription factors in different species and the evolution of cCREs in the corresponding cell types. We found a striking correlation between the divergence of TF expression patterns and the divergence of chromatin accessibility of cCREs (new Figure 2i).
- (b) we integrated DNA methylation data with chromatin accessibility data to examine whether the two epigenetic features co-evolve. Indeed, we found that conservation of chromatin accessibility at cCREs is significantly correlated with conservation of differential DNA methylation (new Figure 3b). Additionally, the conservation of differential DNA methylation at transcription factor motifs is significantly correlated with the conservation of chromatin accessibility at cCREs (new Figure 3f).
- (c) we performed in-depth analysis of evolution of enhancer-gene pairs in different species by jointly analyzing the chromatin accessibility and gene expression in each cell type across the species. In new Figure 5e-5h, we described the different groups of enhancer-gene pairs defined based on the conservation (or lack of). Importantly, we defined a group of 464 human biased enhancers that appear to drive human biased genes (Figure 5g). Among them are FOXP2, which shows human specific expression in microglia, which is correlated with chromatin accessibility at a cCRE exclusively in the human microglia, but not in other primates or rodents (Figure 5h). Another example is RYR3, which exhibits expression in the human astrocytes but not in other species, and corresponding human biased chromatin accessibility in astrocytes but not in other cell types (Figure 5h).
- (d) we developed new machine learning models to predict cell-type-specific open chromatin profiles from DNA sequence information alone in each species, by using single modality (ATAC-seq) or dual modalities (ATAC-seq and DNA methylation) as training data. We showed that including DNA methylation significantly elevated the performance of the neural network models (new Figure 6, new expanded data figure 11).

In summary, through these additional integrative analyses, we gained new insights into the evolution of gene regulatory programs, made only possible because of the single cell multiomic datasets generated in this study.

2. The manuscript should discuss an additional category of epigenetic (non)conservation, which are candidate cis-regulatory elements (cCREs) that are conserved in sequence (level 0) across mammals or primates, but they show no evidence of epigenetic conservation. For mammal level 0, there are about 205,000 cCREs, of which about 132,000 were assigned to one of the three levels of epigenetic conservation. That would appear to leave about 73,000 with no evidence of epigenetic conservation. Is that a correct estimate? How does one interpret them? Perhaps it is an acquisition of a new regulatory function in human for a segment of DNA that pre-dates the mammal-rodent separation. This process has been called "exaptation" or "repurposing" in previous work. Others may argue that they are largely "noise" in the establishment of the regulatory landscape. That is not my favorite notion, but it is an

example of an interpretation espoused by some investigators.

Response: Thank you for the note. You are correct that our criteria for categorizing epigenetic conservation or divergence is not all encompassing, causing cCREs to be left out of conservation and divergent groups. We took a stringent approach, where cCREs were excluded from both categories if they did not have conservation across all four species or all three primates, and were not considered divergent if they did not show significantly biased activity in one species compared to all others. This means many elements which were conserved or divergent only between one or two species comparisons are not represented in downstream analyses. To investigate new regulatory functions for cCREs where its DNA predates the mammal-rodent separation, we focused on a high confidence list of ~11k cCREs that are significantly higher in humans than the other species in the same cell type. And an additional ~12k cCREs that are sequence conserved only in primates but have activity only in human. Although these strict criteria left out many potentially interesting cCREs, we were still able to utilize these lists to identify “repurposed” DNA for human specific gene regulation.

3. L77-79 need to be clarified: "Additionally, not all functional elements are conserved, and some non-functional elements may appear to be conserved due to sequence similarity". The first clause is fine, but the second clause is hard to follow. Perhaps what is being referred to is that DNA sequences align across different species does not necessarily mean that they have a function. Even neutral DNA can align across a particular phylogenetic span.

Response: We thank the reviewer for this thoughtful comment! We have updated the text of the manuscript to further clarify: “sequence conservation alone cannot provide definitive evidence of the functional role of a regulatory element since not all functional elements have conserved sequences, and some non-functional elements may be sequenced conserved. Also, sequence conservation cannot reveal information about the cell/tissue-specific activity of an element.”

4. L81-81 It is likely that this sentence should refer to altered expression across species "In other words, how sequence divergence leads to altered gene expression patterns across different cell types remains largely unexplored."

Response: Thank you for catching this error. We’ve updated the text to reflect this suggestion: “In other words, how sequence divergence leads to altered gene expression patterns across different species remains largely unexplored.”

5. The graph of divergence index versus conservation index in Fig. 2c is currently colored by numbers of genes in each bin. It would be highly informative to also include a graph colored by the deduced

categories described on L171-175. Those category labels are listed in the graph but it is not shown where the genes in each category lie on the spectrum (i.e. divergence index versus conservation index).

Response: Thank you for this suggestion! We have added these plots as Extended Data figure 3g and altered the plots in the main figure, now Fig. 1i,h.

6. The text L178-179 says that "the majority of mammal-conserved genes displayed broad gene expression patterns across cell types" but the pie charts in Ext. Data Fig. 2d shows Mammal Conserved with the smallest fraction of ubiquitous genes. This apparent discrepancy needs to be resolved.

Response: Thank you for catching this discrepancy, which was our error. The pie charts in Ext. Data Fig. 2d were mislabeled in the initial submission. We have updated the figure. We now see that ubiquitous genes are enriched in our mammal conserved gene category.

7. L189: The reference to Extended data Fig. 5c,d may be an error, since that figure does not have the data described in the text. Perhaps it should be Extended data Fig. 2c,d.

Response: Thank you for catching this error, we have updated the text to reference the correct figures.

8. It appears that Fig. 2d and 2e are not described or discussed in the text.

Response: Thanks again for your careful and perceptive examination of the manuscript. We are deeply sorry for the errors in the initial submission. We've updated the text to reference these figures, now Extended Data Figure 2c, by adding the following: **"Non-ubiquitous conserved genes show highly correlated patterns of expression across cell types (Extended data Fig. 2c)."**

*9. L213: The diagram in Fig. 3a indicates that the criterion for level 2 epiconserved elements was that they displayed chromatin accessibility in *at least one of the* same cell types across species (*..* demarcate a suggested addition). If this inference is correct, then the text should be revised accordingly.*

Response: Thank you for the helpful suggestion! We agree that this helps to clarify level 2 conservation. We have updated to the text to on L213 to read **" 'level 2' are cCREs displaying accessibility in at least one of the same cell types across species"**

10. The text in L219-230 is difficult to follow, especially with respect to the cited figure panels (Fig. 3b-d). This text and accompanying figures needs to be re-worked and clarified, with simpler figure panels to show the relevant data. The next 4 points are specific examples of this issue.

Response: We agree this figure and text can be difficult to decipher. We have panels 3b-d to the extended data as extended data figure 3d,e and h. We've additionally visually elaborated the process of

identifying conservation levels in extended data 3a,b, and c. We've additionally updated the corresponding text for increased clarity.

11. *L223 states an informative conclusion, but the numbers of cCREs in each category listed in L223-228 cannot readily be deduced from the cited panels in Fig. 3.*

Response: To demonstrate that the number of cCREs decreased with increasing conservation level, we have added figure 2b, which displays the numbers in parentheses.

12. *Fig. 3b: It is not clear what the second set of bar graphs ("level 1") are showing. Is this figure supposed to support the observation on L224 that "66,781 (32.5%) were classified as level 1"? That number is not apparent in any of the three graphs. Why use 3 graphs to support one number? What are the partial levels?*

Response: We added a Venn diagram to illustrate relationships of inclusion or overlap among cCREs in different levels (Fig. 2b) in a simpler way than the previous three bar plots. The partial levels show pairwise conservation number between human and each species (e.g. a peak that switches from neuron to astrocyte activity in between human and marmoset). To preserve this information we moved the bar graphs into supplemental data (Ext. Data. Fig. 3d).

13. *Fig. 3c: This complex figure is difficult to interpret, and it is not clear what messages are being conveyed. Neither the Results nor the legend provide the needed information. Do the species images in the middle refer to the cell types on the outer circle? What is learned from the breakdown of the results by cell types? What conclusion can be drawn from the inner circles, and can it be shown in a less complicated figure panel? Is this figure supposed to support the observation on L224 that "49,135 (24.0%) as level 2"? That number is one value in very small print on one of the inner circles; this information could be conveyed in a MUCH more accessible manner.*

Response: We appreciate this valuable feedback. We have added diagrams to provide a clearer explanation of how the levels are classified (Ext. Data. Fig. 3a-c). We have made the figure into a simplified Venn diagram that provides an overview of level 2 cCREs, without displaying individual proportions for each cell type. However, we retained the detailed level 2 figure for each cell type in the supplemental data (Ext. Data. Fig. 3e).

14. *Fig. 3d: This panel uses the divergence index versus conservation index spectrum again, apparently aiming to support the statement on L225 of "16,068 (7.8%) as level 3 epi-conservation". That number is provided as a inset to one of the spectra (graphs). The actual data shown are not for cCREs with level 3 epi-conservation, but rather it is for all the cCREs with level 0 sequence conservation. One cannot deduce which points in the spectra correspond to the cCREs in the category of interest (level 3 epi-conservation).*

Response: Thank you for your feedback. We have updated the figure to clarify that the figure shows all level 0 cCREs, and have reworked the figure to highlight both mammal level 3 and human biased peaks in panels Fig. 2C and Fig 2d.

15. *The data in Fig. 3h are shown at a high level of granularity, with a full matrix of TE classes versus cell types. However, the results as presented are difficult to align with the statements in the text (L249-252). For example, the text says that "Human-specific cCREs from excitatory neurons (except for L5/6 NP and L6 CT) are most enriched in L1", but the image in Fig. 3h shows a strong enrichment for these TEs across all the cell types. The more general groups of excitatory neurons and glial cells are not explicitly labeled in the figure panel. A simpler figure focused on bringing out the key result would be more effective.*

Response: We've updated the Figure (now Extended Data Fig. 4a,b) to emphasize our conclusions on the dynamics between cells. We've added major cell type class labels below 4a, and a separate panel with bar plots for ERV1, ERVK, and L1 between IT neurons and glia. We've updated the text for additional clarity: "Human-specific cCREs from intratelencephalic (IT) excitatory neurons had the highest overlap within L1, while glial cells (MGC, OPC, ODC, and ASC) had the highest overlap with ERV1 and ERVK LTRs (Extended Data Fig. 4a,b)."

16. *L228-230: The criteria for categorizing cCREs as having "human-biased" accessibility are not clear. The level 0 cCREs have a peak of accessibility only in human despite having orthologous sequences in other species. How do human-biased accessibility peaks differ from these?*

Response: Level 0 cCREs include any cCRE identified in human that has sequence conservation in all four species. Epigenetic conserved cCREs (level1,2,3) are a subset of level 0 cCREs. The difference between level 0 cCREs that aren't level 1, 2, or 3 and Human-biased cCREs is a statistical cut off for being significantly more accessible in human compared to all other species for a given cell type. Additionally, level 0 Peaks not represented in human biased peaks, or higher levels may have a corresponding peak in just one other species, or may not have strong enough signal to be considered differentially accessible. We have updated the text to emphasize this difference: "Analogous to species-biased genes, species-biased cCREs are defined as peaks with differential accessibility that are consistently higher in one species compared to the three other species in the same cell type as identified through differential accessibility analysis performed using EdgeR(Robinson et al. 2010) (methods)."

17. *Pages 8-9 and Fig 4: Many of the problems of unclear and overly complex figure panels described for the analysis of ATAC-seq peaks apply to this analysis of differentially methylated regions (DMRs).*

Response: We have addressed the problems with clarity by moving the previous Fig. 3a,b into supplemental data (Ext. Data. Fig. 5a,b) and replaced it with a simplified Venn diagram in main figures (Fig. 3a). We have moved figures for primate levels in the previous Fig. 3c,d,f into supplemental data (Ext. Data. Fig. 5f,g)

18. *It is not clear how panels e and f for Fig. 4 support the statement on L280, "elevated levels of conservation correlate with lower cell-type specificity".*

Response: Thank you for identifying the issue. The correct figure to reference is now Fig 3d. and we have updated the text.

19. *In several places, the "cell-type specificity" is compared for a particular modality across categories, such as levels of epigenetic conservation. The relevant data are shown as a distribution over a range of entropy values, such as in Fig. 3e and Fig. 4d. Given that there are many values for the specificity-related entropy, the manuscript should clarify how the distributions were compared to reach conclusions such as "the level 3 conserved cCREs that are promoter distal have reduced cell type specificity" (L238) and "elevated levels of conservation correlate with lower cell-type specificity" (L280).*

Response: To evaluate the differences in cell type specificity, we have conducted T-tests between different conservation levels and updated the text accordingly. For example: "Furthermore, the level 3 conserved cCREs that are promoter distal have reduced cell type specificity compared to level 0 ($p < 2e-16$, T-test)". And "Much like chromatin accessible cCREs, promoter distal mammal and primate level 3 DMRs showed lower cell type specificity than level 0 ($p < 2e-16$, T-test)"

20. *Pages 9-10 and Fig 5: Many of the problems of unclear and overly complex figure panels described for the analysis of ATAC-seq peaks apply to this analysis of TAD boundaries and loops.*

Response: We appreciate this valuable feedback. As we have made changes to the analysis of ATAC-seq peaks (Fig. 2) and differentially methylated regions (Fig. 3), we replaced bar plots (previous Fig. 5b,c,h,i) with a simplified Venn diagram to represent the breakdown of boundaries and loops for each group (Fig. 4b,g)

21. *The text in L305-308 implies that rules similar to those used for ATAC-peaks and DMRs were used for TAD boundaries and chromatin loops. However, Fig. 5a and 5g introduce new categories, which may be "tissue conserved" and "cell type conserved" (the text is pixelated and difficult to read even at high magnification). The criteria for those categories are not explained, and the schematic figure panels are confusing. It is not apparent what distinguishes "tissue conserved" from "cell type conserved". L308 says that mammal level 2 in this context refers to TADs that are conserved in the same cell types, but Fig. 5a*

shows the TADs in this category at the same place in DIFFERENT cell types. Fig. 5 and accompanying text needs to be corrected and revised extensively to provide clear evidence to support the claims, and a legible revised figure needs to be provided.

Response: We thank the reviewer for pointing out the clarity issues in these figures. We apologize for having provided a low resolution image for this figure, and have made sure to address this with our resubmission. What we are illustrating is that level 1 boundaries are called across species but not required to be in the same cell type. While level 2 has the same boundary called in at least one of the same cell types. This matches our criteria for chromatin accessibility conservation levels. We've added additional clarity to the text: "Of the mammal level 0 (sequenced conserved) boundaries, we identified 40% (5,118) found in all four species from any cell type (mammal level 1, tissue-level conserved) (Fig. 45b, Extended Data Fig. 6a), and 10% (1,290) conserved in any of the same cell types in all four species (mammal level 2)."

22. *L352 suggests that cCREs assigned to a target gene by ABC are more likely to function as enhancers than are cCREs that are not assigned. However, the cCREs not assigned to a target gene by ABC could still act as enhancers, but they could be regulating one or more gene targets that are not revealed by the current chromatin conformation capture data. A more accurate conclusion is that "conserved cCREs are more readily assigned as enhancers for specific target genes". Similarly, the y-axis on Fig. 6d is the "Percent predicted to regulate a target gene".*

Response: Thank you for your clarifying statement. While it is true ABC aims to identify the gene regulatory targets of cCREs, the identified putative enhancer gene pairs are also enriched for enhancer function. Given ABCs integration of cCRE activity, and 3D contact we consider it a meaningful measure of putative enhancer function. Additionally, we have included single cell H3k27ac for human and mouse which shows concordant increases across categories with percent predicted as enhancers of a target gene (Fig 5a,b, Ext data Fig 10 b,c). We also directly show H3K27ac is significantly higher at predicted ABC enhancers than other cCREs (Extended data Fig 10, e). Taken together we believe this measure of function is appropriate.

23. *"Syntenic" is not used correctly. Syntenic genes are on the same chromosome ("same string"). Genes that are syntenic in one species may be on the same chromosome in a second species, in which case synteny is conserved. Beyond that, gene order and even arrangement can be conserved. The manuscript should clarify what is meant by "syntenic".*

Response: Thank you for clarifying the definition of syntenic! We've replaced uses of the word "syntenic" with "chromosomes with conserved sequence identity".

24. *The clause on L389-390 is hard to follow, because the training set consisted of orthologous regions in*

other species. What exactly is the risk that is being avoided? Also, in L387, "data leakage" is not a concept used widely by experimental scientists, and it would be helpful to explain what it means.

Response: Thank you for your comment, we have updated the text to clarify and better highlight the risks of data leakage:

"We first constructed testing and validation sets on chromosomes with conserved sequence identity from across all four species (Fig. 6b). This mitigates the risk of data leakage⁴⁰, caused by an orthologous region trained on in another species appearing in the test set. In such a case one might greatly overestimate the model's understanding and predictive ability when applied to unseen DNA"

We have additionally added a citation to "Navigating the pitfalls of applying machine learning in genomics". Whalen et al, 2022 Nature Reviews Genetics. Which describes issues like data leakage to a broader audience.

25. In Fig. 7, panels d-g are not described or discussed in the text. These panels have helpful information, showing an overall correspondence between predicted and actual chromatin accessibility signals, BUT with some notable false positives and false negatives. These rich landscapes are reduced to a single number, a correlation across bins, in panel h, which loses the variance in predicting accuracy. Was there anything distinctive about the regions that the model missed for chromatin accessibility or that it over-predicted for accessibility?

Response: Thank you for the feedback! We have adopted a new model which boasts improved predictive ability, and now reference the track level predictions in the text (Fig 6e,f).

While we weren't able to describe in specificity the difference between model false positives vs false negatives, evaluating model accuracy we find that human biased regions are slightly more difficult to predict (Extended data Fig 11a,c,d), as are intergenic peaks (Extended data Fig. E). And these trends are true both in single and multi-species contexts.

26. The "correlations across cell types" in Fig. 7 panels i and j (text L394-398) are not explained sufficiently for the results to be understood. If this correlation is related to the correlation across bins shown in panel h, how the values for the correlation become so much smaller? It is not clear what values are being correlated across cell types.

Response: Thank you for your comment. With an improved model comes improved cross-cell correlation. We have updated the text to clarify:

"For each peak in the test dataset, we correlated the model predictions to the true accessibility across cell types (methods)."

And in the methods:

“Cross-cell type evaluation:

To evaluate the model's ability to predict patterns of chromatin accessibility across cell types we subset to regions in the testing dataset overlapping all peaks called in each species genome. For each peak in the testing dataset, we calculate the correlation of the predicted cell type chromatin accessibility with the true chromatin accessibility at the same locus”

27. Ref. 2 should be journal article.

Response: Thank you for your careful investigation of the text. We have updated the citation.

To boost transparency, I sign this reviewer's report:

Ross Hardison

Response: Thank you, Dr. Hardison, for your extremely helpful comments!

Referee #2 (Remarks to the Author):

In Zemke et al, the authors seek to identify the evolutionary impact on gene regulatory organization in the neocortex. To do so, they apply multiomic measurements of transcription and chromatin state, methylation, and physical chromatin contacts in the mouse, marmoset, macaque, and human primary motor cortex. This is a monumental undertaking and one certainly worth doing. Not surprisingly, this is a well-executed genomic investigation, as the senior authors are experienced and prominent members of the genomics community. The authors identify differences in cellular composition between species, differences in cell-type specific enhancer usage and methylation state, and corresponding changes in nuclear architecture.

Response: Thank you for recognizing the importance of the present study.

Unfortunately, the biological insights of this study fall short of a significant advance. A tremendous amount of interesting data has been collected in this study and certainly can serve as a useful resource article, but it is unclear if Nature is the proper journal for publication. The manuscript drifts between related genomic measurements but is not effective in harmonizing these data sets to identify key

biological insights relevant to the evolution of gene regulatory strategies and grammar across species. Instead, a string of observations is made with no overarching biological conclusion or outcome.

Response: We are extremely grateful for the constructive comments from this reviewer, which have prompted us to perform many new integrative analyses, obtain new biological insights into the evolution of cortical gene regulatory programs in mammals and primates. The new analyses and biological insights are briefly summarized below and discussed in more detail in the point-by-point responses:

- a) Through integrative analysis of transcription factor gene expression patterns and chromatin accessibility in the 21 cell types across four species, we showed that divergent expression patterns of transcription factors correlated with divergent utilization of cCREs in each cell type (Figure 2i), providing evidence for the role of evolving transcription factors in the evolution of cCREs in different cell types. This hypothesis is further supported by the observation that conservation of TF binding motifs is correlated with the conservation of chromatin accessibility and DNA methylation patterns across species (new Figure 2j, Figure 3f). This analysis highlights the evolution of both transcription factor expression levels and their DNA binding sites as contributors of species specific gene expression programs.
- b) We further underscore this point by carrying out in-depth integrative analysis of evolution of enhancer-gene pairs in different cell types and species. Our results revealed 464 human biased enhancer-gene pairs that include many genes involved in human disease such as FOXP2 and RYR3 (Figure 5e-h). The list of human biased genes and their predicted enhancers will be valuable for study of the molecular and cellular basis of human specific cognitive traits (such as language).
- c) We have used the single cell multimodal datasets to train a machine-learning model based on deep neural network (Basenji), and achieved remarkable performance (new Figure 6). The sequence-based predictors of chromatin accessibility match incredibly well the actual open chromatin landscapes in different brain cell types and species. We showed that the key to achieving this high level of performance is the use of both ATAC-seq and DNA methylation data from multiple species. We believe that the deep learning models developed in this study would be valuable for understanding the evolution of gene regulatory programs, and interpretation of noncoding disease risk variants.
- d) We wish to also highlight another new insight gained in this study. We provided multiple lines of evidence that evolutionary conservation of epigenetic state is strongly associated with the functional role of the cCREs. Noncoding variants in these cCREs are contributing to much higher levels of heritability of human disease than variants outside of the epi-conserved cCREs. This finding would enable investigators to prioritize the noncoding variants associated with disease and traits in their pursuit of causal disease variants.

In the interest of improving this manuscript, the authors should consider addressing the below issues.

1) In Figure 1, the authors show harmonization between 10X multiome RNA and snm3C-seq data. It is difficult to get a handle on what the authors seek to demonstrate in this figure. Compositional differences are observed, with little commentary on what this actually means in context of the structure and function of motor circuitry.

Response: Thank you for the comments on the presentation of Figure 1. We agree that the initial submission was confusing. We had two primary purposes in mind for this figure. In the new figure 1 we try to illustrate the experimental design and main datasets generated. We also aim to demonstrate the importance of cell type resolved datasets when making cross-species comparisons. We would like to highlight the huge cell type compositional differences of cortex across species, and show how a bulk tissue level comparison would be difficult to interpret at the molecular level. We now merge original Figure 1 and Figure 2 to further illustrate how we compute the conservation and divergence of gene expression patterns across species.

Is there a key take home regarding patterns of methylation, accessibility and transcriptional activation that can explain the variance observed across species? Presumably this is handled in Figure 2, but we are left with gene ontologies and little explanation about how enhancer differences can explain variability in transcriptional activation. The authors show gene ontologies of conserved groups, but can they offer insight into species specific ontological processes and the molecular etiology associated with functional divergence. More broadly, is there any indication that errors in peak-gene association (often done in multiome data) to identify enhancers may simply account for some differences that are observed across species? Can data from these sequencing modalities be integrated effectively to address that concern?

Response: This is a very important comment. To address it we have performed additional integrative analysis of the multimodal datasets to provide insight into the molecular processes associated with evolution divergence of gene expression patterns. We highlight the potential role of species divergent transcription factor expression in shaping species divergent chromatin accessibility (Fig. 2i) and as putative drivers of human divergent gene expression (Fig. 2h). We have added analysis where we assigned human divergent putative enhancers to human biased genes to identify enhancers driving human divergent gene expression and highlighted specific examples (Fig. 5g,h,i, Extended Data Fig. 8l). Accurately assigning enhancers to gene targets remains challenging, which is why we've taken an integrated approach that combines chromatin accessibility, gene expression, and chromatin contacts with the ABC model. While we do not have ground truth for our enhancer-gene predictions, we corroborate the predictions by integrating newly generated H3K27ac Paired-Tag data from human M1 (Fig. 5b,g,h,i, Extended Data Fig. 8l). We also demonstrated the relationship between conservation of 3D genome to gene expression conservation (Fig. 4e), and in new analysis of boundaries, we find human divergent boundaries are associated with more divergent gene expression than conserved boundaries (Extended data Fig. 6c).

2) I find the categorization and nomenclature of conservation levels is difficult to handle in this

manuscript. What are the thresholds for this categorization? How “epi-conserved” does a site need to be to be placed in a given conservation level?

Response: Thank you for your insight! We agree that the description of conservation levels in the original submission was confusing. In the revision, we have made strong efforts to revise the presentation to make the points clearer. For visual clarity of the relationship between levels we’ve included simple Venn diagrams (Fig. 2b, Fig. 3a, Fig. 4b,g, 5e), and added a detailed workflow into Extended Data fig. 3a. Level 1 and level 2 are simply binary comparisons (same peak is called in any cell types across species for level1 and same peak is called in any of the same cell types across species for level 2). Level 3 requires the signals of matching cell types to be significantly similar by GLS with a threshold of adjusted $P < 0.05$. To explain our criteria for each level, we have the following in the text: *“level 1” are tissue conserved cCREs with a peak called across species regardless of which cell type, “level 2” are cCREs displaying accessibility in at least one of the same cell types across species, and “level 3” are cCREs with matching patterns of chromatin accessibility across all the cell types, as measured to be significant (BY adjusted $P < 0.05$) by GLS.”*

The authors might want to consider clarifying in the text the significance of these conservation levels and integrate other sequencing modalities (methylome and chromatin contact) into the interpretation of these results.

Response: Thank you for this very helpful suggestion! To address this point we’ve added conservation comparison between chromatin accessibility and methylome (Fig. 3b). We have already included comparisons of conservation between chromatin contacts and TAD boundaries with methylation for each conservation level (Fig. 4e,h) since the levels of chromatin contacts are only binary calls.

In figure 3b, the authors show “Level 0 conserved elements, indicating 8K human specific elements, can they offer gene ontology for these categories?”

Response: Thank you for the suggestion! We’ve performed Gene Ontology analysis for human specific elements using nearby gene association (GREAT) and included these results here. While we find significant great enrichment for olfactory genes, we don’t feel these should be included in the manuscript since they are not expressed in motor cortex cells. Using ABC targets of these sequences we find no enrichment.

Figure 3d is labeled with “Level 3” but the figure caption refers to a “Level 0” comparison between mammalian conservation and primate conservation. If the result is for “Level 3” conservation, then the authors should explain how strong “epi-conservation” could lead to such divergent results.

Response: Thank you for your suggestion! We apologize for the confusing figure. We’ve made updates to the figure (now Fig. 2c) and clarified the legend. The new figure has the mammal level 3 elements highlighted to distinguish them from all other elements.

Figure 3g and 3h are the most interesting observations in this manuscript. That said, mammalian “rewiring” by TEs been observed in Choudhary et al (Nature Communications 14, 634 (2023)). Basically, there doesn’t appear to be an overwhelming amount of variability in TEs in cell types shown in figure 3h. Perhaps the authors would uncover exciting biology in figure 3h by showing how cell types change in %cCREs in TEs across species? Unfortunately, an exciting result is diminished by a lack of substantive variation in TEs across cell types, bringing into question what insights have been made beyond what could be gained from bulk experiments.

Response: We’ve taken the reviewer’s suggestion by adjusting the figure to better display the dynamics between cell types (now Extended Data Fig. 4a,b). We’ve included a new panel that focuses on ERV1, ERVK, and L1 that clearly shows the differences between IT neurons and glia (Extended Data Fig. 4b). We’ve updated the text for more clarity on our take away message: “Human-specific cCREs from intratellencephalic (IT) excitatory neurons had the highest overlap within L1, while glial cells (MGC, OPC, ODC, and ASC) had the highest overlap with ERV1 and ERVK LTRs (Extended Data Fig. 4a,b).” We now display human TE enrichment next to TE enrichment for mouse cCREs (Extended Data Fig. 4d), making it easier to appreciate the species-specific enrichment, particularly for LTR TEs.

3) In figure 4, the authors profile the methylome across species. How do these results relate to the

findings of figure 3? For example, are there patterns of methylation that might explain genomic silencing that buffers evolutionary differences and results in conservation across species?

Response: To address this question, we've added a comparison of conservation between chromatin accessibility and methylome (Fig. 3b). We've additionally compared the DNA sequence motif basis of DMRs and chromatin accessibility revealing impressive correspondence (Fig. 3f).

There appears to be a trend where chromatin accessibility in figure 3 is more conserved in proximal elements (shown in level 3 conserved elements) and the trend is similar in the methylome, albeit to a lesser extent. What is the interpretation of these findings? How does it relate to regulatory convergence and divergence across species?

Response: Our interpretation on the discrepancy between promoter proximal conservation between chromatin accessible cCREs and DMRs is that this is due to the definition of DMRs being differential between cell types. Promoter proximal elements tend to have broad cell type activities, so there is a de-enrichment for DMRs being called at promoters unless they are cell type differential, which makes them less likely to be conserved. This results in a lower percentage in DMRs that are conserved compared to chromatin accessibility cCREs.

In figure 4, can the authors offer insight into human biased methylation and mammalian conserved patterns similar to figure 3f?

Response: Differences in global methylation patterns across species confound such analyses, even between human and chimpanzee confounding such analyses (Zeng et al. 2012; Blake et al. 2020). Additionally, current methods to identify DMRs are based on calling differences at individual CPG sites. Because we measured conservation at a DMR level it would be difficult to align these differences. We have however, highlighted the patterns of conserved DMRs across mammals and primates (Fig. 3c, Extended Data Fig. 5f).

Again, are there instances where methylation abrogates the contribution of human specific accessibility? Ultimately, the authors need to integrate these sequencing modalities more effectively with one another to explain the impact of the differences they observe across species.

Response: Thank you for this suggestion. To address the potential impact of DNA methylation on human sequence-specific cCRE activity we examined DNA methylation levels at DMRs overlapping human peaks (Extended Data Fig. 5h). While there is a considerable increase in methylation level at these human-specific sequences relative to other cCREs, it's beyond the scope of our study to say if the methylation has an abrogating effect.

Again, the authors share insight about the % of DMRs overlapping TEs, but there appears to be little variability across cell types in the human. Can they show differences across species and show cell type

specific changes in DMRs? How do these data, alongside differences in accessibility, yield differences and similarities in transcriptional activation or lack thereof?

Response: We agree that for most classes of TEs there is generally low variability across cell types and we have moved this panel to Extended Data Fig. 5. However, high variability is seen for the human specific sequence DMRs in Alu elements, where there is a clear depletion for non neurons compared to neurons. We've pointed this out in the text by adding "Human specific hypomethylated DMRs in Alu elements showed preference for neurons (Extended Data Fig. 5e)."

4) The authors continue their work looking at conserved and epi-conserved TAD boundaries across species. An interesting observation is that conserved boundaries yield conserved transcriptional profiles. Can the authors expand their interpretation of these results.

Response: Thank you for this suggestion! We've expanded on our interpretation by adding: "Our results suggest a co-evolution of the 3D genome along with epigenome and gene expression, where evolutionary constraints are placed on the 3D genome in order to preserve gene expression." We've also expanded on this interesting finding by showing that divergent TAD boundaries yield divergent transcriptional profiles (Extended Data Fig. 6c).

CTCF is known to be sensitive to methylation. Can the authors rationalize any of their observed differences in TAD formation, with changes in the methylome binding? Granted, within a given species ~1.5% of CTCF sites are influenced by methylation. Are there differences of these effects across species and could that explain differences in TAD boundaries and subsequent transcriptional differences?

Response: Thank you for the great suggestion. We added comparisons of species DNA methylation patterns at CTCF sites in conserved and divergent boundaries (Extended Data Fig. 6f). Additionally, we performed other analysis investigating the contribution of CTCF to divergent and conserved boundaries by quantifying their overlap with CTCF motifs and CHIP-seq peaks (Extended Data Fig. 6d,e).

5) In Figure 6, the authors identify epigenetic conservation at cCREs and identify predicted conserved enhancers and their influence on transcriptional conservation.

The authors have published single cell methods that can offer significant value in these studies, profiling chromatin state (enhancers) and transcriptional activation simultaneously and in single cells (Nature Methods, 2021). I think it is essential that the authors show agreement between their predicted enhancer classes and their agreement with experimental proof that there are bona fide enhancers that activate transcription across species.

Response: We agree with reviewer's suggestion that adding active chromatin state profiling data would add evidence for our enhancer analysis. During the revision, we have added single cell H3K27ac + gene expression profiles by performing Droplet Paired-Tag data from human M1 tissue and reanalyzed

recently published mouse Droplet Paired-Tag cortex data (Extended Data Fig. 10a). The new cell-type resolved H3K27ac data strengthened our claim that chromatin accessible conserved distal cCREs are more likely to act as enhancers (Fig. 5b) and our ABC predicted enhancers have higher H3K27ac levels than cCREs not predicted as enhancers (Extended Data Fig. 10e).

I recognize this is an undertaking but given that the method was pioneered by the senior investigator, a modest attempt at validation is reasonable. Speaking more generally, throughout the manuscript, the authors appear to avoid a key question: what is the significance of species specific TEs that are not enhancers? What functionality do they offer and how might this impact the evolution of transcriptional regulation?

Response: We thank the reviewer for their insightful comment and important questions. From our current analysis we have determined that cCREs at TEs are less often predicted as enhancers via ABC model or correlation linkage between chromatin accessibility and RNA (not shown), and TEs have lower H3K27ac, even when accounting for mappability. Furthermore, human specific sequences that are at TEs have less enhancer predictability than human specific sequences not at TEs (Extended Data Fig. 10e). Taken together, TEs are less likely to be functioning as enhancers. And for those TEs not predicted as enhancers we can only speculate that they may be neutral passengers of epigenome evolution. We feel any further speculation would be inappropriate without more direct evidence on the matter.

6) In Figure 7, the authors modify a deep learning approach to predict cell type specific patterns of chromatin accessibility. Can the authors comment on this approach in context of other comparative studies of enhancer identification and impact, notably TACIT (Kaplow et al. Science, vol 380, no 6643).

Response: . Thank you for this reference, we have made reference to it in the revised text “**These datasets increase power for modeling the epigenome and our ability to link sequence changes to phenotype (Kaplow et al. 2023).**”

This section of the manuscript, frankly, feels like a distraction. It is unclear how this relates to the broader study and, furthermore, the authors should consider identifying chromatin accessibility using this deep learning model in an out group and then then conduct a simple snATAC-Seq experiment to demonstrate that this approach is general. If the idea is that this resource offers value towards the interrogation of other species, with predictive power, then an out-group study is essential.

Response: Thank you for your thoughtful comment! In the revision, we have thoroughly revised the deep-learning models, by adopting the Basenji framework, a popular neural network architecture. To assess our model’s predictive ability on an outgroup, we trained the model with each species as an outgroup (e.g. the mouse outgroup model was trained on human, macaque and marmoset data). To demonstrate the model performance under evolutionary dissimilarity, we perform predictions using the least accurate species-specific predictor for each model. Under these conditions, predictions of relative activity within a cell type remain high, but the ability of the model to distinguish patterns across cell

types fall notably (Extended Data Fig 11f,g).

7) The authors end this study with an important statement that these comparative studies could offer insights into noncoding genetic variation related to neurological disease and disorders. The authors show that a conserved regulatory element upstream of ELMO1 intersects with risk variant rs606000003, which is exclusively accessible in microglia. There is the suggestion that expression of ELMO1 may be impacted in a cell type specific manner that contributes to MS. While this may offer some rationale for prioritization of noncoding variation, the authors have tools to test this through a single cell CUT&Tag measurement of differential binding of the regulator that binds at this impacted cognate binding site.

Response: We agree that it would be ideal to corroborate this predicted enhancer with an orthogonal approach. Unfortunately, single cell CUT&Tag is not suitable to profile DNA binding proteins at least in our hands. As an alternative, we have performed droplet Paired-Tag targeting the active enhancer mark, H3K27ac. This experiment revealed a preference for H3K27ac at this element in microglia cells (Extended Data Fig. 12b).

Furthermore, it remains unclear how epigenetic conservation offered more value than a simple multiomic measurement.

Response: Thank you for this important comment! We recognize that we did not make this point clear in the initial submission. We have revised the results and discussion to make this point clearer. In short, by looking for variants in epigenetically conserved peaks, we gain greater power to associate a given trait with a particular cell type, and can allow investigators to prioritize associations which can otherwise be extremely noisy.

Summary

There are several claims made in the abstract of this manuscript. Namely: 1) Cell type specific gene expression evolves more rapidly than broadly expressed genes, 2) Epigenetic status at distal candidate cCREs evolves faster than promoters and 3) That transposable elements contribute to 80% of human specific cCREs in the primary motor cortex. I find that claims 1 and 2 are weakly supported and claim 3, which is interesting, is difficult to biologically interpret given the authors statement that "In general, cCREs located in TEs are less often predicted to function as enhancers."

The authors should consider toning down language regarding "evolutionary speed" given that it is not addressed formally within the manuscript.

Response: Thank you for your insightful comments and helpful suggestions! We have updated the text to better reflect what was formally evaluated. We have change the abstract to say:

We find that cell type-specific gene expression has less evolutionary conservation than broadly expressed genes and that epigenetic status at distal candidate cis-regulatory elements (cCREs) is highly divergent.

This suggests that the turnover rates are different between the promoter-proximal and distal cCREs during evolution, where distal cCREs have much lower evolutionary constraint than proximal cCREs.

Finally, the authors should be commended for pointing out that “Previous bulk sequencing assays have revealed general principles concerning the conservation of cis-regulatory elements and tissue-specific gene expression patterns. For example, enhancers exhibit rapid turnover during mammalian evolution and those conserved have lower cell type specific activity. In contrast, sequence divergent enhancers play a significant role in establishing tissue and species-specific traits. Such divergent enhancers are often mediated by de novo insertion of transposable elements carrying clusters of transcription factor binding sites.” The authors follow up with references supporting the aforementioned. Unfortunately, the bulk studies demonstrate biological phenomena that are seen in this study as well, bringing into question the biological novelty of this study.

Basically, I struggled with the question of whether the biological significance of this study is sufficient to justify publication as a Nature article, or should be published as a data resource. I am happy to opine on a revised manuscript that addresses my concerns or to be convinced that my concerns are unreasonable. I would hope that with all of the intellectual firepower in the authors list, more compelling, and novel biological insights could be provided based on this extraordinary data set.

Response: Thank you for your comments, which are fair and thoughtful! We took them in heart and have carried out extensive new analyses and assays in order to more clearly demonstrate the value of the current study and the novelty of the findings. Below we would like highlight again the novel biological lessons that we have gained through this work:

- The dataset is an extraordinary resource for investigation of the evolution of gene regulatory programs in mammalian and primate brain cells.
- We provide new evidence for the role of evolving transcription factor expression patterns and their binding motifs in the evolution of gene regulatory programs in different cell types. We observe that degree of conservation of TF binding motifs is correlated with that of chromatin accessibility and DNA methylation patterns across species (new Figure 2j, Figure 3f).
- We revealed 464 human biased enhancer-gene pairs that include many genes involved in human disease such as *FOXP2* and *RYR3* (Figure 5e-h). The list of human biased genes and their predicted enhancers will be valuable for study of the molecular and cellular basis of human specific cognitive traits (such as language).
- We have trained a deep neural network (Basenji) with the use of both ATAC-seq and DNA methylation data from multiple species to develop the sequence-based predictors of chromatin accessibility. The predictions match incredibly well the actual open chromatin landscapes in different brain cell types and species. We believe that this deep learning model would be valuable for understanding the evolution of gene regulatory programs, and interpretation of noncoding disease risk variants.

- We provided multiple lines of evidence that evolutionary conservation of epigenetic state is strongly associated with the functional role of the cCREs. Noncoding variants in these cCREs are contributing to much higher levels of heritability of human disease than variants outside of the epi-conserved cCREs. This finding would enable investigators to prioritize the noncoding variants associated with disease and traits in their pursuit of causal disease variants.

Reviewer Reports on the First Revision:

Referees' comments:

Referee #1 (Remarks to the Author):

The manuscript has been extensively revised to improve clarity, with simpler and easily interpreted figures. Many additional analyses were performed to integrate information across the modalities. These new and expanded analyses reveal and emphasize the important biological messages that can be derived from the large and very impressive datasets reported. The revised text does a good job of emphasizing the major biological messages. Illustrative examples are provided to show how both human-specific and epigenetically conserved elements can be used to improve the interpretation of genetic variants associated with traits and diseases. Overall, the revisions address well the issues raised in the initial review, giving a manuscript that communicates well the importance of the data and analyses reported.

It looks like "Figure 8" in the text is actually the new Figure 7.

To boost transparency, I sign this reviewer's report:
Ross Hardison

Referee #2 (Remarks to the Author):

In my original review, I cited three overarching concerns with this manuscript:

1) Readability of the study

Much has been done address the concerns of both reviewers

2) Biological Novelty and Insight

This remains a problem, especially regarding species specific, cell type restricted patterns of gene regulation, in spite of considerable effort. For example, the authors cite the existence of 7,532 human specific cCREs. They utilize paired-seq to identify active enhancer elements and transcriptional activation. How many of these are identified by paired seq? What fraction of ABC predicted enhancers are confirmed by paired seq? The authors should provide more explicit information about the overlap of their paired seq data. Given the relative differences in cellular stoichiometry reported in Figure 1A. Is it possible that sparsity of ATAC signal is cell type specific (in a species dependent manner)? If so, how does this impact analysis and interpretation of paired seq overlap with predicted enhancers and subsequent analysis of regulatory divergence? The authors should clarify how the extended data of Figure 10a is affected by potential false positives discovered by the ABC model and how paired seq constrains subsequent analyses. There is a thin line between a "data resource" paper and novel insights required for publication in Nature.

For example, human specific enrichment of "negative regulation of blood vessel morphogenesis" in OPCs is highlighted as evidence for functional significance. However, the authors are referring to 4 genes (out of a total of 199 human genes within this GO category). Furthermore, the significance

of statements about human restricted regulation of FOXP2 and RYR3 in microglia and astrocytes, respectively, are mentioned, but not well explained. In addition, an interesting point is made about human biased, non-ubiquitous, expression of extracellular matrix genes. However, there is no information regarding enrichment in any given cell type, nor meaningful insight into how species actually regulate this process differently. A major update in the author response is: "We showed that divergent transcription factor gene expression likely promotes species-biased gene expression patterns in corresponding cell types." I cannot identify how this claim is supported effectively in the manuscript, particularly in the examples cited. This speaks to my general struggle with assessing the biological significance of the manuscript; the amount of work performed is exhaustive and impressive, and much effort has been made to address concerns of functional significance. If the primary observation of this study is: Integrating RNA-Seq, HiC and ATAC-Seq provides enhanced detection of regulatory variation, this has been done before (Zhang et al., 2022, Neuron 110, 992–1008). If the observation is that transposable elements drive evolutionary innovations and phenotypic outcomes through CREs, this is also well documented (Cell, 24 September 2015, Pages 68–88). This work does clearly show the existence of differences between species at different levels of gene regulation, but in its current form focuses heavily on narrow examples and two-way comparisons, thereby lacking a novel biological insight.

3) The multimodal data offered in this study are extraordinary. However, they are not integrated sufficiently to coalesce into a clear advance in our understanding of the evolution of gene regulatory mechanisms. I understand the difficulty associated with integrating single cell RNA and ATAC-seq measurements with methylome and assays for nuclear architecture. Absent an integrated analysis, however, this study results in disjointed narrative lacking a cohesive biological outcome. As I mentioned in my previous remarks, this is technically a well-executed study, and offers an extraordinary data resource to the scientific community. The authors have tried to identify an evolutionary nugget, but I believe have fallen short.

I believe that this is a general problem with the state of single cell multiomic studies – massive amounts of sparse data, and a struggle to find biological insights. There is no question that this study is well designed and executed, and the data set novel (primate evolution), but in the end it falls short of providing fundamentally novel and rigorous biological insights. The editors of Nature must decide whether to publish data resources (as does Cell and other journals), which is clearly valuable, and state of the art – but falls short of the fundamental biological advances expected of a Nature article.

Author Rebuttals to First Revision Comments:

We are grateful to both reviewers for their constructive comments throughout the revision process! . The reviewer comments are pasted below in *blue italic* font followed by our responses in black regular font.

Referee #1 (Remarks to the Author):

The manuscript has been extensively revised to improve clarity, with simpler and easily interpreted figures. Many additional analyses were performed to integrate information across the modalities. These new and expanded analyses reveal and emphasize the important biological messages that can be derived from the large and very impressive datasets reported. The revised text does a good job of emphasizing the major biological messages. Illustrative examples are provided to show how both human-specific and epigenetically conserved elements can be used to improve the interpretation of genetic variants associated with traits and diseases. Overall, the revisions address well the issues raised in the initial review, giving a manuscript that communicates well the importance of the data and analyses reported.

It looks like "Figure 8" in the text is actually the new Figure 7.

To boost transparency, I sign this reviewer's report:
Ross Hardison

Response: We are thankful for the kind feedback on our previous revision. We sincerely apologize for mis-referencing figure 7 as figure 8 and have revised the text to address this.

Referee #2 (Remarks to the Author):

In my original review, I cited three overarching concerns with this manuscript:

1) Readability of the study

Much has been done address the concerns of both reviewers

Response: Thank you for your very helpful feedback throughout the revision!

2) Biological Novelty and Insight

This remains a problem, especially regarding species specific, cell type restricted patterns of gene regulation, in spite of considerable effort. For example, the authors cite the existence of 7,532 human specific cCREs. They utilize paired-seq to identify active enhancer elements and transcriptional activation. How many of these are identified by paired seq?

Response: We presume that the reviewer was referring to the Paired-Tag assays when he/she mentioned Paired-seq, which was actually equivalent to 10x multiome (ATAC/RNA-seq). Paired-Tag enables jointly profiling of histone modification and gene expression in single cells, and was used in the present study to assay H3K27ac distribution along the genome in different cell types in human brain tissues, as a way to corroborate the functional interpretation of cCREs. We found human-specific cCREs are generally depleted for H3K27AC compared to conserved elements (Fig. 5b). In contrast we find concordant patterns of H3k27AC, and open chromatin at human Biased regions (Fig 5b, h, Extended Data Fig. 10b).

What fraction of ABC predicted enhancers are confirmed by paired seq? The authors should provide more explicit information about the overlap of their paired seq data.

Response: We highlight concordant changes in distal enhancer ABC predictions, and H3k27AC, in panels (Fig5b,h Extended Data Fig 10b,e).

Given the relative differences in cellular stoichiometry reported in Figure 1A. Is it possible that sparsity of ATAC signal is cell type specific (in a species dependent manner)? If so, how does this impact analysis and interpretation of paired seq overlap with predicted enhancers and subsequent analysis of regulatory divergence?

Response: While there are known differences in ATAC-seq signal between single neurons and glia (e.g. Neurons show a larger number of open peaks), this trend is consistent in humans and mice giving us no reason to hypothesize species-specific differences. Paired-tag uses a modified ProteinA Tn5 to tag antibody proximal regions. Given the common enzyme used between the two assays we expect no negative impact on the overlap of the assays.

The authors should clarify how the extended data of Figure 10a is affected by potential false positives discovered by the ABC model and how paired seq constrains subsequent analyses.

Response: The droplet paired-tag dataset generated in Extended Data Fig. 10a contains both RNA and H3K27ac measurements. Extended data Fig. 10a has no reliance on ABC to generate any results, rather we employ RNA-based reference mapping as a bridge between datasets independent of any epigenetic modality.

There is a thin line between a “data resource” paper and novel insights required for publication in Nature.

Response: We do believe that our work not only represents a valuable resource for the community, but also provides novel biological insight into the divergent and conserved gene regulatory programs in mammalian neocortex. We have tried to emphasize these novel biological insights in the discussion session.

For example, human specific enrichment of “negative regulation of blood vessel morphogenesis” in OPCs is highlighted as evidence for functional significance. However, the authors are referring to 4 genes (out of a total of 199 human genes within this GO category).

Response: While we respect the reviewer's eye for rigor in results, we have taken extensive measures to conduct rigorous and statistically controlled analysis. The genes in question display a high level of divergence across many comparisons, and represent 8% of the non-ubiquitous human-biased OPC genes. Across all species only 78 of the genes in this category are expressed in OPCs.

Furthermore, the significance of statements about human restricted regulation of FOXP2 and RYR3 in microglia and astrocytes, respectively, are mentioned, but not well explained. In addition, an interesting point is made about human biased, non-ubiquitous, expression of extracellular matrix genes. However, there is no information regarding enrichment in any given cell type, nor meaningful insight into how species actually regulate this process differently. A major update in the author response is: “We showed that divergent transcription factor gene expression likely promotes species-biased gene expression patterns in corresponding cell types.” I cannot identify how this claim is supported effectively in the manuscript, particularly in the examples cited. This speaks to my general struggle with assessing the biological significance of the manuscript; the amount of work performed is exhaustive and impressive, and much effort has been made to address concerns of functional significance. If the primary observation of this study is: Integrating RNA-Seq, HiC and ATAC-Seq provides enhanced detection of regulatory variation, this has been done before (Zhang et al., 2022, Neuron 110, 992–1008).

Response: Again, we appreciate the reviewer's comments on the need to further improve the clarity of our presentation especially related to the statements related to the novel biological findings. In the revision we have indeed further clarified on these points, highlighting differences between the current work and previous studies including ones cited by the reviewer.

If the observation is that transposable elements drive evolutionary innovations and phenotypic outcomes through CREs, this is also well documented (Cell, 24 September 2015, Pages 68-88). This work does clearly show the existence of differences between species at different levels of gene regulation, but in its current form focuses heavily on narrow examples and two-way comparisons, thereby lacking a novel biological insight.

Response: Additionally, while the reviewer seems to have particular interest in human divergent biology, many of the important biological insights of this study pertain to conserved gene regulatory programs. Key results such as those in Fig. 7 demonstrate the utility of epigenetic

conservation as a lens to identify key variants, regulatory elements, and cell types in human disease. Fig. 5 demonstrates the utility of such analyses in identifying putative enhancers.

3) The multimodal data offered in this study are extraordinary. However, they are not integrated sufficiently to coalesce into a clear advance in our understanding of the evolution of gene regulatory mechanisms. I understand the difficulty associated with integrating single cell RNA and ATAC-seq measurements with methylome and assays for nuclear architecture. Absent an integrated analysis, however, this study results in disjointed narrative lacking a cohesive biological outcome. As I mentioned in my previous remarks, this is technically a well-executed study, and offers an extraordinary data resource to the scientific community. The authors have tried to identify an evolutionary nugget, but I believe have fallen short.

I believe that this is a general problem with the state of single cell multiomic studies – massive amounts of sparse data, and a struggle to find biological insights. There is no question that this study is well designed and executed, and the data set novel (primate evolution), but in the end it falls short of providing fundamentally novel and rigorous biological insights. The editors of Nature must decide whether to publish data resources (as does Cell and other journals), which is clearly valuable, and state of the art – but falls short of the fundamental biological advances expected of a Nature article.

Response: We thank the reviewer for highlighting the unique and valuable resource nature of the data.